# High-accuracy sampling from constrained spaces with the Metropolis-adjusted Preconditioned Langevin Algorithm

**Vishwak Srinivasan**                                              VISHWAKS@MIT.EDU
*Department of Electrical Engineering and Computer Science, MIT*

**Andre Wibisono**                                          ANDRE.WIBISONO@YALE.EDU
*Department of Computer Science, Yale University*

**Ashia Wilson**                                                   ASHIA07@MIT.EDU
*Department of Electrical Engineering and Computer Science, MIT*

**Editors:** Gautam Kamath and Po-Ling Loh

## Abstract

We propose a first-order sampling method called the Metropolis-adjusted Preconditioned Langevin Algorithm for approximate sampling from a target distribution whose support is a proper convex subset of $\mathbb{R}^d$. Our proposed method is the result of applying a Metropolis-Hastings filter to the Markov chain formed by a single step of the preconditioned Langevin algorithm with a metric $\mathscr{G}$, and is motivated by the natural gradient descent algorithm for optimisation. We derive non-asymptotic upper bounds for the mixing time of this method for sampling from target distributions whose potentials are bounded relative to $\mathscr{G}$, and for exponential distributions restricted to the support. Our analysis suggests that if $\mathscr{G}$ satisfies stronger notions of self-concordance introduced in Kook and Vempala (2024), then these mixing time upper bounds have a strictly better dependence on the dimension than when $\mathscr{G}$ is merely self-concordant. Our method is a *high-accuracy* sampler due to the polylogarithmic dependence on the error tolerance in our mixing time upper bounds.

**Keywords:** sampling, constrained spaces, preconditioning, Langevin algorithm

## 1. Introduction

Several statistical estimation and inference tasks involve drawing samples from a distribution; examples include estimating functionals, generating credible intervals for point estimates, and structured exploration of state spaces. The complexity of the distribution is influenced by the problem being studied, or by modelling choices made which can make it infeasible to sample from such distributions exactly. Markov chain Monte Carlo (MCMC) algorithms have helped tackle this challenge over the past several decades (Brooks et al., 2011), and have seen renewed interest in machine learning and statistics, especially in high-dimensional settings. These distributions of interest could be supported on the entire space (say $\mathbb{R}^d$), or on a proper subset of $\mathbb{R}^d$. Our focus is on the latter kind of distributions, which we refer to as *constrained*, and these arise in several practical problems. A non-exhaustive collection of applications are Bayesian modelling (Gelfand et al., 1992; Pakman and Paninski, 2014), regularised regression (Tian et al., 2008; Celeux et al., 2012), modelling metabolic networks (Heirendt et al., 2019), and differential privacy (Bassily et al., 2014).

Formally, we are interested in generating (approximate) samples from a distribution $\Pi$ with support $\mathcal{K} \subset \mathbb{R}^d$ that is convex, and density $\pi$ (with respect to the Lebesgue measure) of the form

$$\pi(x) \propto \exp(-f(x)) .$$

The function $f : \mathrm{int}(\mathcal{K}) \to \mathbb{R}$ is termed the *potential* of $\Pi$, and we refer to the above sampling problem as the **constrained sampling problem** henceforth. MCMC methods have been proposed and studied for the constrained sampling problem since at least as early as the 1980s (Smith, 1984). The earliest methods were originally catered towards generating points uniformly distributed over $\mathcal{K}$ (where $f \equiv 0$) with applications in volume computation. The aforementioned collection of applications also concern non-uniform distributions over $\mathcal{K}$.

In general, the task of sampling from a distribution $\Pi$ with potential $f$ supported on $\mathcal{K}$ can be transformed into a constrained sampling problem from a distribution $\widetilde{\Pi}$ whose potential is linear and supported over a larger domain through *lifting*. More precisely, suppose the potential $f$ satisfies $f = \sum_{i=1}^{N} f_i$ for a collection of functions $\{f_i\}_{i=1}^{N}$. Lifting defines a new domain $\widetilde{\mathcal{K}}$ and potential $\widetilde{f}$ (referred to as the *lifted domain* and *lifted potential* respectively) which are stated below.

$$\widetilde{\mathcal{K}} = \left\{ y = (x,t) : x \in \mathcal{K}, \ t \in \mathbb{R}^N, f_i(x) \le t_i \ \forall \ i \in [N] \right\} \ \widetilde{f}(y) = \widetilde{f}(x,t) = [\mathbf{0}_d^\top, \mathbf{1}_N^\top](y) .$$

We include more details about lifting in Section A.1. While lifting is a versatile technique that enables working with general convex potentials $f$, this is not necessary to deal with non-uniform distributions over $\mathcal{K}$, particularly in settings where the potential $f$ is "compatible" with geometric properties of the domain $\mathcal{K}$. We expand on such notions of compatibility in the sequel.

In this work, we propose a method for the constrained sampling problem, and our method is motivated by the rich connection between optimisation methods and sampling algorithms. The analogue of the constrained sampling problem in optimisation is the task of minimising $f$ over a constrained feasibility set ($\mathcal{K}$). For this constrained optimisation problem, it is usually beneficial to impose a non-Euclidean geometric structure over this feasibility set through a *metric* $\mathscr{G}$, and this defines a Riemannian manifold $(\mathcal{K}, \mathscr{G})$. Amari (1998) adapts the gradient descent method to incorporate the metric $\mathscr{G}$ as the Euclidean gradient $\nabla f$ no longer corresponds to the direction of steepest descent, and they refer to this new method as the *natural gradient descent* method. In this work, we propose a new MCMC method inspired by the natural gradient descent method called the *Metropolis-adjusted Preconditioned Langevin Algorithm* (MAPLA) for the constrained sampling problem. Each iteration of this algorithm is composed of the steps below.

(1) Generate a proposal $Z$ from $X$ (current iterate) with one step of PLA with metric $\mathscr{G}$.

(2) If $Z \notin \mathcal{K}$, reject $Z$ and set $X$ to be $X'$ (next iterate).

(3) If $Z \in \mathcal{K}$, compute the Metropolis-Hastings acceptance probability $p_{X \to Z}$.

(4) With probability $p_{X \to Z}$, set $X' = Z$ (accept); otherwise set $X' = X$ (reject).

Steps (3) and (4) are commonly referred to as the Metropolis-Hastings filter, and the above steps form the Metropolis adjustment of the Preconditioned Langevin Algorithm (PLA) with respect to $\Pi$. The Markov chain defined by MAPLA has two essential properties: it is both reversible with respect to $\Pi$ and ergodic. The reversibility is due to the fact that the Metropolis adjustment of any Markov chain with respect to an arbitrary distribution $\nu$ creates a new Markov chain that is reversible with

respect to $\nu$. The ergodicity of MAPLA follows from the ergodicity of PLA. Therefore, MAPLA is an *unbiased* MCMC algorithm; this means that the law of an iterate generated by MAPLA as the number of iterations tend to infinity is the target distribution $\Pi$. While MAPLA draws inspiration from natural gradient methods in optimisation, it is not the first MCMC method based on incorporating geometric properties of $\mathcal{K}$ encoded in the metric $\mathscr{G}$. DikinWalk and ManifoldMALA are two other methods that directly use the metric $\mathscr{G}$ in its design, and both of these methods are formed by including a Metropolis-Hastings filter to accept/reject proposals as well. In Table 1, we show the forms of the proposal distributions of these algorithms and that of MAPLA, which makes the relation of MAPLA to each of them more apparent.

Table 1: Proposal distributions $\mathcal{P}_X$ of certain algorithms. All of these proposal distributions are Gaussians with covariance $2h \cdot \mathscr{G}(X)^{-1}$ and only differ in the mean $\mathfrak{m}(X)$; $h > 0$ is the step size.

| Algorithm | Mean $\mathfrak{m}(X)$ |
|:---:|:---:|
| MAPLA 
 This work | $X - h \cdot \mathscr{G}(X)^{-1} \nabla f(X)$ |
| ManifoldMALA 
 Girolami and Calderhead (2011) | $X + h \cdot \{(\nabla \cdot \mathscr{G}^{-1})(X) - \mathscr{G}(X)^{-1} \nabla f(X)\}$ |
| DikinWalk 
 Kook and Vempala (2024) | $X$ |

From this comparison, we can view MAPLA as an interpolation between ManifoldMALA (based on the weighted Langevin algorithm introduced later) and DikinWalk (based on a geometric random walk with metric $\mathscr{G}$). When $\mathscr{G}$ is identity, DikinWalk is equivalent to the MetropolisRandomWalk (Mengersen and Tweedie, 1996), and both ManifoldMALA and MAPLA reduce to the Metropolis-adjusted Langevin Algorithm (MALA) (Roberts and Tweedie, 1996). The key difference between MAPLA and ManifoldMALA is the absence / presence of the $(\nabla \cdot \mathscr{G}^{-1})$ term[1] in the mean of the proposal distributions, respectively. The omission of this term in MAPLA is computationally advantageous, as this can be difficult to compute for choices of $\mathscr{G}$ suited for the constrained sampling problem, like for instance the Hessian of the log-barrier function of a polytope when $\mathcal{K}$ is that polytope. On the other hand, while the proposal distribution of DikinWalk is the simplest amongst the algorithms, it lacks any information about the target distribution $\Pi$, and hence MAPLA can be interpreted as performing a natural drift correction to DikinWalk. Another algorithm that is related to MAPLA is the Metropolis-adjusted Mirror Langevin algorithm (MAMLA) (Srinivasan et al., 2024) that was recently proposed and analysed for the constrained sampling problem. In the interest of space, we defer a discussion of this to Section A.2.

As noted previously, MAPLA is unbiased. The quality of a MCMC method is related to its *mixing time* that is defined as the minimum number of iterations of the method to generate an iterate whose distribution is "close" to the target distribution (see Section 4.1.3 for a precise definition). From a practical standpoint, it is useful to understand the effect of problem parameters such as the dimension $d$ and the properties of $\Pi$ on the mixing time of MAPLA. In this work, we also obtain non-asymptotic upper bounds for the mixing time of MAPLA under certain sufficient conditions on the potential $f$ and the metric $\mathscr{G}$ for the constrained sampling problem. The central problem parameters as characterised by the sufficient conditions and their dependence on the $\delta$-mixing time of MAPLA are summarised in Table 2.

---

1. $\nabla \cdot M$ for a matrix function $M$ is the vector-valued function defined as $(\nabla \cdot M)(x)_i = (\nabla \cdot M_{:,i})(x)$.

| Conditions | | Mixing time scaling |
|---|---|---|
| $f$ is $\begin{cases} (\mu,\mathscr{G})\text{-curv. lower-bdd.} \\ (\lambda,\mathscr{G})\text{-curv. upper-bdd.} \\ (\beta,\mathscr{G})\text{-grad. upper-bdd.} \end{cases}$ & $\mathscr{G}$ is $\begin{cases} \text{self-concordant} \\ \nu\text{-symmetric} \end{cases}$ | $\min\left\{\nu,\frac{1}{\mu}\right\}\cdot\max\{d^3,d\lambda,\beta^2\}$ (Theorem 9) |
| | $\mathscr{G}$ is $\begin{cases} \text{self-concordant}_{++} \\ \nu\text{-symmetric} \end{cases}$ | $\min\left\{\nu,\frac{1}{\mu}\right\}\cdot\max\{d\beta,d\lambda,\beta^2\}$ (Theorem 10) |
| $f$ is linear $\quad$ & $\mathscr{G}$ is $\begin{cases} \text{self-concordant}_{++} \\ \nu\text{-symmetric} \end{cases}$ | | $\nu\cdot d^2$ (Theorem 11) |

Table 2: Summary of dependence of problem parameters on $\delta$-mixing times of MAPLA from a warm start. All scalings above hide a polynomial dependence on $\log(1/\delta)$, and assume $\mu,\beta \geq 1$.

The precise definitions of the conditions are discussed in Sections 4.1.1 and 4.1.2. To contextualise these mixing time scalings, we provide a brief comparison of these mixing time scalings to those derived for other constrained sampling algorithms. We begin by remarking that ManifoldMALA has no known mixing time guarantees for either the unconstrained or the constrained sampling problem. For DikinWalk, Kook and Vempala (2024) show that the mixing time of DikinWalk scales as $\min\{\nu,1/\mu\}\cdot\max\{d,d\lambda\}$ under the same conditions on $f$ and $\mathscr{G}$ as stated in the second row of Table 2, modulo the $(\beta,\mathscr{G})$-gradient upper-bounded condition. The "self-concordant$_{++}$" condition in Table 2 is a collection of stronger notions of self-concordance that were identified by Kook and Vempala (2024). These conditions are sufficient to establish mixing time guarantees for DikinWalk beyond uniform sampling from polytopes, and interestingly also suffice to provide mixing time guarantees for MAPLA. These stronger notions of self-concordance yield a strictly better dependence on the dimension $d$ on the mixing time than when $\mathscr{G}$ is considered to be just self-concordant (first row vs. second row). As a special case, consider the choice where $\mathscr{G} = \nabla^2\phi$ for a *Legendre type function* $\phi$, and assume that potential $f$ and $\nabla^2\phi$ satisfy the conditions stated in the first row of Table 2. In this setting, the mixing time scaling of MAMLA with $\phi$ as the mirror function scales as $\min\{\nu,1/\mu\}\cdot\max\{d^3,d\lambda,\beta^2\}$, which matches the scaling derived for MAPLA. However, when $\phi$ is such that $\nabla^2\phi$ satisfies "self-concordant$_{++}$", the analysis of MAMLA fails to leverage the stronger self-concordant properties due to the *dual* nature of the method as highlighted previously. Lastly, we also derive mixing time guarantees for constrained sampling with exponential densities like those which arise from lifting. Specifically, the potential here is defined as $f(x) = \sigma^\top x$ for non-zero $\sigma$, and the guarantees we derive are notably independent of $\sigma$. This independence of $\sigma$ is also a feature of the mixing time guarantee for the Riemannian Hamiltonian Monte Carlo (RHMC) algorithm (Girolami and Calderhead, 2011), which is another MCMC method that also uses a geometric properties of the domain in the form of a metric $\mathscr{G}$, and can be applied to the constrained sampling problem. Kook et al. (2023) specifically show that the mixing time of discretised RHMC for sampling from $\Pi$ with a linear potential whose support $\mathcal{K}$ is a polytope formed by $m$ constraints scales as $m\cdot d^3$ when $\mathscr{G}$ is given by the Hessian of the log-barrier function of this polytope. For this specific constrained sampling problem and choice of $\mathscr{G}$, the mixing time of MAPLA scales as $m\cdot d^2$ as this metric is both self-concordant$_{++}$ and $\nu$-symmetric with $\nu = m$.

**Organisation**  The remainder of this paper is organised as follows. The next section (Section 2) is dedicated to a review of related work. We formally introduce MAPLA in Section 3, and state its mixing time guarantees in Section 4. We conclude with a summary and some open questions in Section 5. Additional details excluded from the main text due to the space constraints such as the proof of the theorems in Section 4 are given in the Appendix.

## 2. Background and related work

Many of the proposed MCMC methods for sampling can be roughly separated into two classes: zeroth-order and first-order methods. Zeroth-order methods are based on querying the potential $f$ or equivalently, the density up to normalisation constants, while first-order methods are based on querying the gradient of the potential $\nabla f$ in addition to $f$. For constrained sampling, DikinWalk (Narayanan and Rakhlin, 2017; Kook and Vempala, 2024) is a practical and well-studied zeroth order method, which was proposed originally for the task of uniform sampling from $\mathcal{K}$ when $\mathcal{K}$ is a polytope (Kannan and Narayanan, 2009), and with applications in approximate volume computation. This is a notable innovation over earlier methods for uniform sampling $\mathcal{K}$ applied to polytopes such as Hit-And-Run (Smith, 1984; Bélisle et al., 1993; Lovász, 1999) and BallWalk (Lovász and Simonovits, 1993; Kannan et al., 1997) due to its independence on the conditioning of $\mathcal{K}$. MetropolisRandomWalk (Mengersen and Tweedie, 1996; Dwivedi et al., 2018) is also another zeroth-order method related to BallWalk that is suited for unconstrained sampling.

Several first-order algorithms for sampling are based on discretisations of the Langevin dynamics and its variants. For unconstrained sampling, the unadjusted Langevin algorithm (ULA) is a simple yet popular first-order method which is derived as the forward Euler-Maruyama discretisation of the continuous time Langevin dynamics (LD). The Langevin dynamics is equivalent to the gradient flow of the KL divergence $\mu \mapsto \mathrm{d}_{\mathrm{KL}}(\mu\|\Pi)$ in the space of probability measures equipped with the Wasserstein metric $(\mathcal{P}_2(\mathbb{R}^d), W_2)$ (Jordan et al., 1998; Wibisono, 2018).

$$\mathrm{d}X_t = -\nabla f(X_t)\mathrm{d}t + \sqrt{2}\,\mathrm{d}B_t \;\; \Leftrightarrow \;\; \partial_t\rho_t = \nabla \cdot \left(\rho_t \nabla \log \frac{\rho_t}{\pi}\right), \; \rho_t = \mathrm{Law}(X_t)\,. \quad \text{(LD)}$$

$$X_{k+1} - X_k = -h \cdot \nabla f(X_k) + \sqrt{2h} \cdot \xi_k; \quad \xi_k \sim \mathcal{N}(\mathbf{0}, \mathrm{I}_{d\times d})\,. \quad \text{(ULA)}$$

When the Brownian motion $\mathrm{d}B_t$ is excluded in LD, we obtain an ODE that is the gradient flow of $f$. Analogously, without the Gaussian vector $\xi_k$ in ULA, the iteration resembles the gradient descent algorithm with step size $h$. ULA has been shown to be *biased* i.e., the limiting distribution of $X_k$ as $k \to \infty$ does not coincide with $\Pi$ (Durmus et al., 2017; Dalalyan, 2017; Dalalyan and Karagulyan, 2019). Applying the Metropolis-Hastings filter to the proposal distribution defined by a single step of ULA leads to the Metropolis-adjusted Langevin algorithm (MALA) (Roberts and Tweedie, 1996) which is *unbiased*. Much like ULA, MALA has seen a variety of non-asymptotic analyses in recent years (for e.g., Dwivedi et al. (2018); Chewi et al. (2021); Durmus and Moulines (2022); Wu et al. (2022)). Other first-order methods related to the Langevin dynamics (LD) include the underdamped Langevin MCMC (Cheng and Bartlett, 2018; Eberle et al., 2019) and Hamiltonian Monte Carlo (HMC) (Neal, 2011; Durmus et al., 2017; Bou-Rabee et al., 2020).

For optimising $f$, the gradient flow of $f$ can be modified to incorporate geometric information about the state space through the metric $\mathscr{G}$, and this results in the natural gradient flow (Amari, 1998), where the gradient $\nabla f$ is replaced by the *natural gradient*: this is defined as $\mathscr{G}^{-1}\nabla f$, and corresponds to the direction of steepest ascent under the metric $\mathscr{G}$. Similarly for sampling, the

Langevin dynamics (LD) can also be modified to incorporate geometric information, and this leads to the weighted Langevin dynamics (WLD) (Girolami and Calderhead, 2011). The SDE that defines WLD differs from LD in both the drift and the diffusion matrix: the drift consists of the natural gradient of $f$ and an additional correction term $(\nabla \cdot \mathscr{G}^{-1})$ that accounts for change in the diffusion matrix in this SDE. The correction term ensures that the stationary distribution of the SDE is $\Pi$, a fact that is apparent by considering the equivalent PDE of the weighted Langevin dynamics. The forward Euler-Maruyama discretisation of WLD leads to the weighted Langevin algorithm (WLA).

$$\mathrm{d}X_t = \mathfrak{m}(X_t)\mathrm{d}t + \sqrt{2} \cdot \mathscr{G}(X_t)^{-1/2}\,\mathrm{d}B_t$$
$$\mathfrak{m}(X_t) = (\nabla \cdot \mathscr{G}^{-1})(X_t) - \mathscr{G}(X_t)^{-1}\nabla f(X_t) \quad \Leftrightarrow \partial_t \rho_t = \nabla \cdot \left( \rho_t \mathscr{G}^{-1} \nabla \log \frac{\rho_t}{\pi} \right),\ \rho_t = \mathrm{Law}(X_t)\,.$$

$$\text{(WLD)}$$

$$X_{k+1} - X_k = h \cdot \mathfrak{m}(X_k) + \sqrt{2h} \cdot \mathscr{G}(X_k)^{-1/2}\,\xi_k; \quad \xi_k \sim \mathcal{N}\left(\mathbf{0},\ \mathrm{I}_{d\times d}\right)\,. \qquad \text{(WLA)}$$

If $\mathscr{G}$ is *position-independent*, the correction term $\nabla \cdot \mathscr{G}^{-1}$ is identically $\mathbf{0}$. Well-chosen position-independent metrics have proven to be useful by yielding faster mixing of the Markov chain (either of WLA or its Metropolis-adjusted variant) (Cotter et al., 2013; Titsias and Papaspiliopoulos, 2018; Titsias, 2024) when the distribution $\Pi$ is highly anisotropic. On the other hand, *position-dependent* metrics enable leveraging the geometry of the state space better. The PDE form of WLD also lends to the interpretation of WLD as the gradient flow of $\mu \mapsto \mathrm{d}_{\mathrm{KL}}(\mu \| \Pi)$ with respect to the Wasserstein metric on the Riemannian manifold defined by the state space and the metric $\mathscr{G}$. Girolami and Calderhead (2011) propose a Metropolis adjustment to WLA which they call ManifoldMALA, and observe that setting $\mathscr{G}$ to be the Fisher information matrix in ManifoldMALA results in quicker mixing compared to MALA on a collection of unconstrained sampling problems. A simpler but inaccurate discretisation of WLD defined by merely omitting the $\nabla \cdot \mathscr{G}^{-1}$ term in WLA is the *Preconditioned Langevin Algorithm* (PLA):

$$X_{k+1} - X_k = -h \cdot \mathscr{G}(X_k)^{-1}\nabla f(X_k) + \sqrt{2h} \cdot \mathscr{G}(X_k)^{-1/2}\,\xi_k\,; \quad \xi_k \sim \mathcal{N}\left(\mathbf{0},\ \mathrm{I}_{d\times d}\right)\,. \quad \text{(PLA)}$$

It is important to note that even as $h \to 0$, the stationary distribution of PLA is not[2] $\Pi$, and is also likely to carry a higher bias than WLA on account of being an unfaithful discretisation of WLD. Despite this, a single step of PLA serves as a useful proposal distribution for a Metropolis-adjusted scheme as we demonstrate later. Without $\xi_k$ in PLA, this coincides with the natural gradient step with step size $h$, and when $\mathscr{G} = \nabla^2 f$, this resembles the Newton method.

When dealing with constrained distributions, special care has to be taken when working with discretisations of LD or its weighted variant WLD. This is because iterates generated by ULA / WLA / PLA could escape $\mathcal{K}$ since the Gaussian random vector is unconstrained. The projected Langevin algorithm (Bubeck et al., 2018) modifies ULA by including an explicit projection onto $\mathcal{K}$, and is motivated by the projected gradient descent algorithm for optimising functions over a constrained feasibility set. Another class of approaches for constrained sampling are mirror Langevin algorithm (which were briefly introduced in section 1), and are based on discretisations of the mirror Langevin dynamics (MLD) (Zhang et al., 2020; Chewi et al., 2020). The mirror Langevin dynamics are defined as follows; the mirror function $\phi$ here is a Legendre type function.

$$Y_t = \nabla\phi(X_t); \quad \mathrm{d}Y_t = -\nabla f(X_t)\mathrm{d}t + \sqrt{2} \cdot \nabla^2\phi(X_t)^{1/2}\,\mathrm{d}B_t\,. \qquad \text{(MLD}_1\text{)}$$

$$\mathrm{d}X_t = ((\nabla \cdot M)(X_t) - M(X_t)\nabla f(X_t))\mathrm{d}t + \sqrt{2} \cdot M(X_t)^{1/2}\mathrm{d}B_t;\ M = \nabla^2\phi^{-1}\,. \quad \text{(MLD}_2\text{)}$$

---

2. unless $\mathscr{G}$ is position-independent.

The assumption that $\phi$ is a Legendre type in conjunction with Itô's lemma establishes the equivalence of $\mathsf{MLD}_1$ and $\mathsf{MLD}_2$ (Zhang et al., 2020; Jiang, 2021). Through $\mathsf{MLD}_2$, we see that the mirror Langevin dynamics is a special case of $\mathsf{WLD}$ with $\mathscr{G} = \nabla^2\phi$. The two-step definition ($\mathsf{MLD}_1$) is particularly amenable to discretisation as this automatically ensures feasibility of iterates when $\mathcal{K}$ is compact[3]. The Euler-Maruyama discretisation of $\mathsf{MLD}_1$ is the mirror Langevin algorithm ($\mathsf{MLA}$) (Zhang et al., 2020; Li et al., 2022). Prior works (Ahn and Chewi, 2021; Jiang, 2021) propose other novel discretisations of $\mathsf{MLD}_1$ which are not practically feasible as they involve simulating a SDE. These discretisations including $\mathsf{MLA}$ are *biased*; for fixed $h > 0$, the law of the iterate $X_k$ does not converge to $\Pi$, and this is also a feature of $\mathsf{ULA}$ for unconstrained sampling, and projected $\mathsf{ULA}$ for constrained sampling. Srinivasan et al. (2024) propose augmenting a Metropolis-Hastings filter to the Markov chain induced by a single step of $\mathsf{MLA}$ ($\mathsf{MAMLA}$, discussed previously in section 1), which eliminates the bias that $\mathsf{MLA}$ carries. Implicitly, both $\mathsf{MLA}$ and $\mathsf{MAMLA}$ make two key assumptions: that $\mathcal{K}$ is compact, and that $\nabla\phi^\star$ can be computed exactly. The compactness of $\mathcal{K}$ ensures that $\mathcal{K}^\star = \mathbb{R}^d$, which is necessary to generate a proposal. When $\mathcal{K}$ is not compact, $\mathcal{K}^\star$ is not necessarily $\mathbb{R}^d$, and an explicit projection onto $\mathcal{K}^\star$ is required for the proposal to be well-defined. For $\mathsf{MAMLA}$, it suffices to induce an explicit rejection of dual proposals that don't belong in $\mathcal{K}^\star$, but this requires a membership oracle for $\mathcal{K}^\star$ instead, which is non-trivial to obtain. Additionally, in general, closed form expressions for $\nabla\phi^\star$ cannot be derived and in these cases, $\nabla\phi^\star$ can only be computed approximately by solving the convex problem which defines the convex conjugate to within a certain tolerance. $\mathsf{MAPLA}$ addresses both of these difficulties by operating in the primal space $\mathcal{K}$, which also the motivation behind the development of this method. We also review other methods for constrained sampling in Appendix B.

## 3. The Metropolis-adjusted Preconditioned Langevin Algorithm

Here, we formally introduce the Metropolis-adjusted Preconditioned Langevin Algorithm ($\mathsf{MAPLA}$). The pseudocode presentation of $\mathsf{MAPLA}$ is given in Section C.2.

**Notation** The set of $d \times d$ symmetric positive definite matrices is denoted by $\mathbb{S}_+^d$. For $A \in \mathbb{S}_+^d$, and $x, y \in \mathbb{R}^d$, we use $\langle x, y \rangle_A$ to denote $\langle x, Ay \rangle$ and $\|x\|_A = \sqrt{\langle x, x \rangle_A}$. When the subscript $A$ is omitted as in $\langle x, y \rangle$, then this corresponds to the $A = \mathrm{I}_{d\times d}$. The $\ell_p$ norm of $x$ is denoted by $\|x\|_p$. For a distribution $\nu$, $d\nu(x)$ denotes its density at $x$ (if it exists) unless otherwise specified.

A (time-homogeneous) Markov chain over domain $\mathcal{K}$ is defined as a collection of transition probability measures $\mathbf{P} = \{\mathcal{P}_x : x \in \mathcal{K}\}$, and we refer to $\mathcal{P}_x$ as the *one-step* distribution associated with $x$. Assume that for every $x \in \mathcal{K}$, the proposal distribution $\mathcal{P}_x$ has density function $p_x$. The transition operator $\mathbb{T}_{\mathbf{P}}$ on the space of probability measures is defined as

$$(\mathbb{T}_{\mathbf{P}}\mu)(S) = \int_{\mathcal{K}} \mathcal{P}_y(S) \cdot \mathrm{d}\mu(y) \qquad \text{and} \quad (\mathbb{T}_{\mathbf{P}}^k\mu)(S) = \left(\mathbb{T}_{\mathbf{P}}(\mathbb{T}_{\mathbf{P}}^{k-1}\mu)\right)(S) .$$

A probability measure $\nu$ is a *stationary* measure of a Markov chain $\mathbf{P}$ if $\mathbb{T}_{\mathbf{P}}\nu = \nu$. If $\mathbf{P}$ is *ergodic*, then $\nu$ is the unique stationary measure. A related but stronger notion is *reversibility*; a Markov chain $\mathbf{P}$ is reversible with respect to $\nu$ if for any measurable subsets of $A$, $B$ of $\mathcal{K}$,

$$\int_A \mathcal{P}_x(B) \cdot \mathrm{d}\nu(x) = \int_B \mathcal{P}_y(A) \cdot \mathrm{d}\nu(y) .$$

---

3. This is due to the fact that the range of $\nabla\phi^\star$ is $\mathbb{R}^d$ when $\mathcal{K}$ is compact (Rockafellar, 1970, Corr. 13.3.1).

If $\mathbf{P}$ is ergodic and reversible with respect to $\nu$, then $\nu$ is the stationary measure of $\mathbf{P}$.

The Metropolis adjustment is one technique to produce a new Markov chain that is reversible with respect to a certain distribution. Formally, suppose we intend to adjust (i.e., make reversible) $\mathbf{P}$ with respect to a distribution $\Pi$ whose density is $\pi$. The Metropolis adjustment works by including an explicit accept-reject step called the Metropolis-Hastings (MH) filter. The implementation of the MH filter involves computing an *acceptance ratio* between feasible points $X$ and $Z$ defined as

$$p_{\text{accept}}(Z; X) \stackrel{\text{def}}{=} \min\left\{1, \frac{\pi(Z) \cdot p_Z(X)}{\pi(X) \cdot p_X(Z)}\right\} = \min\left\{1, \frac{e^{-f(Z)}}{e^{-f(X)}} \cdot \frac{p_Z(X)}{p_X(Z)}\right\} . \tag{1}$$

Notably, this only requires the unnormalised target density function $e^{-f}$ unlike rejection sampling. The Markov chain formed by the Metropolis adjustment is also known to be optimal in a certain sense (Billera and Diaconis, 2001), and is generally preferred to other similar techniques like the Barker correction which uses $\frac{A}{1+A}$ in lieu of $\min\{1, A\}$ in Equation (1).

Consider the Markov chain $\mathbf{P}$ defined by a single step of PLA. From any $x \in \text{int}(\mathcal{K})$[4], PLA returns

$$x' = x - h \cdot \mathscr{G}(x)^{-1} \nabla f(x) + \sqrt{2h} \cdot \mathscr{G}(x)^{-1/2} \, \xi$$

where $\xi$ is independently drawn from $\mathcal{N}\left(\mathbf{0},\ \mathbf{I}_{d \times d}\right)$. Hence, for any $x \in \text{int}(\mathcal{K})$, $\mathcal{P}_x$ is defined as

$$\mathcal{P}_x = \mathcal{N}\left(x - h \cdot \mathscr{G}(x)^{-1} \nabla f(x),\ 2h \cdot \mathscr{G}(x)^{-1}\right) .$$

As introduced in Section 1, MAPLA results from performing a Metropolis-adjustment of the Markov chain $\mathbf{P}$ with respect to $\Pi$. We use $\mathbf{T} = \{\mathcal{T}_x : x \in \mathcal{K}\}$ to denote this Metropolis-adjusted version of $\mathbf{P}$, where $\mathcal{T}_x$ is the one-step distribution from $x \in \mathcal{K}$ per MAPLA. A key aspect of MAPLA is choosing the metric $\mathscr{G}$ for a given domain. To this end, we include a discussion about how ideas from interior point methods in optimisation (Nesterov and Nemirovski, 1994) can be useful in picking $\mathscr{G}$ for certain domains in Section C.1.

## 4. Mixing time guarantees for MAPLA

In this section, we state our main theorems that provide upper bounds on the mixing time for MAPLA under certain sufficient conditions as previously alluded to in Table 2. We begin with certain preliminaries that include the definitions of the conditions that we assume on the metric $\mathscr{G}$ and potential $f$ to provide mixing time guarantees.

### 4.1. Preliminaries

**Additional Notation** Let $B$ be a $d \times d$ matrix. The operator and Frobenius norms of $B$ are denoted by $\|B\|_{\text{op}}$ and $\|B\|_{\text{F}}$ respectively. For a smooth map $g$ and $x$ in its domain, the directional derivative of $g$ at $x$ in direction $v$ is denoted by $\mathrm{D}g(x)[v]$. The (second-order) directional derivative of $g$ at $x$ in directions $v, w$ is denoted by $\mathrm{D}^2 g(x)[v, w]$, which is equal to both $\mathrm{D}(\mathrm{D}g(x)[v])[w]$ and $\mathrm{D}(\mathrm{D}g(x)[w])[v]$ as $g$ is smooth. Given a set $\mathcal{A}$, the set of all of its measurable subsets is denoted by $\mathcal{F}(\mathcal{A})$, and the interior and boundary of $\mathcal{A}$ are denoted by $\text{int}(\mathcal{A})$ and $\partial\mathcal{A}$ respectively.

---

4. Since $\mathcal{K}$ is convex, $\partial\mathcal{K}$ is a Lebesgue null set, and hence points on the boundary can be safely disregarded.

### 4.1.1. CLASSES OF METRICS

In the rest of this paper, we assume the following regularity conditions about the metric: $\mathscr{G}$ is only defined on $\text{int}(\mathcal{K})$ and becomes unbounded as it approaches the boundary i.e., $\|\mathscr{G}(x_k)\|_{\text{op}} \to \infty$ for any sequence $\{x_k\} \to \partial\mathcal{K}$, and is twice differentiable. The first two conditions ensures that the solution to the continuous time dynamics (WLD) stays within $\mathcal{K}$.

**Self-concordance**   This classical property is key in the analysis of interior points methods for constrained optimisation (Nesterov and Nemirovski, 1994), and quantifies the rate of change of a matrix-valued function in a certain sense as defined below.

**Definition 1**   *The metric* $\mathscr{G} : \text{int}(\mathcal{K}) \to \mathbb{S}_+^d$ *is* self-concordant *if for all* $x \in \text{int}(\mathcal{K})$ *and* $v \in \mathbb{R}^d$

$$|\text{D}\mathscr{G}(x)[v,v,v]| \leq 2 \cdot \|v\|_{\mathscr{G}(x)}^3 \ .$$

The design of DikinWalk and MAPLA implicitly assume the invertibility of the metric $\mathscr{G}$. Notably, when $\mathscr{G}$ is self-concordant and its domain $\text{int}(\mathcal{K})$ contains no straight lines, then $\mathscr{G}$ is always invertible (see section E.2). The definition above is equivalent to (Nesterov, 2018, Corr. 5.1.1).

$$\forall \, x \in \text{int}(\mathcal{K}), \ v \in \mathbb{R}^d, \quad \|\mathscr{G}(x)^{-1/2}\text{D}\mathscr{G}(x)[v]\mathscr{G}(x)^{-1/2}\|_{\text{op}} \leq 2 \cdot \|v\|_{\mathscr{G}(x)} \ .$$

**Strong self-concordance**   This property (Laddha et al., 2020) and replaces the operator norm in the equivalent characterisation of self-concordance by the Frobenius norm. This is a stronger notion than self-concordance due to the fact that $\|A\|_{\text{op}} \leq \|A\|_{\text{F}}$ for any matrix $A$.

**Definition 2**   *The metric* $\mathscr{G} : \text{int}(\mathcal{K}) \to \mathbb{S}_+^d$ *is* strongly self-concordant *if for all* $x \in \text{int}(\mathcal{K})$ *and* $v \in \mathbb{R}^d$,
$$\|\mathscr{G}(x)^{-1/2}\text{D}\mathscr{G}(x)[v]\mathscr{G}(x)^{-1/2}\|_{\text{F}} \leq 2 \cdot \|v\|_{\mathscr{G}(x)} \ .$$

**Symmetry**   The role of the symmetrised set $\mathcal{K} \cap 2x - \mathcal{K}$ for $x \in \text{int}(\mathcal{K})$ was originally observed by Gustafson and Narayanan (2018) in their study of the John walk, which was separately isolated by Laddha et al. (2020) as property of metrics. This property yields an isoperimetric inequality that results in mixing time bounds for several constrained sampling algorithms discussed previously.

For any $x \in \text{int}(\mathcal{K})$, and $r > 0$, the Dikin ellipsoid of radius $r$ (denoted by $\mathcal{E}_x^{\mathscr{G}}(r)$) is defined as

$$\mathcal{E}_x^{\mathscr{G}}(r) = \{y : \|y - x\|_{\mathscr{G}} < r\} \ . \tag{2}$$

**Definition 3**   *The metric* $\mathscr{G} : \text{int}(\mathcal{K}) \to \mathbb{S}_+^d$ *is said to be* $\nu$-symmetric ($\nu \geq$) *if for any* $x \in \text{int}(\mathcal{K})$,

$$\mathcal{E}_x^{\mathscr{G}}(1) \subseteq \mathcal{K} \cap (2x - \mathcal{K}) \subseteq \mathcal{E}_x^{\mathscr{G}}(\sqrt{\nu}) \ .$$

**Lower trace and average self-concordance**   The properties were recently proposed in Kook and Vempala (2024) and are abstractions from prior analyses of DikinWalk (Sachdeva and Vishnoi, 2016; Narayanan and Rakhlin, 2017). Specifically, these analyses were catered to the setting where $\mathcal{K}$ is a polytope and $\mathscr{G} = \nabla^2\phi$ for $\phi$ being the log-barrier function of the polytope. The salient features of these analyses that were abstracted by Kook and Vempala (2024) for general metrics

are: (1) a lower bound on the curvature of the function $x \mapsto \log \det \mathscr{G}(x)$, and (2) an upper bound on the likelihood of $\|x - z\|_{\mathscr{G}(x)}^2 - \|x - z\|_{\mathscr{G}(z)}^2$ being large for a Dikin proposal $z$ from $x$. Lower trace self-concordance (along with strong self-concordance) yields the first property, and average self-concordance yields the second property, which are defined below.

**Definition 4** *The metric* $\mathscr{G} : \mathrm{int}(\mathcal{K}) \to \mathbb{S}_+^d$ *is said to be* lower trace self-concordant *with parameter* $\alpha \geq 0$ *if for all* $x \in \mathrm{int}(\mathcal{K})$ *and* $v \in \mathbb{R}^d$,

$$\mathrm{trace}(\mathscr{G}(x)^{-1}\mathrm{D}^2\mathscr{G}(x)[v, v]) \geq -\alpha \cdot \|v\|_{\mathscr{G}(x)}^2 .$$

**Definition 5** *The metric* $\mathscr{G} : \mathrm{int}(\mathcal{K}) \to \mathbb{S}_+^d$ *is said to be* average self-concordant *if for any* $x \in \mathrm{int}(\mathcal{K})$ *and* $\varepsilon > 0$, *there exists* $r_\varepsilon > 0$ *such that for any* $h \in (0, \frac{r_\varepsilon^2}{2d}]$,

$$\mathbb{P}_{\xi \sim \mathcal{N}(x, 2h \cdot \mathscr{G}(x)^{-1})} \left( \|\xi - x\|_{\mathscr{G}(\xi)}^2 - \|\xi - x\|_{\mathscr{G}(x)}^2 \leq 4h \cdot \varepsilon \right) \geq 1 - \varepsilon$$

### 4.1.2. FUNCTION CLASSES

**Curvature lower and upper-boundedness** This is a generalisation of the standard second-order definitions of smoothness and convexity where $\mathscr{G} = \mathrm{I}_{d \times d}$. These are related to relative convexity and smoothness (Bauschke et al., 2017; Lu et al., 2018). Specifically, if $f$ is $\lambda$-relative smooth (or $\mu$-relative convex) with respect to $\psi$, then $f$ satisfies a $(\lambda, \nabla^2\psi)$-curvature upper bound (or $(\mu, \nabla^2\psi)$-curvature lower bound) respectively.

**Definition 6** *Given a metric* $\mathscr{G} : \mathrm{int}(\mathcal{K}) \to \mathbb{S}_+^d$, *the potential* $f : \mathrm{int}(\mathcal{K}) \to \mathbb{R}$ *satisfies a* $(\mu, \mathscr{G})$-curvature lower bound *where* $\mu \geq 0$ *if for any* $x \in \mathrm{int}(\mathcal{K})$,

$$\nabla^2 f(x) \succeq \mu \cdot \mathscr{G}(x) .$$

**Definition 7** *Given a metric* $\mathscr{G} : \mathrm{int}(\mathcal{K}) \to \mathbb{S}_+^d$, *the potential* $f : \mathrm{int}(\mathcal{K}) \to \mathbb{R}$ *satisfies a* $(\lambda, \mathscr{G})$-curvature upper bound *where* $\lambda \geq 0$ *if for any* $x \in \mathrm{int}(\mathcal{K})$,

$$\nabla^2 f(x) \preceq \lambda \cdot \mathscr{G}(x) .$$

**Gradient upper bound** This class of functions can be viewed as an extension of the standard notion of Lipschitz continuity for differentiable functions to take into account the metric. This property has been useful in the analysis of algorithms based on the mirror Langevin dynamics (Ahn and Chewi, 2021; Srinivasan et al., 2024) which is equivalent to setting $\mathscr{G} = \nabla^2\phi$.

**Definition 8** *Given a metric* $\mathscr{G} : \mathrm{int}(\mathcal{K}) \to \mathbb{S}_+^d$, *the potential* $f : \mathrm{int}(\mathcal{K}) \to \mathbb{R}$ *satisfies a* $(\beta, \mathscr{G})$-gradient upper bound *where* $\beta \geq 0$ *if for any* $x \in \mathrm{int}(\mathcal{K})$,

$$\|\nabla f(x)\|_{\mathscr{G}(x)^{-1}} \leq \beta .$$

### 4.1.3. CONDUCTANCE AND MIXING TIME

Let $s \in (0, 1/2)$. The $s$-conductance of a Markov chain $\mathbf{P} = \{\mathcal{P}_x : x \in \mathcal{K}\}$ with stationary distribution $\nu$ supported on $\mathcal{K}$ is defined as

$$\Phi^s_{\mathbf{P}} = \inf_{\substack{A \in \mathcal{F}(\mathcal{K}) \\ \nu(A) \in (s, 1-s)}} \frac{1}{\min\{\nu(A) - s,\ 1 - \nu(A) - s\}} \cdot \int_A \mathcal{P}_x(\mathcal{K} \setminus A) \cdot \mathrm{d}\nu(x) \ .$$

The (ordinary) conductance $\Phi_{\mathbf{P}}$ of $\mathbf{P}$ is the limit of $\Phi^s_{\mathbf{P}}$ as $s \to 0$. To obtain mixing time guarantees for MAPLA, we use a strategy which assumes that the initial distribution $\Pi_0$ is *warm* relative to the stationary measure $\Pi$. Two notions of warmness that are relevant to this technique are listed below.

- a distribution $\mu_0$ is $(L_\infty, M)$-warm w.r.t. $\nu$ if $\sup\limits_{A \in \mathcal{F}(\mathcal{K})} \frac{\mu_0(A)}{\nu(A)} \le M$.

  The set of all such $\mu_0$ is denoted by $\mathsf{Warm}(L_\infty, M, \nu)$.

- a distribution $\mu_0$ is $(L_1, M)$-warm w.r.t. $\nu$ if $\left\| \frac{\mu_0}{\nu} \right\|_{L^1(\mu_0)} = \mathbb{E}_{\mu_0} \left[ \frac{d\mu_0(x)}{d\nu(x)} \right] = M$.

  The set of all such $\mu_0$ is denoted by $\mathsf{Warm}(L_1, M, \nu)$.

For $\delta \in (0, 1)$, the $\delta$-mixing time in TV distance of a Markov chain $\mathbf{P}$ with stationary distribution $\nu$ starting from a distribution $\mu_0 \in \mathcal{C}$ is defined as

$$\tau_{\mathrm{mix}}(\delta; \mathbf{P}, \mathcal{C}) \overset{\mathrm{def}}{=} \sup_{\pi_0 \in \mathcal{C}} \inf\{k \ge 0 : \mathrm{d}_{\mathrm{TV}}(\mathbb{T}^k_{\mathbf{P}} \pi_0, \nu) \le \delta\} \ .$$

## 4.2. Main results

Now, we state our main mixing time guarantees for MAPLA. For clarity, we state the assumptions made on the potential $f$ of $\Pi$ separately below.

**A$_1$** $f$ satisfies $(\mu, \mathscr{G})$-curvature lower bound (Definition 6) and let $\widetilde{\mu} = \frac{\mu}{8 + 4\sqrt{\mu}}$.

**A$_2$** $f$ satisfies $(\lambda, \mathscr{G})$-curvature upper bound (Definition 7).

**A$_3$** $f$ is $\beta$-relatively Lipschitz continuous w.r.t. $\mathscr{G}$ (Definition 8).

**Warmup: working with self-concordance of $\mathscr{G}$** Here, we assume that the metric $\mathscr{G}$ satisfies self-concordance (Definition 1), which is the most basic notion of self-concordance defined previously.

**Theorem 9** *Consider a distribution $\Pi$ supported over $\mathcal{K}$ that is a closed, convex subset of $\mathbb{R}^d$ whose density is $\pi(x) \propto e^{-f(x)}$. Let the metric $\mathscr{G} : \mathrm{int}(\mathcal{K}) \to \mathbb{S}^d_+$ be self-concordant and $\nu$-symmetric, and assume that the potential $f : \mathrm{int}(\mathcal{K}) \to \mathbb{R}$ satisfies* **A$_1$**, **A$_2$** *and* **A$_3$**. *Define the quantity $b_{SC}(d, \lambda, \beta)$*

$$b_{SC}(d, \lambda, \beta) \overset{\mathrm{def}}{=} c_1 \cdot \min \left\{ \frac{1}{d^3}, \frac{1}{d \cdot \lambda}, \frac{1}{\beta^2}, \frac{1}{\beta^{2/3}}, \frac{1}{(\beta \cdot \lambda)^{2/3}} \right\} \ .$$

*For precision $\delta \in (0, 1/2)$ and warmness parameter $M \ge 1$, if the step size $h$ is bounded as $0 < h \le b_{SC}(d, \lambda, \beta)$,* **MAPLA** *satisfies for $\mathcal{C} \in \{\mathsf{Warm}(L_\infty, M, \Pi), \mathsf{Warm}(L_1, M, \Pi)\}$ that*

$$\tau_{\mathrm{mix}}(\delta; \mathbf{T}, \mathcal{C}) = \frac{c_2}{h} \cdot \max \left\{ 1, \min \left\{ \frac{1}{\widetilde{\mu}^2}, \nu \right\} \right\} \cdot \log \left( \frac{\mathfrak{M}_\mathcal{C}}{\delta} \right) \ , \quad \mathfrak{M}_\mathcal{C} = \begin{cases} M^{1/2}, & \mathcal{C} = \mathsf{Warm}(L_\infty, M, \Pi) \\ M^{1/3}, & \mathcal{C} = \mathsf{Warm}(L_1, M, \Pi) \end{cases}$$

*where $c_1, c_2$ are universal positive constants.*

**Beyond standard self-concordance of $\mathscr{G}$** Now, we assume that the metric $\mathscr{G}$ satisfies *self-concordance$_{++}$*, which is a combination of the stronger notions of self-concordance (Definitions 2, 4 and 5) as described previously. Self-concordance$_{++}$ enables a larger bound on the step size $h$ than $b_1$ in Theorem 9, which consequently yields better mixing time guarantees.

**Theorem 10** *Consider a distribution $\Pi$ supported over $\mathcal{K}$ that is a closed, convex subset of $\mathbb{R}^d$ whose density is $\pi(x) \propto e^{-f(x)}$. Let the metric $\mathscr{G} : \mathrm{int}(\mathcal{K}) \to \mathbb{S}_+^d$ be strongly, $\alpha$-lower trace, and average self-concordant and $\nu$-symmetric, and assume that the potential $f : \mathrm{int}(\mathcal{K}) \to \mathbb{R}$ satisfies $\mathsf{A}_1$, $\mathsf{A}_2$ and $\mathsf{A}_3$. Define the quantity $b_{\mathsf{SC}_{++}}(d, \lambda, \alpha, \beta)$*

$$b_{\mathsf{SC}_{++}}(d, \lambda, \beta, \alpha) \stackrel{\mathrm{def}}{=} c_1 \cdot \min \left\{ \frac{1}{d \cdot \beta}, \frac{1}{d \cdot \lambda}, \frac{1}{d \cdot (\alpha + 4)}, \frac{1}{\beta^2}, \frac{1}{(\beta \cdot (\alpha + 4))^{2/3}}, \frac{1}{(\beta \cdot \lambda)^{2/3}} \right\} .$$

*For precision $\delta \in (0, 1/2)$ and warmness parameter $M \geq 1$, if the step size $h$ is bounded as $0 < h \leq b_{\mathsf{SC}_{++}}(d, \lambda, \beta, \alpha)$, MAPLA satisfies for $\mathcal{C} \in \{\mathsf{Warm}(L_\infty, M, \Pi), \mathsf{Warm}(L_1, M, \Pi)\}$ that*

$$\tau_{\mathrm{mix}}(\delta; \mathbf{T}, \mathcal{C}) = \frac{c_2}{h} \cdot \max \left\{ 1, \min \left\{ \frac{1}{\widetilde{\mu}^2}, \nu \right\} \right\} \cdot \log \left( \frac{\mathfrak{M}_\mathcal{C}}{\delta} \right) , \quad \mathfrak{M}_\mathcal{C} = \begin{cases} M^{1/2}, & \mathcal{C} = \mathsf{Warm}(L_\infty, M, \Pi) \\ M^{1/3}, & \mathcal{C} = \mathsf{Warm}(L_1, M, \Pi) \end{cases}$$

*where $c_1, c_2$ are universal positive constants.*

**Handling linear $f$** Here, we discuss the setting where $f(x) = \sigma^\top x \big|_\mathcal{K}$ for $\sigma \neq \mathbf{0}$. Recall that both Theorems 9 and 10 assume that the potential $f$ satisfies $(\beta, \mathscr{G})$-gradient upper bound ($\mathsf{A}_3$), and hence directly invoking these theorems for this setting would result in a dependence on $\beta(\sigma) \stackrel{\mathrm{def}}{=} \sup_{x \in \mathrm{int}(\mathcal{K})} \|\sigma\|_{\mathscr{G}(x)^{-1}}$. This uncovers two issues. First, for any $\mathfrak{c} \in \mathbb{R}$, $\beta(\mathfrak{c} \cdot \sigma) = |\mathfrak{c}| \cdot \beta(\sigma)$, which would imply that the scale of $\sigma$ affects the mixing time guarantee. Second, even when $\sigma$ is normalised (i.e., $\|\sigma\| = 1$) for a self-concordant $\mathscr{G}$, $\beta(\sigma)$ could depend on the size of $\mathcal{K}$. Hence, it is crucial that the mixing time guarantees in this setting is independent of $\beta(\sigma)$. In the following theorem, we derive such a *scale-independent* guarantee for MAPLA which uses properties of densities whose potential $f$ is linear (Kook et al., 2023).

**Theorem 11** *Consider a distribution $\Pi$ supported over $\mathcal{K}$ that is a closed, convex subset of $\mathbb{R}^d$ whose density is $\pi(x) \propto e^{-\sigma^\top x}$. Let the metric $\mathscr{G} : \mathrm{int}(\mathcal{K}) \to \mathbb{S}_+^d$ be strongly and average self-concordant and $\nu$-symmetric, and assume that it also satisfies $\mathrm{D}^2\mathscr{G}(x)[v, v] \succeq \mathbf{0}$ for all $x \in \mathrm{int}(\mathcal{K})$ and $v \in \mathbb{R}^d$. Define the quantity $b_{\mathsf{Exp}}(d, M, \delta)$*

$$b_{\mathsf{Exp}}(d, M, \delta) \stackrel{\mathrm{def}}{=} c_1 \cdot \frac{1}{d^2 \log^2(\frac{M}{\delta})} .$$

*For precision $\delta \in (0, 1/2)$ and warmness parameter $M \geq 1$, if the step size $h$ is bounded as $0 < h \leq b_{\mathsf{Exp}}(d, M, \delta)$, MAPLA satisfies*

$$\tau_{\mathrm{mix}}(\delta; \mathbf{T}, \mathsf{Warm}(L_\infty, M, \Pi)) = \frac{c_2}{h} \cdot \max \{1, \nu\} \cdot \log \left( \frac{M}{\delta} \right)$$

*where $c_1, c_2$ are universal positive constants.*

### 4.2.1. A DISCUSSION OF THE RESULTS

The underlying technique used to prove Theorems 9 to 11 is due to is due to Lovász (1999). Given a Markov chain $\mathbf{T}$, Lovász's technique involves showing that for any two points $x, y$ close enough in a sufficiently large subset of $\mathcal{K}$, the TV distance between the one-step distributions $\mathcal{T}_x, \mathcal{T}_y$ is uniformly bounded away from 1, and is hence referred to as the *one-step overlap* technique. Given this one-step overlap, we rely on isoperimetric inequalities which lead to a lower bound on the $s$-conductance / conductance of $\mathbf{T}$, which results in mixing time guarantees by the classical result of Lovász and Simonovits (1993). The $\nu$-symmetry of the metric $\mathscr{G}$ results in an isoperimetric inequality for log-concave distributions (Laddha et al., 2020), which can be complemented by another isoperimetric inequality when the potential satisfies a $(\mu, \mathscr{G})$-curvature lower bound ($\mathbf{A}_1$) for a self-concordant $\mathscr{G}$. One of our contributions is deriving the latter isoperimetric inequality, which is a generalisation of prior results by Gopi et al. (2023, Lem. 7). Other conditions placed on $\mathscr{G}$ and the potential $f$ namely self-concordance or self-concordance$_{++}$, and the $(\lambda, \mathscr{G})$-curvature upper bound ($\mathbf{A}_2$) and $(\beta, \mathscr{G})$-gradient upper bound ($\mathbf{A}_3$) conditions on $f$ yield bounds on the step size which ensures that the expected acceptance rate is away from 0 as $d$ increases, which is related to the one-step overlap. Our analysis reveals that $\mathbf{A}_3$ is not necessary to derive mixing time guarantees for MAPLA, and also extends to the analysis of MAMLA (Srinivasan et al., 2024) where $\mathbf{A}_3$ is also considered for the potential $f$ following Ahn and Chewi (2021). More precisely, we find that it is sufficient if the function $x \mapsto \|\nabla f(x)\|_{\mathscr{G}(x)^{-1}}$ is uniformly bounded in a sufficiently large convex subset of $\mathcal{K}$. This weaker sufficient condition is satisfied in two cases which our theorems cover: (1) when $f$ satisfies the $(\beta, \mathscr{G})$-gradient upper bound condition (as assumed in Theorems 9 and 10), and (2) when $f$ is linear (Theorem 11). Finally, the convexity-style assumption on the metric $\mathscr{G}$ in Theorem 11 implies that $\mathscr{G}$ is 0-lower trace self-concordance, which is used to guarantee that the large enough subset identified for the weaker sufficient condition described above is convex.

While we use self-concordance$_{++}$ in Theorem 10, the proofs to establish mixing time guarantees for DikinWalk in Kook and Vempala (2024) and MAPLA in this work differ in the details, specifically in how the one-step overlap is established. In the analysis of DikinWalk presented by Kook and Vempala (2024), they work with an exact analytical expression for the TV distance between the one-step distributions induced by an iteration of DikinWalk. More precisely, for a Markov chain $\mathbf{T}$ formed by the Metropolis adjustment of $\mathbf{P}$ w.r.t. $\Pi$ and $p_{\text{accept}}$ as defined in Equation (1), we have

$$\mathrm{d}_{\mathrm{TV}}(\mathcal{T}_x, \mathcal{T}_y) = \frac{1}{2}(r_x + r_y) + \frac{1}{2}\int_{z \in \mathcal{K}} |p_{\text{accept}}(z; x) \cdot p_x(z) - p_{\text{accept}}(z; y) \cdot p_y(z)|\, \mathrm{d}z\,,$$

$$r_x = 1 - \mathbb{E}_{z \sim \mathcal{P}_x}[p_{\text{accept}}(z; x) \cdot \mathbf{1}\{z \in \mathcal{K}\}]\,.$$

Kook and Vempala (2024) remark that working with this is essential to analyse DikinWalk, and use sophisticated techniques to obtain a bound for the above quantity where $\mathcal{P}_x = \mathcal{N}(x, 2h \cdot \mathscr{G}(x)^{-1})$ and $\mathbf{P} = \{\mathcal{P}_x : x \in \mathcal{K}\}$. Due to the drift correction in MAPLA, we are able to take a relatively simpler approach that involves giving bounds on $\mathrm{d}_{\mathrm{TV}}(\mathcal{T}_x, \mathcal{P}_x)$ and $\mathrm{d}_{\mathrm{TV}}(\mathcal{P}_x, \mathcal{P}_y)$ and noting that

$$\mathrm{d}_{\mathrm{TV}}(\mathcal{T}_x, \mathcal{T}_y) \leq \mathrm{d}_{\mathrm{TV}}(\mathcal{T}_x, \mathcal{P}_x) + \mathrm{d}_{\mathrm{TV}}(\mathcal{P}_x, \mathcal{P}_y) + \mathrm{d}_{\mathrm{TV}}(\mathcal{T}_y, \mathcal{P}_y)\,.$$

This simpler approach is known to yield a vacuous bound $\mathrm{d}_{\mathrm{TV}}(\mathcal{T}_x, \mathcal{T}_y)$ for the analysis of DikinWalk; however this is not the case for the proposal Markov chain in MAPLA due to the inclusion of the drift correction $-h \cdot \mathscr{G}^{-1}\nabla f$ to the proposal distribution of DikinWalk that MAPLA is based on, and results in a crisper and less complicated proof for MAPLA.

## 5. Conclusion

In summary, we propose a new first-order algorithm for the constrained sampling problem called MAPLA, which is algorithmically motivated by the natural gradient descent algorithm in optimisation. This method performs a Metropolis adjustment of the Markov chain resulting from an approximate version of the preconditioned Langevin algorithm (PLA), and supersedes the Metropolis-adjusted Mirror Langevin algorithm proposed by Srinivasan et al. (2024) by working with a general metric $\mathscr{G}$. We derive non-asymptotic mixing time guarantees for our method under a variety of assumptions made on the target distribution $\Pi$ and the metric $\mathscr{G}$. We find that when $\mathscr{G}$ satisfies certain stronger notions of self-concordance, the dimension dependence in the mixing time guarantee is strictly better than that obtain with standard self-concordance. Our numerical experiments showcase how including first-order information about the $f$ through $\nabla f$ using the natural gradient can be beneficial in comparison to DikinWalk which only uses $f$, and could motivate the design of more sophisticated first-order methods for constrained sampling.

Several open questions remain. We exclude the correction term $(\nabla \cdot \mathscr{G}^{-1})$ in PLA, which is the key difference compared to ManifoldMALA. Notwithstanding the computational difficulty, it would be interesting to see the what the effect of including this correction term would be on the mixing time. More theoretically, drawing from the discussion of the results, it would be interesting to identify other scenarios where the weaker sufficient condition pertaining to $\|\nabla f(\cdot)\|_{\mathscr{G}(\cdot)^{-1}}$ holds, and its implications for the mixing time of MAPLA. Another course to eliminating the gradient upper bound is showing that the above local norm quantity concentrates when $f$ and $\mathscr{G}$ satisfy certain properties such as the $(\mu, \mathscr{G})$ and $(\lambda, \mathscr{G})$-curvature lower and upper bounds, as done in more recent analyses (Lee et al., 2020) in the case where $\mathscr{G} = \mathrm{I}_{d \times d}$ i.e., MALA. Algorithmically, it would be also be interesting to find other candidate proposal Markov chains that can yield similar or better mixing time guarantees. While PLA serves as a useful proposal Markov chain, its efficacy as a standalone algorithm (with a projection to ensure feasibility) is not investigated in this work. As noted earlier, PLA is likely to be biased, but whether this bias is *vanishing* (i.e., when the bias $\to 0$ as $h \to 0$) under certain conditions on the metric $\mathscr{G}$ would be interesting to check.

## Acknowledgments

We would like to thank the reviewers for their feedback, and Nawaf Bou-Rabee for helpful remarks. Vishwak Srinivasan was supported by a Simons Foundation Collaboration on Theory of Algorithmic Fairness Grant. Andre Wibisono was supported by NSF Award CCF #2403391.

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

# Appendices

**Organisation:** This appendix is organised as follows. Appendix A includes additional technical details that were deferred from Section 1, specifically about lifting and the relation between MAPLA and MAMLA. Appendix B discusses other related work to MAPLA, especially about methods that have been proposed others that geometric random walks, or the mirror Langevin algorithm. Appendix C gives the pseudocode for MAPLA and discusses choices for $\mathscr{G}$. Appendix D gives detailed proofs for the theorems in Section 4, and finally Appendix E contains a collection of miscellaneous results used in this work.

## Appendix A. Technical details deferred from Section 1

### A.1. Details about lifting

Recall that *lifting* takes a potential $f$ and a domain $\mathcal{K}$, and defines a new domain $\widetilde{\mathcal{K}}$ and potential $\widetilde{f}$ (referred to as the *lifted domain* and *lifted potential* respectively) which are stated below.

$$\widetilde{\mathcal{K}} = \left\{ y = (x,t) : x \in \mathcal{K}, \ t \in \mathbb{R}^N, f_i(x) \le t_i \ \forall \ i \in [N] \right\} \ ,$$
$$\widetilde{f}(y) = \widetilde{f}(x,t) = [\underbrace{0,\ldots,0}_{d \text{ times}}, \underbrace{1,\ldots,1}_{N \text{ times}}]^\top (y) \ .$$

If every $f_i$ is convex, then the lifted domain $\widetilde{\mathcal{K}}$ is convex. This is because the lifted domain is formed by the intersection of $N$ epigraphs defined by each $f_i$. Lifting described here is the sampling analogue of converting an optimisation problem of the form $\min_{x \in \mathcal{K}} f(x)$ into $\min_{(x,t) \in \widetilde{\mathcal{K}}} \mathbf{1}^\top t$. The principle behind *lifting* for sampling is the fact that the marginal distribution of the first $d$ elements of $y \sim \widetilde{\Pi}$ coincides with the distribution of $x \sim \Pi$. This is made evident by the calculation below.

$$\int_{t \in \widetilde{\mathcal{K}}} e^{-\widetilde{f}((x,t))} \mathrm{d}t = \prod_{i=1}^N \int_{t_i \ge f_i(x)} e^{-t_i} \mathrm{d}t_i = \prod_{i=1}^N e^{-f_i(x)} = e^{-f(x)} \ .$$

### A.2. Relation between MAPLA and MAMLA

Mirror Langevin methods are based on another approach to solve constrained optimisation problems called mirror methods (Nemirovski and Yudin, 1983), and their key feature is the use of a Legendre type function (Rockafellar, 1970, Chap. 26) $\phi$ referred to as the *mirror function*. The primary role of the mirror function is to define a mirror map $\nabla\phi$ which maps points in $\mathcal{K}$ (called the *primal* space) to $\mathcal{K}^\star = \mathrm{range}(\nabla\phi)$ (called the *dual space*) where updates are performed, and are mapped back to $\mathcal{K}$ using the inverse of $\nabla\phi$. Since $\phi$ is of Legendre type, the inverse of $\nabla\phi$ is $\nabla\phi^\star$, where $\phi^\star$ is the convex conjugate of $\phi$. The mirror function also induces non-Euclidean geometric structures on $\mathcal{K}$ and $\mathcal{K}^\star$ through metrics $\nabla^2\phi$ and $\nabla^2\phi^\star$ respectively. The proposal distribution $\mathcal{P}_X$ of MAMLA is the law of the iterate obtained with one step of the Mirror Langevin algorithm (Zhang et al., 2020) from $X$ and is defined as

$$\mathcal{P}_X = \mathrm{Law}(\nabla\phi^\star(Z)); \quad Z \sim \widetilde{\mathcal{P}}_X = \mathcal{N}\left(\nabla\phi(X) - h \cdot \nabla f(X), \ 2h \cdot \nabla^2\phi(X)\right) \ .$$

Let $Y = \nabla\phi(X)$ and $f^\star = f \circ \nabla\phi^\star$. Using the property that $\nabla\phi \circ \nabla\phi^\star = \mathrm{Id}$, it can be shown that the distribution $\widetilde{\mathcal{P}}_X$ above is equivalent to $\mathcal{N}\left(Y - h \cdot \nabla^2\phi^\star(Y)^{-1}\nabla f^\star(Y),\ 2h \cdot \nabla^2\phi^\star(Y)^{-1}\right)$. This lends to the interpretation of MAMLA as the *dual* version of MAPLA with the metric $\nabla^2\phi^\star$ in the dual space, which also complements the connection between the mirror descent and natural gradient descent methods in optimisation (Raskutti and Mukherjee, 2015). However, a central challenge with mirror methods is computing $\nabla\phi^\star$ which in generality requires solving the convex program that defines $\phi^\star$, although in a few cases, a closed form expression exists.

## Appendix B.  Additional related work deferred from Section 2

**Other discretisations of WLD**    Recall that the primary bottleneck in implementing WLA is that in general, computing $\nabla \cdot \mathscr{G}^{-1}$ for WLA can be difficul. This was also previously highlighted in Bou-Rabee et al. (2014) who note that WLA corresponds to an Ermak-Cammon scheme for Brownian dynamics with hydrodynamic interactions. Bou-Rabee et al. (2014) propose a novel discretisation for WLD based on a two-stage Runge-Kutta scheme in combination with a Metropolis adjustment, and give asymptotic guarantees for this discretisation.

**Other constrained sampling techniques**    In addition to proposing ManifoldMALA, Girolami and Calderhead (2011) also adapt Hamiltonian Monte Carlo to the geometry of the state space and propose the *Riemannian* Hamiltonian Monte Carlo algorithm (RHMC). The primary challenge with RHMC pertains to implementation, as its discretisation is not as straightforward as WLA for WLD. Lee and Vempala (2018); Gatmiry et al. (2024) derive guarantees for RHMC given an ideal integrator, while Kook et al. (2023); Noble et al. (2023) investigate discretised integrators for RHMC. Recent advances in proximal methods for sampling (Chen et al., 2022) have also led to novel samplers for sampling over non-Euclidean spaces using the log-Laplace transform (Gopi et al., 2023), and for uniform sampling from compact $\mathcal{K}$ under minimal assumptions (Kook et al., 2024). Another class of first-order approaches for constrained sampling (Brosse et al., 2017; Gürbüzbalaban et al., 2024) propose obtaining a good unconstrained approximation $\widetilde{\Pi}$ of the target $\Pi$, but this does not eliminate the likelihood of iterates lying outside $\mathcal{K}$. This enables running the simpler unadjusted Langevin algorithm over $\widetilde{\Pi}$ to obtain approximate samples from $\Pi$. Recent work by Bonet et al. (2024) translate mirror and natural gradient methods for minimising functions over Euclidean spaces to instead minimise functionals over space of probability measures endowed with the Wasserstein metric.

## Appendix C.  Additional details about MAPLA

### C.1.  Examples of metric $\mathscr{G}$ for certain domains

Let $\mathcal{K}$ be a polytope of the form $\{x : x \in \mathbb{R}^d,\ a_i^\top x \le b_i \ \forall\, i \in [m]\}$ where $m \ge d$. A natural choice of $\mathscr{G}$ for this domain is the Hessian of its log-barrier function, and can be generalised to incorporate weights for each linear constraint as

$$\mathscr{G}_{\{Ax \le b\}}(x) = A_x^\top W A_x \text{ where } [A_x]_i^\top = \frac{a_i}{(b_i - a_i^\top x)} \in \mathbb{R}^d \text{ and } W = \mathrm{diag}(\boldsymbol{w}) \,.$$

The vector $\boldsymbol{w} \in \mathbb{R}^m$ contains non-negative entries, and may depend on $x$. Special cases include

1. the Vaidya metric (Vaidya, 1996), where $w_i = \frac{d}{m} + a_i^\top (A_x^\top A_x)^{-1} a_i$,

2. the John metric (Gustafson and Narayanan, 2018), where $\boldsymbol{w}$ is defined as

$$\boldsymbol{w} = \operatorname*{argmin}_{w \in \mathbb{R}_+^m} \log \det A_x^\top W A_x \quad \text{such that} \quad \mathbf{1}^\top w = m \ .$$

3. the $p$-Lewis-weights metric (Lee and Sidford, 2019) which modifies the John metric to ensure $\boldsymbol{w}$ varies smoothly as a function $x$. More precisely, it is defined as

$$\boldsymbol{w} = \operatorname*{argmax}_{w \in \mathbb{R}_+^m} - \log \det(A_x^\top W^{1 - 2/p} A_x) + (1 - 2/p) \cdot \mathbf{1}^\top w$$

Polytopes are regions defined by intersection of half-spaces defined by a collection of linear inequalities. Moving past linear inequalities, we consider quadratic inequalities, which define ellipsoids. An ellipsoid is defined as $\{x : x \in \mathbb{R}^d, \|x - c\|_D^2 \le 1\}$ where $D \in \mathbb{R}^{d \times d}$ is a symmetric positive definite matrix and $c \in \mathbb{R}^d$. Analogous to the polytope, a natural choice of $\mathscr{G}$ here is the Hessian of its log-barrier function. An ellipsoid can also be interpreted at the 1-sublevel set of a quadratic function. Relatedly, the epigraph of this quadratic function is the indexed union of all its $t$-sublevel sets for $t \ge 0$, and this arises when lifting the Gaussian distribution as remarked previously in section 1. An example of a metric for this domain is

$$\mathscr{G}_{\text{Ellip}(c,D)}(x, t) = \nabla_{(x,t)}^2 \varphi_{\text{Ellip}(c,D)}(x, t) \ ; \qquad \varphi_{\text{Ellip}(c,D)}(x, t) = -\log(t - \|x - c\|_D^2) \ .$$

Generalising from the $\ell_1$-ball (a polytope formed by $m = 2^d$ constraints), and the $\ell_2$-ball (an ellipsoid with $c = \mathbf{0}$ and $D = \mathrm{I}_{d \times d}$), we have the $\ell_p$-ball for an arbitrary $p \ge 1$ defined as $\{x : x \in \mathbb{R}^d, \|x\|_p^p \le 1\}$. Using the Hessian of the log-barrier of $-\log(1 - \|x\|_p^p)$ as the metric for this domain poses non-trivial difficulties theoretically. To circumvent this, we use the following equivalence

$$\|x\|_p \le 1 \Leftrightarrow \exists \, v \in \mathbb{R}^d \text{ such that } |x_i|^p \le v_i \, \forall \, i \in \{1, \ldots, d\} \text{ and } \sum_{i=1}^d v_i \le 1 \ ,$$

which suggests considering an extended domain in $\mathbb{R}^{2d}$. This extended domain is the intersection of the halfspace $\{v : v \in \mathbb{R}^d, \mathbf{1}^\top v \le 1\}$ and the product of $d$ subsets of $\mathbb{R}^2$ given as $\prod_{i=1}^d \{(x_i, v_i) : |x_i|^p \le v_i\}$. Therefore, we can define a metric for the extended domain given a metric $\mathscr{G}_{2,p}$ for the set $\{(y, t) : |y|^2 \le t\} \subset \mathbb{R}^2$. A popular choice for $\mathscr{G}_{2,p}(y, t)$ is $\nabla_{(y,t)}^2 \varphi_{\ell_p}(y, t)$ where $\varphi_{\ell_p}(y, t) = -\log(t) - \log(t^{2/p} - y^2)$, with which we can be define the metric for the extended domain as

$$\mathscr{G}_p(x, v) = \mathfrak{P} \left( \bigoplus_{i=1}^d \mathscr{G}_{2,p}(x_i, v_i) \right) \mathfrak{P}^\top + \mathbf{0}_{d \times d} \oplus \mathscr{G}_{\{\mathbf{1}^\top v \le 1\}}(v) \ .$$

The operation $A \oplus B$ creates a block diagonal matrix with $A$ and $B$ on the diagonal[5], and $\mathfrak{P}$ is a permutation matrix that ensures consistency with respect to the ordering of inputs ($x$ first, $v$ second). This extension procedure is more generally applicable to level sets of separable functions (Nesterov,

---

5. also referred to as the *direct sum*.

2018, §5.4.8). Another example is the entropic ball defined as $\{x : x \in \mathbb{R}^d_+, \sum_{i=1}^d x_i \log x_i \leq 1\}$. Similar to the $\ell_p$-ball, the metric $\mathscr{G}_{\mathsf{ent}}$ for the extended entropic ball is dependent on the metric $\mathscr{G}_{2,\mathsf{ent}}(y, t)$ for the 2-dimensional set $\{(y, t) : y \log y \leq t\}$. A viable option for $\mathscr{G}_{2,\mathsf{ent}}(y, t)$ is $\nabla^2_{(y,t)} \varphi_{\mathsf{ent}}(y, t)$ where $\varphi_{\mathsf{ent}}(y, t) = -\log y - \log(t - y \log y)$.

We refer the reader to Nesterov and Nemirovski (1994, §5.2) for a general calculus to combine metrics. Their specific focus is the self-concordance of these combinations, which is central to the analysis of interior-point methods for optimisation. Kook and Vempala (2024) investigate other properties ("self-concordance$_{++}$") of such metrics which are pertinent for constrained sampling, and are useful for deriving mixing time guarantees for DikinWalk and as we show, for MAPLA.

### C.2. Pseudocode for MAPLA

**Algorithm 1:** Metropolis-adjusted Preconditioned Langevin Algorithm (MAPLA)

**Input** : Potential $f$ of $\Pi$, convex support $\mathcal{K} \subset \mathbb{R}^d$, metric $\mathscr{G} : \mathrm{int}(\mathcal{K}) \to \mathbb{S}^d_+$, step size $h > 0$, iterations $K$, initial distribution $\Pi_0$

Sample $x_0 \sim \Pi_0$.

**for** $k \leftarrow 0$ **to** $K - 1$ **do**

    Sample a random vector $\xi_k \sim \mathcal{N}(\mathbf{0}, \mathrm{I}_{d \times d})$.

    Generate proposal $z = x_k - h \cdot \mathscr{G}(x_k)^{-1} \nabla f(x_k) + \sqrt{2h} \cdot \mathscr{G}(x_k)^{-1/2} \xi_k$.

    **if** $z \notin \mathcal{K}$ **then**

        | Set $x_{k+1} = x_k$.

    **else**

        Compute acceptance ratio $p_{\mathrm{accept}}(z; x_k)$ defined in eq. (1), where $p_y$ is the density of $\mathcal{N}\left(y - h \cdot \mathscr{G}(y)^{-1} \nabla f(y), \; 2h \cdot \mathscr{G}(y)^{-1}\right)$.

        Obtain $U \sim \mathrm{Unif}([0, 1])$.

        **if** $U \leq p_{\mathrm{accept}}(z; x_k)$ **then**

            | Set $x_{k+1} = z$.

        **else**

            | Set $x_{k+1} = x_k$.

        **end**

    **end**

**end**

**Output:** $x_K$

## Appendix D. Proofs of the theorems in Section 4

In this section, we give the proofs of the theorems in Section 4. Section D.1 states the key lemmas which form the proofs of the theorems in Section 4, and we state these proofs in Section D.2. Next in Section D.3, we focus on the proofs of these key lemmas which involve the new isoperimetric inequality that we mentioned earlier, whose proof concludes this section in Section D.4.

### D.1. A pathway to obtain mixing time guarantees

Let $\mathbf{Q} = \{\mathcal{Q}_x : x \in \mathcal{K}\}$ be a Markov chain that is reversible with respect to $\rho$ supported over $\mathcal{K}$. The conductance / $s$-conductance of $\mathbf{Q}$ (defined in Section 4.1) quantifies the likelihood of escaping

a set in a worst-case sense, and hence intuitively determines how quickly a $\mathbf{Q}$ mixes to its stationary distribution. This intuition is made more precise in the following result from Vempala (2005) (a collection of special cases arising from the classical analysis of Lovász and Simonovits (1993)) that relates the conductance / $s$-conductance of $\mathbf{Q}$ to its mixing time. Recall that $\mathbb{T}_{\mathbf{Q}}^k$ is the transition operator defined by $\mathbf{Q}$ applied $k$ times.

**Proposition 12 (Vempala (2005, Corr. 3.5))** *Let* $\mathbf{Q} = \{\mathcal{Q}_x : x \in \mathcal{K}\}$ *be a lazy, reversible Markov chain with stationary distribution* $\rho$, *and let* $\rho_0$ *be a distribution whose support is contained in* $\mathcal{K}$.

*1. If* $\rho_0 \in \mathsf{Warm}(L_\infty, M, \rho)$, *then*

$$\mathrm{d}_{\mathrm{TV}}(\mathbb{T}_{\mathbf{Q}}^k \rho_0, \rho) \leq \sqrt{M} \cdot \left(1 - \frac{{\Phi_{\mathbf{Q}}}^2}{2}\right)^k .$$

*2. If* $\rho_0 \in \mathsf{Warm}(L_1, M, \rho)$, *then for any* $\gamma > 0$

$$\mathrm{d}_{\mathrm{TV}}(\mathbb{T}_{\mathbf{Q}}^k \rho_0, \rho) \leq \gamma + \sqrt{\frac{M}{\gamma}} \cdot \left(1 - \frac{{\Phi_{\mathbf{Q}}}^2}{2}\right)^k .$$

*3. If* $\rho_0 \in \mathsf{Warm}(L_\infty, M, \rho)$, *then for any* $s \in (0, 1/2)$

$$\mathrm{d}_{\mathrm{TV}}(\mathbb{T}_{\mathbf{Q}}^k \rho_0, \rho) \leq M \cdot s + M \cdot \left(1 - \frac{{\Phi_{\mathbf{Q}}^s}^2}{2}\right)^k .$$

Therefore, given a lower bound on the conductance / $s$-conductance of $\mathbf{Q}$, we can obtain non-asymptotic mixing time guarantees for the Markov chain $\mathbf{Q}$ from a warm initial distribution. Indeed, these lower bounds have to be bounded away from $0$ for a meaningful mixing time guarantee.

As discussed briefly previously in Section 4.2.1 we use the one-step overlap technique pioneered by Lovász (1999) to obtain lower bounds on the conductance / $s$-conductance. The following lemma formally states how the one-step overlap yields the likelihood of escaping a set.

**Lemma 13** *Consider a Markov chain* $\mathbf{T} = \{\mathcal{T}_x : x \in \mathcal{K}\}$ *that is reversible with respect to a log-concave distribution* $\Pi$ *supported on* $\mathcal{K}$. *Let the metric* $\mathscr{G} : \mathrm{int}(\mathcal{K}) \to \mathbb{S}_+^d$ *be self-concordant and* $\nu$-*symmetric. Assume that there exists a convex subset* $\mathcal{S}$ *of* $\mathrm{int}(\mathcal{K})$ *such that for any* $x, y \in \mathcal{S}$,

$$\|x - y\|_{\mathscr{G}(y)} \leq \Delta \Rightarrow \mathrm{d}_{\mathrm{TV}}(\mathcal{T}_x, \mathcal{T}_y) \leq \frac{1}{4}$$

*where* $\Delta \leq \frac{1}{2}$. *If the potential* $f$ *of* $\Pi$ *satisfies a* $(\mu, \mathscr{G})$-*curvature lower bound, then for any measurable partition* $\{A_1, A_2\}$ *of* $\mathcal{K}$,

$$\int_{A_1} \mathcal{T}_x(A_2)\, \pi(x)\mathrm{d}x \geq \frac{3\Delta}{16} \cdot \min\left\{1, \max\left\{\frac{\widetilde{\mu}}{4}, \frac{1}{8\sqrt{\nu}}\right\}\right\} \cdot \min\left\{\Pi(A_1 \cap \mathcal{S}), \Pi(A_2 \cap \mathcal{S})\right\} ,$$

*where* $\widetilde{\mu} = \frac{\mu}{8 + 4\sqrt{\mu}}$.

We give a proof of Lemma 13 in Section D.3.1. This technique has been widely employed to obtain mixing time upper bounds for several MCMC algorithms that induce Markov chains which are reversible with respect to the target distribution which can either be constrained or unconstrained.

We recall that our focus is on obtaining lower bounds on the conductance / $s$-conductance of Markov chain $\mathbf{T}$ induced by MAPLA, which results from the Metropolis adjustment of the Markov chain $\mathbf{P}$ defined by a single step of PLA. To use Lemma 13 for $\mathbf{T}$, we require checking the existence of a convex subset $\mathcal{S}$ where the one-step overlap assumed in its statement holds. We identify $\mathcal{S}$ based on properties of the $f$ of the target $\Pi$ and state these in the proofs of the theorems. Our strategy to establish the one-step overlap in this subset is to bound $\mathrm{d_{TV}}(\mathcal{P}_x, \mathcal{P}_y)$ and $\mathrm{d_{TV}}(\mathcal{T}_x, \mathcal{P}_x)$ for any $x, y \in \mathcal{S}$, and then use the triangle inequality for the TV distance like so:

$$\mathrm{d_{TV}}(\mathcal{T}_x, \mathcal{T}_y) \leq \mathrm{d_{TV}}(\mathcal{T}_x, \mathcal{P}_x) + \mathrm{d_{TV}}(\mathcal{P}_x, \mathcal{P}_y) + \mathrm{d_{TV}}(\mathcal{T}_y, \mathcal{P}_y) \,.$$

Note that $\mathbf{T}$ is defined based on the potential $f$ of $\Pi$, the metric $\mathscr{G}$, and the step size $h > 0$.

**Lemma 14** *Let the metric $\mathscr{G} : \mathrm{int}(\mathcal{K}) \to \mathbb{S}_+^d$ be self-concordant. Assume that there exists a convex set $\mathcal{S} \subseteq \mathrm{int}(\mathcal{K})$ and $\mathsf{U}_{f,\mathcal{S}} \geq 0$ such that for any $x \in \mathcal{S}$*

$$\sup_{z: \|z-x\|_{\mathscr{G}(x)} \leq \frac{1}{2}} \|\nabla f(z)\|_{\mathscr{G}(z)^{-1}} \leq \mathsf{U}_{f,\mathcal{S}} \,.$$

*Then, for any $x, y \in \mathcal{S}$ such that $\|x - y\|_{\mathscr{G}(y)} \leq \frac{\sqrt{h}}{10}$ with $h < 1$,*

$$\mathrm{d_{TV}}(\mathcal{P}_x, \mathcal{P}_y) \leq \frac{1}{2} \cdot \sqrt{\frac{h \cdot d}{20} + \frac{1}{80} + \frac{9h \cdot \mathsf{U}_{f,\mathcal{S}}^2}{2}} \,.$$

**Lemma 15** *Let the metric $\mathscr{G} : \mathrm{int}(\mathcal{K}) \to \mathbb{S}_+^d$ be self-concordant, and the potential $f : \mathrm{int}(\mathcal{K}) \to \mathbb{R}$ satisfy $(\lambda, \mathscr{G})$-curvature upper bound. Assume that there exists a convex set $\mathcal{S} \subseteq \mathrm{int}(\mathcal{K})$ and $\mathsf{U}_{f,\mathcal{S}} \geq 0$ such that for any $x \in \mathcal{S}$,*

$$\sup_{z: \|z-x\|_{\mathscr{G}(x)} \leq \frac{1}{2}} \|\nabla f(z)\|_{\mathscr{G}(z)^{-1}} \leq \mathsf{U}_{f,\mathcal{S}} \,.$$

*If the step size $h > 0$ is bounded from above as*

$$h \leq c \cdot \min \left\{ \frac{1}{\mathsf{U}_{f,\mathcal{S}}^2}, \ \frac{1}{\mathsf{U}_{f,\mathcal{S}}^{2/3}}, \ \frac{1}{(\mathsf{U}_{f,\mathcal{S}} \cdot \lambda)^{2/3}}, \ \frac{1}{d \cdot \lambda}, \ \frac{1}{d^3} \right\}; \quad c \leq \frac{1}{20}$$

*then for any $x \in \mathcal{S}$*

$$\mathrm{d_{TV}}(\mathcal{T}_x, \mathcal{P}_x) \leq \frac{1}{16} \,.$$

**Lemma 16** *Let the metric $\mathscr{G} : \mathrm{int}(\mathcal{K}) \to \mathbb{S}_+^d$ be strongly, $\alpha$-lower trace, and average self-concordant, and the potential $f : \mathrm{int}(\mathcal{K}) \to \mathbb{R}$ satisfy $(\lambda, \mathscr{G})$-curvature upper bound. Assume that there exists a convex set $\mathcal{S} \subseteq \mathrm{int}(\mathcal{K})$ and $\mathsf{U}_{f,\mathcal{S}} \geq 0$ such that for any $x \in \mathcal{S}$,*

$$\sup_{z: \|z-x\|_{\mathscr{G}(x)} \leq \frac{1}{2}} \|\nabla f(z)\|_{\mathscr{G}(z)^{-1}} \leq \mathsf{U}_{f,\mathcal{S}} \,.$$

*If the step size $h > 0$ is bounded from above as*

$$h \leq c \cdot \min \left\{ \frac{1}{\mathsf{U}_{f,\mathcal{S}}^2}, \; \frac{1}{(\mathsf{U}_{f,\mathcal{S}} \cdot (\alpha + 4))^{2/3}}, \; \frac{1}{(\mathsf{U}_{f,\mathcal{S}} \cdot \lambda)^{2/3}}, \; \frac{1}{d \cdot \lambda}, \; \frac{1}{d \cdot \mathsf{U}_{f,\mathcal{S}}}, \; \frac{1}{d \cdot (\alpha + 4)} \right\}; \quad c \leq \frac{1}{20},$$

*then for any $x \in \mathcal{S}$*

$$\mathrm{d}_{\mathrm{TV}}(\mathcal{T}_x, \mathcal{P}_x) \leq \frac{1}{16} .$$

## D.2. Complete proofs of Theorems 9 to 11

With the key lemmas (Lemmas 13 to 16), we can now prove the main theorems.

**Proof** [Proofs of Theorems 9 and 10]

In both theorems, the potential is assumed to satisfy the $(\beta, \mathscr{G})$-gradient upper bound condition, which implies by definition that

$$\sup_{x \in \mathrm{int}(\mathcal{K})} \|\nabla f(x)\|_{\mathscr{G}(x)^{-1}} \leq \beta .$$

Thus, we can now use Lemmas 15 and 16 with $\mathcal{S} \leftarrow \mathrm{int}(\mathcal{K})$, and $\mathsf{U}_{f,\mathcal{S}} \leftarrow \beta$. The resulting upper bounds on the step size from these lemmas are exactly the bounds $b_{\mathsf{SC}}(d, \lambda, \beta)$ and $b_{\mathsf{SC}_{++}}$ in Theorems 9 and 10 respectively. Hence, for any $x \in \mathrm{int}(\mathcal{K})$, we have

$$\mathrm{d}_{\mathrm{TV}}(\mathcal{T}_x, \mathcal{P}_x) \leq \frac{1}{16} .$$

In Theorem 9, the metric $\mathscr{G}$ is assumed to be self-concordant, and in Theorem 10, the metric $\mathscr{G}$ is assumed to be strongly self-concordant which implies that it is self-concordant as well. Due to this, we can use Lemma 14 with $\mathcal{S} \leftarrow \mathrm{int}(\mathcal{K})$ and $\mathsf{U}_{f,\mathcal{S}} \leftarrow \beta$ to obtain for any $x, y \in \mathrm{int}(\mathcal{K})$ that satisfy $\|x - y\|_{\mathscr{G}(y)} \leq \frac{\sqrt{h}}{10}$ for $h < 1$,

$$\mathrm{d}_{\mathrm{TV}}(\mathcal{P}_x, \mathcal{P}_y) \leq \frac{1}{2} \cdot \sqrt{\frac{h \cdot d}{20} + \frac{1}{80} + \frac{9h \cdot \beta^2}{2}} .$$

The bounds $b_{\mathsf{SC}}(d, \lambda, \beta)$ and $b_{\mathsf{SC}_{++}}$ ensure that $h \leq \frac{1}{20\beta^2}$ and $h \leq \frac{1}{20d}$, which yields

$$\mathrm{d}_{\mathrm{TV}}(\mathcal{P}_x, \mathcal{P}_y) \leq \frac{1}{2} \cdot \sqrt{\frac{h \cdot d}{20} + \frac{1}{80} + \frac{9h \cdot \beta^2}{2}} \leq \frac{1}{2} \cdot \sqrt{\frac{1}{400} + \frac{1}{80} + \frac{9}{40}} \leq \frac{1}{8} .$$

Combining the bounds on $\mathrm{d}_{\mathrm{TV}}(\mathcal{T}_x, \mathcal{P}_x)$ and $\mathrm{d}_{\mathrm{TV}}(\mathcal{P}_x, \mathcal{P}_y)$, we obtain that for any $x, y \in \mathrm{int}(\mathcal{K})$ satisfying $\|x - y\|_{\mathscr{G}(y)} \leq \frac{\sqrt{h}}{10}$,

$$\mathrm{d}_{\mathrm{TV}}(\mathcal{T}_x, \mathcal{T}_y) \leq \mathrm{d}_{\mathrm{TV}}(\mathcal{T}_x, \mathcal{P}_x) + \mathrm{d}_{\mathrm{TV}}(\mathcal{P}_x, \mathcal{P}_y) + \mathrm{d}_{\mathrm{TV}}(\mathcal{T}_y, \mathcal{P}_y) \leq \frac{1}{16} + \frac{1}{8} + \frac{1}{16} = \frac{1}{4} .$$

With this result, we apply Lemma 13 with $\Delta \leftarrow \frac{\sqrt{h}}{10}$, $\mathcal{S} \leftarrow \mathrm{int}(\mathcal{K})$ as the potential $f$ is assumed to satisfy a $(\mu, \mathscr{G})$-curvature lower bound and the metric $\mathscr{G}$ is assumed to be $\nu$-symmetric. From this, we know that there exists a universal constant $C > 0$ such that for any $A \subseteq \mathcal{K}$,

$$\int_A \mathcal{T}_x(\mathcal{K} \setminus A) \pi(x) \, \mathrm{d}x \geq C \cdot \sqrt{h} \cdot \min \left\{ 1, \max \left\{ \widetilde{\mu}, \frac{1}{\sqrt{\nu}} \right\} \right\} \cdot \min\{\Pi(A), \Pi(\mathcal{K} \setminus A)\} .$$

Above, we have used the fact that $\mathcal{K}$ is a convex subset of $\mathbb{R}^d$ and $\Pi$ is log-concave, and hence

$$\Pi(A \cap \text{int}(\mathcal{K})) = \Pi(A); \quad \Pi((\mathcal{K} \setminus A) \cap \text{int}(\mathcal{K})) = \Pi(\mathcal{K} \setminus A) .$$

This directly implies by definition that

$$\Phi_{\mathbf{T}} \geq C \cdot \sqrt{h} \cdot \min\left\{1, \max\left\{\widetilde{\mu}, \frac{1}{\sqrt{\nu}}\right\}\right\} . \tag{3}$$

For the final step of this proof, we invoke Proposition 12. We have the following cases.

**For** $\Pi_0 \in \mathcal{C} := \mathsf{Warm}(L_\infty, M, \Pi)$**:**

$$d_{\mathrm{TV}}(\mathbb{T}_{\mathbf{T}}^k \Pi_0, \Pi) \leq \sqrt{M} \cdot \exp\left(-\frac{k \cdot \Phi_{\mathbf{T}}^2}{2}\right) .$$

By setting the right hand side to be less than $\delta$ for $\delta \in (0, 1/2)$ and using Equation (3), we get

$$\tau_{\mathrm{mix}}(\delta; \mathbf{T}, \mathcal{C}) = \frac{2}{\Phi_{\mathbf{T}}^2} \cdot \log\left(\frac{\sqrt{M}}{\delta}\right) \leq \frac{C'}{h} \cdot \max\left\{1, \min\left\{\frac{1}{\widetilde{\mu}^2}, \nu\right\}\right\} \cdot \log\left(\frac{\sqrt{M}}{\delta}\right)$$

for some universal positive constant $C'$.

**For** $\Pi_0 \in \mathcal{C} := \mathsf{Warm}(L_1, M, \Pi)$**:**

$$d_{\mathrm{TV}}(\mathbb{T}_{\mathbf{T}}^k \Pi_0, \Pi) \leq \gamma + \sqrt{\frac{M}{\gamma}} \cdot \exp\left(-\frac{k \cdot \Phi_{\mathbf{T}}^2}{2}\right)$$

for any $\gamma > 0$. For $\delta \in (0, 1/2)$, by setting $\gamma = \frac{\delta}{2}$ and the second term on the right hand side to be less than $\frac{\delta}{2}$, the upper bound is at most $\delta$. With Equation (3), we get

$$\tau_{\mathrm{mix}}(\delta; \mathbf{T}, \mathcal{C}) = \frac{6}{\Phi_{\mathbf{T}}^2} \cdot \log\left(\frac{(2M)^{1/3}}{\delta}\right) \leq \frac{C''}{h} \cdot \max\left\{1, \min\left\{\frac{1}{\widetilde{\mu}^2}, \nu\right\}\right\} \cdot \log\left(\frac{M^{1/3}}{\delta}\right)$$

for some universal positive constant $C''$.

∎

**Proof** [Proof of Theorem 11] For $s \in (0, 1/2)$,

$$\mathcal{K}_s \stackrel{\text{def}}{=} \left\{x \in \text{int}(\mathcal{K}) : \|\sigma\|_{\mathscr{G}(x)^{-1}}^2 \leq 25d^2 \cdot \log^2\left(\frac{1}{s}\right)\right\} ,$$

where $\sigma = \nabla f(x)$ for all $x \in \text{int}(\mathcal{K})$. Equivalently,

$$\sup_{x \in \mathcal{C}_s} \|\nabla f(x)\|_{\mathscr{G}(x)^{-1}} \leq 5d \cdot \log\left(\frac{1}{s}\right) .$$

In Kook et al. (2023), we have the following properties of $\mathcal{K}_s$:

**Lem. 43:** If $\mathscr{G}$ is self-concordant, $\Pi(\mathcal{K}_s) \geq 1 - s$.

**Lem. 42:** If $\mathscr{G}$ satisfies $\mathrm{D}^2\mathscr{G}(x)[v, v] \succeq \mathbf{0}$ for all $x \in \mathrm{int}(\mathcal{K})$ and $v \in \mathbb{R}^d$, $\mathcal{K}_s$ is convex.

In the setting of Theorem 11, the metric $\mathscr{G}$ is assumed to satisfy both preconditions in the properties above. Additionally, since the metric $\mathscr{G}$ is self-concordant, from Lemma 26 we have for any $z \in \mathcal{E}_x^{\mathscr{G}}(1)$ (defined in Equation (2)) and $u \in \mathbb{R}^d$ that

$$(1 - \|z - x\|_{\mathscr{G}(x)})^2 \cdot \mathscr{G}(x) \preceq \mathscr{G}(z) \Leftrightarrow \|u\|_{\mathscr{G}(x)} \leq \frac{\|u\|_{\mathscr{G}(z)}}{(1 - \|z - x\|_{\mathscr{G}(x)})} \ .$$

As a corollary, if $\|z - x\|_{\mathscr{G}(x)} \leq \frac{1}{2}$ and $\|u\|_{\mathscr{G}(z)} \leq 1$, then $\|u\|_{\mathscr{G}(x)} \leq \frac{1}{(1 - \|z - x\|_{\mathscr{G}(x)})} \leq 2$, and

$$\begin{aligned}
\|\nabla f(z)\|_{\mathscr{G}(z)^{-1}} = \|\sigma\|_{\mathscr{G}(z)^{-1}} &= \sup_{u:\ \|u\|_{\mathscr{G}(z)} \leq 1} \langle \sigma, u \rangle \\
&\leq \sup_{u:\ \|u\|_{\mathscr{G}(x)} \leq 2} \langle \sigma, u \rangle \\
&= 2 \cdot \|\sigma\|_{\mathscr{G}(x)^{-1}} = 2 \cdot \|\nabla f(x)\|_{\mathscr{G}(x)^{-1}} \ .
\end{aligned}$$

Therefore, for any $x \in \mathcal{K}_s$,

$$\sup_{z:\ \|z - x\|_{\mathscr{G}(x)} \leq \frac{1}{2}} \|\nabla f(z)\|_{\mathscr{G}(z)^{-1}} \leq 2 \cdot 5d \cdot \log\left(\frac{1}{s}\right) = 10d \cdot \log\left(\frac{1}{s}\right) \ .$$

The property $\mathrm{D}^2\mathscr{G}(x)[v, v] \succeq \mathbf{0}$ for all $x \in \mathrm{int}(\mathcal{K})$ and $v \in \mathbb{R}^d$ implies that $\mathscr{G}$ is 0-lower trace self-concordant, due to the following calculation.

$$\mathrm{trace}(\mathscr{G}(x)^{-1/2}\mathrm{D}^2\mathscr{G}(x)[v, v]\mathscr{G}(x)^{-1/2}) = \sum_{i=1}^{d} \mathrm{D}^2\mathscr{G}(x)[v, v, \mathscr{G}(x)^{-1/2}e_i, \mathscr{G}(x)^{-1/2}e_i] \geq 0 \ .$$

For warmness parameter $M \geq 1$ and error tolerance $\delta \in (0, 1/2)$, we pick $s = \frac{\delta}{2M} \leq \frac{1}{4}$. Since $f(x) = \sigma^\top x$, it satisfies a $(0, \mathscr{G})$-curvature lower and upper bounds. The step size bound $b_{\mathsf{Exp}}(d, M, \delta)$ in the theorem exactly matches the bound arising from applying Lemma 16 with $\mathcal{S} \leftarrow \mathcal{K}_{\delta/2M}$ and $\mathsf{U}_{f,\mathcal{S}} = 10d \cdot \log(2M/\delta)$, and we have for any $x \in \mathcal{S}$ that

$$d_{\mathrm{TV}}(\mathcal{T}_x, \mathcal{P}_x) \leq \frac{1}{16} \ .$$

We use Lemma 14 with $\mathcal{S} \leftarrow \mathcal{K}_{\delta/2M}$ and $\mathsf{U}_{f,\mathcal{S}} \leftarrow 10d \cdot \log(2M/\delta)$ to obtain for any $x, y \in \mathcal{S}$ such that $\|x - y\|_{\mathscr{G}(y)} \leq \frac{\sqrt{h}}{10}$,

$$d_{\mathrm{TV}}(\mathcal{P}_x, \mathcal{P}_y) \leq \frac{1}{2} \cdot \sqrt{\frac{h \cdot d}{20} + \frac{1}{80} + \frac{9h^2 \cdot d^2 \cdot \log^2(2M/\delta)}{2}} \ .$$

The bound $b_{\mathsf{Exp}}(d, M, \delta)$ ensures that $h \cdot d \leq h \cdot d^2 \leq \frac{1}{2000 \cdot \log^2(2M/\delta)}$ and therefore

$$d_{\mathrm{TV}}(\mathcal{P}_x, \mathcal{P}_y) \leq \frac{1}{2} \cdot \sqrt{\frac{h \cdot d}{20} + \frac{1}{80} + \frac{9h \cdot d^2}{2}} \leq \frac{1}{2} \cdot \sqrt{\frac{1}{400} + \frac{1}{80} + \frac{9}{40}} \leq \frac{1}{8} \ .$$

Therefore, for any $x, y \in \mathcal{K}_{\delta/2M}$ such that $\|x - y\|_{\mathscr{G}(y)} \leq \frac{\sqrt{h}}{10}$,

$$\mathrm{d}_{\mathrm{TV}}(\mathcal{T}_x, \mathcal{T}_y) \leq \mathrm{d}_{\mathrm{TV}}(\mathcal{T}_x, \mathcal{P}_x) + \mathrm{d}_{\mathrm{TV}}(\mathcal{P}_x, \mathcal{P}_y) + \mathrm{d}_{\mathrm{TV}}(\mathcal{T}_y, \mathcal{P}_y) \leq \frac{1}{16} + \frac{1}{8} + \frac{1}{16} = \frac{1}{4} .$$

With these results, we can obtain a lower bound on the $s$-conductance of $\mathbf{T}$ for $s = \frac{\delta}{2M}$ as indicated previously. Using Lemma 13 with $\Delta \leftarrow \frac{\sqrt{h}}{10}$ and $\mathcal{S} \leftarrow \mathcal{K}_s$, we have for any $A \subseteq \mathcal{K}$

$$\int_A \mathcal{T}_x(\mathcal{K} \setminus A)\pi(x)\,\mathrm{d}x \geq C \cdot \sqrt{h} \cdot \min\left\{1, \frac{1}{\sqrt{\nu}}\right\} \cdot \min\{\Pi(A \cap \mathcal{K}_s), \Pi((\mathcal{K} \setminus A) \cap \mathcal{K}_s)\} \quad (4)$$

for a universal constant $C$. Note that for any subset $B$ of $\mathcal{K}$, it holds that $\Pi(B) = \Pi(B \cap \mathcal{K}_s) + \Pi(B \cap (\mathcal{K} \setminus \mathcal{K}_s))$ as $\Pi$ is supported on $\mathcal{K}$. Since $\Pi(\mathcal{K}_s) \geq 1 - s$, we get

$$\Pi(B \cap \mathcal{K}_s) = \Pi(B) - \Pi(B \cap (\mathcal{K} \setminus \mathcal{K}_s)) \geq \Pi(B) - \Pi(\mathcal{K} \setminus \mathcal{K}_s) \geq \Pi(B) - s .$$

We apply this fact for $B \leftarrow A$ and $B \leftarrow (\mathcal{K} \setminus A)$ for $A \subseteq \mathcal{K}$, which gives

$$\Pi(A \cap \mathcal{K}_s) \geq \Pi(A) - s, \quad \Pi((\mathcal{K} \setminus A) \cap \mathcal{K}_s) \geq \Pi(\mathcal{K} \setminus A) - s .$$

Substituting these inequalities in Equation (4), we get for any $A \subset \mathcal{K}$ satisfying $\Pi(A) \in (s, 1 - s)$ that

$$\int_A \mathcal{T}_x(\mathcal{K} \cap A)\pi(x)\,\mathrm{d}x \geq C \cdot \sqrt{h} \cdot \min\left\{1, \frac{1}{\sqrt{\nu}}\right\} \cdot \min\left\{\Pi(A) - s, \Pi(\mathcal{K} \setminus A) - s\right\} ,$$

which implies by definition that

$$\Phi_{\mathbf{T}}^s \geq C \cdot \sqrt{h} \cdot \min\left\{1, \frac{1}{\sqrt{\nu}}\right\} , \quad s = \frac{\delta}{2M} . \quad (5)$$

Finally, we call Proposition 12 which states that for $\Pi_0 \in \mathcal{C} := \mathsf{Warm}(L_\infty, M, \Pi)$,

$$\mathrm{d}_{\mathrm{TV}}(\mathbb{T}_{\mathbf{T}}^k \Pi_0, \Pi) \leq M \cdot s + M \cdot \left(1 - \frac{\Phi_{\mathbf{T}}^{s\,2}}{2}\right)^k \leq \frac{\delta}{2} + M \cdot \exp\left(-\frac{k \cdot \Phi_{\mathbf{T}}^{s\,2}}{2}\right) .$$

Setting the second term on the right hand side to be less than $\frac{\delta}{2}$, and using Equation (5), we obtain

$$\tau_{\mathrm{mix}}(\delta; \mathbf{T}, \mathcal{C}) = \frac{2}{\Phi_{\mathbf{T}}^{s\,2}} \cdot \log\left(\frac{2M}{\delta}\right) \leq \frac{C'''}{h} \cdot \max\{1, \nu\} \cdot \log\left(\frac{M}{\delta}\right)$$

for some universal positive constant $C'''$. ∎

### D.3. Proving the key lemmas in Section D.1

D.3.1. PROOF OF LEMMA 13

Lemma 13 is derived from two isoperimetric inequalities. The first inequality is a consequence of log-concavity of $\Pi$ and $\nu$-symmetry of the metric $\mathscr{G}$, and the other holds when the potential $f$ of $\Pi$ satisfies $(\mu, \mathscr{G})$-curvature lower bound for a self-concordant metric $\mathscr{G}$.

**Lemma 17** *Let $\Pi$ be a log-concave distribution whose support is $\mathcal{K}$, and consider a metric $\mathscr{G}$ : $\mathrm{int}(\mathcal{K}) \to \mathbb{S}_+^d$ that is $\nu$-symmetric. Let $\Pi_\mathcal{A}$ denote the restriction of $\Pi$ to $\mathcal{A} \subseteq \mathcal{K}$. For any partition $\{S_1, S_2, S_3\}$ of a convex subset $\mathcal{S}$ of $\mathcal{K}$, we have*

$$\Pi_\mathcal{S}(S_3) \geq \frac{1}{\sqrt{\nu}} \cdot \inf_{y \in S_2, x \in S_1} \|x - y\|_{\mathscr{G}(y)} \cdot \Pi_\mathcal{S}(S_2) \cdot \Pi_\mathcal{S}(S_1) \ .$$

For the following lemma, we introduce some additional notation. The geodesic distance between $x, y \in \mathrm{int}(\mathcal{K})$ with respect to the metric $\mathscr{G}$ is denoted by $d_\mathscr{G}(x, y)$.

**Lemma 18** *Consider a log-concave distribution $\Pi$ whose support $\mathcal{K}$ and a metric $\mathscr{G}$ : $\mathrm{int}(\mathcal{K}) \to \mathbb{S}_+^d$. If $\mathscr{G}$ is self-concordant and the potential $f$ of $\Pi$ satisfies $(\mu, \mathscr{G})$-curvature lower bound, then for any partition $\{S_1, S_2, S_3\}$ of $\mathcal{K}$,*

$$\Pi(S_3) \geq \frac{\mu}{8 + 4\sqrt{\mu}} \cdot \inf_{y \in S_2, x \in S_1} d_\mathscr{G}(x, y) \cdot \min\{\Pi(S_2), \Pi(S_1)\} \ .$$

Lemma 17 is the combination of a isoperimetric inequality for log-concave distributions (Lovász and Vempala, 2003, Thm. 2.2) and $\nu$-symmetry (Laddha et al., 2020, Lem. 2.3). Lemma 18 is a new isoperimetric inequality, and is a generalisation of the result in Gopi et al. (2023, Lem. 9) and Kook and Vempala (2024, Lem. B.7). These prior results assume $\mu$-relative convexity with respect to a self-concordant function $\psi$ whose Hessian is approximately the metric $\mathscr{G}$. It is interesting to note that when $\mu$ is small, $\frac{\mu}{8+4\sqrt{\mu}}$ scales as $\mu$, whereas when $\mu$ is large, this ratio scales as $\sqrt{\mu}$ instead. To work with Lemma 18, we require relating $d_\mathscr{G}(x, y)$ to $\|x - y\|_\mathscr{G}$, and this is possible when $d_\mathscr{G}(x, y)$ is sufficiently small as given in the following lemma.

**Lemma 19 (Nesterov and Todd (2002, Lem. 3.1))** *Let the metric $\mathscr{G}$ : $\mathrm{int}(\mathcal{K}) \to \mathbb{S}_+^d$ be self-concordant. For $x, y \in \mathrm{int}(\mathcal{K})$ and $\kappa \in [0, 1)$, if $d_\mathscr{G}(x, y) \leq \kappa - \frac{\kappa^2}{2}$, then $\|x - y\|_{\mathscr{G}(x)} \leq \kappa < 1$.*

Equipped with the above lemmas that we will prove later, we now state the proof of Lemma 13.

**Proof** Our proof follows the same structure as prior results based on the one-step overlap. In particular, here we follow Dwivedi et al. (2018, Proof of Lem. 2). Due to the reversibility of $\mathbf{T}$,

$$\int_{A_1} \mathcal{T}_x(A_2)\pi(x) \, \mathrm{d}x = \int_{A_2} \mathcal{T}_y(A_1)\pi(y) \, \mathrm{d}y \ .$$

Define the sets

$$A_1' \stackrel{\text{def}}{=} \left\{ x \in A_1 \cap \mathcal{S} \ : \ \mathcal{T}_x(A_2) < \frac{3}{8} \right\} \qquad A_2' \stackrel{\text{def}}{=} \left\{ y \in A_2 \cap \mathcal{S} \ : \ \mathcal{T}_y(A_1) < \frac{3}{8} \right\} \ .$$

Using the definitions above, and the fact that $A_i \cap (\mathcal{S} \setminus A'_i) \subseteq A_i$,

$$
\begin{aligned}
\int_{A_1} \mathcal{T}_x(A_2)\pi(x)\,\mathrm{d}x &= \frac{1}{2}\left(\int_{A_1} \mathcal{T}_x(A_2)\pi(x)\,\mathrm{d}x + \int_{A_2} \mathcal{T}_y(A_1)\pi(y)\,\mathrm{d}y\right) \\
&\geq \frac{1}{2}\left(\int_{A_1\cap(\mathcal{S}\setminus A'_1)} \mathcal{T}_x(A_2)\pi(x)\,\mathrm{d}x + \int_{A_2\cap(\mathcal{S}\setminus A'_2)} \mathcal{T}_y(A_1)\pi(y)\,\mathrm{d}y\right) \\
&\geq \frac{3}{16}\cdot\left(\int_{A_1\cap(\mathcal{S}\setminus A'_1)} \pi(x)\,\mathrm{d}x + \int_{A_2\cap(\mathcal{S}\setminus A'_2)} \pi(y)\,\mathrm{d}y\right) \\
&= \frac{3}{16}\cdot(\Pi(A_1\cap(\mathcal{S}\setminus A'_1)) + \Pi(A_2\cap(\mathcal{S}\setminus A'_2))) .
\end{aligned}
\tag{6}
$$

If $\Pi(A'_i) \leq \frac{\Pi(A_i\cap\mathcal{S})}{2}$ for given $i \in \{1,2\}$,

$$
\Pi(A_i\cap(\mathcal{S}\setminus A'_1)) \geq \Pi(A_i\cap\mathcal{S}) - \Pi(A'_i) \geq \frac{1}{2}\cdot\Pi(A_i\cap\mathcal{S}) .
$$

Hence, if $\Pi(A'_1) \leq \frac{\Pi(A_1\cap\mathcal{S})}{2}$ or $\Pi(A'_2) \leq \frac{\Pi(A_2\cap\mathcal{S})}{2}$, continuing from Equation (6),

$$
\int_{A_1} \mathcal{T}_x(A_2)\pi(x)\,\mathrm{d}x \geq \frac{3}{16}\cdot\min\{\Pi(A_1\cap\mathcal{S}),\Pi(A_2\cap\mathcal{S})\} .
\tag{7}
$$

An alternative lower bound continuing from Equation (6) is

$$
\begin{aligned}
\int_{A_1} \mathcal{T}_x(A_2)\pi(x)\,\mathrm{d}x &\geq \frac{3}{16}\cdot(\Pi(A_1\cap(\mathcal{S}\setminus A'_1)) + \Pi(A_2\cap(\mathcal{S}\setminus A'_2))) \\
&= \frac{3}{16}\cdot\Pi(\mathcal{S}\setminus A'_1\setminus A'_2) \\
&= \frac{3}{16}\cdot\Pi(\mathcal{S})\cdot\Pi_{\mathcal{S}}(\mathcal{S}\setminus A'_1\setminus A'_2)
\end{aligned}
\tag{8}
$$

where recall that $\Pi_{\mathcal{S}}$ is the restriction of $\Pi$ to $\mathcal{S}$ with density $\pi_{\mathcal{S}}(x) = \frac{e^{-f_{|\mathcal{S}}(x)}}{\Pi(\mathcal{S})}$. By definition of $A'_1$ and $A'_2$, for any $x' \in A'_1$ and $y' \in A'_2$,

$$
\mathrm{d}_{\mathrm{TV}}(\mathcal{T}_{x'},\mathcal{T}_{y'}) \geq \mathcal{T}_{x'}(A_1) - \mathcal{T}_{y'}(A_1) = 1 - \mathcal{T}_{x'}(A_2) - \mathcal{T}_{y'}(A_1) > 1 - \frac{3}{8} - \frac{3}{8} = \frac{1}{4} .
$$

By the one-step overlap assumed in the statement, we have that $\|x' - y'\|_{\mathscr{G}(y')} > \Delta$ for $\Delta \leq \frac{1}{2}$. The remainder of the proof deals with the case where $\Pi(A'_i) \geq \frac{\Pi(A_i\cap\mathcal{S})}{2}$ for both $i = 1,2$.

**When $\mathscr{G}$ is $\nu$-symmetric**  Since $x' \in A'_1$ and $y' \in A'_2$ are arbitrary, we have

$$
\inf_{x'\in A'_1, y'\in A'_2} \|x' - y'\|_{\mathscr{G}(y')} \geq \Delta .
$$

Now, we use Lemma 17 for the partition $\{A'_1, A'_2, \mathcal{S} \setminus A'_1 \setminus A'_2\}$ of $\mathcal{S}$.

$$\begin{aligned}
\Pi_{\mathcal{S}}(\mathcal{S} \setminus A'_1 \setminus A'_2) &\geq \frac{\Delta}{\sqrt{\nu}} \cdot \Pi_{\mathcal{S}}(A'_1) \cdot \Pi_{\mathcal{S}}(A'_2) \\
&\geq \frac{\Delta}{\sqrt{\nu}} \cdot \frac{\Pi(A'_1)}{\Pi(\mathcal{S})} \cdot \frac{\Pi(A'_2)}{\Pi(\mathcal{S})} \\
&\geq \frac{\Delta}{4\sqrt{\nu}} \cdot \frac{\Pi(A_1 \cap \mathcal{S})}{\Pi(\mathcal{S})} \cdot \frac{\Pi(A_2 \cap \mathcal{S})}{\Pi(\mathcal{S})} ,
\end{aligned}$$

where the last inequality is due to the assumption that $\Pi(A'_i) \geq \frac{\Pi(A_i \cap \mathcal{S})}{2}$ for $i \in \{1, 2\}$. Since $A_1$ and $A_2$ form a partition of $\mathcal{K}$,

$$\Pi(A_1 \cap \mathcal{S}) + \Pi(A_2 \cap \mathcal{S}) = \Pi(\mathcal{S}) \Leftrightarrow \frac{\Pi(A_2 \cap \mathcal{S})}{\Pi(\mathcal{S})} = 1 - \frac{\Pi(A_1 \cap \mathcal{S})}{\Pi(\mathcal{S})} .$$

Using the algebraic fact that $t(1 - t) \geq \frac{1}{2} \min\{t, 1 - t\}$ for $t \in [0, 1]$, we obtain the lower bound

$$\Pi_{\mathcal{S}}(\mathcal{S} \setminus A'_1 \setminus A'_2) \geq \frac{\Delta}{8\sqrt{\nu}} \cdot \frac{\min\{\Pi(A_1 \cap \mathcal{S}), \Pi(A_2 \cap \mathcal{S})\}}{\Pi(\mathcal{S})} . \tag{9}$$

**When $f$ satisfies $(\mu, \mathscr{G})$-curvature lower bound**   Since $\Delta \leq \frac{1}{2}$ and $\|x' - y'\|_{\mathscr{G}(y')} \geq \Delta$, we get from Lemma 19 that

$$d_{\mathscr{G}}(y', x') > \Delta - \frac{\Delta^2}{2} \Rightarrow \inf_{y' \in A'_2, x' \in A'_1} d_{\mathscr{G}}(y', x') > \Delta - \frac{\Delta^2}{2} \geq \frac{\Delta}{2}$$

where the last inequality uses the algebraic fact that $t - \frac{t^2}{2} \geq \frac{t}{2}$ for $t \in [0, 1]$.

By definition of $\Pi_{\mathcal{S}}$, its potential $f_{|\mathcal{S}}$ satisfies $(\mu, \mathscr{G})$-curvature lower bound over $\mathcal{S}$. This permits using Lemma 18 for $\Pi_{\mathcal{S}}$ for the partition $\{A'_1, A'_2, \mathcal{S} \setminus A'_1 \setminus A'_2\}$, which results in

$$\begin{aligned}
\Pi_{\mathcal{S}}(\mathcal{S} \setminus A'_1 \setminus A'_2) &\geq \frac{\mu}{8 + 4\sqrt{\mu}} \cdot \inf_{y' \in A_{2'}, x' \in A'_1} d_{\mathscr{G}}(y', x') \cdot \min\{\Pi_{\mathcal{S}}(A'_1), \Pi_{\mathcal{S}}(A'_2)\} \\
&\geq \frac{\mu}{8 + 4\sqrt{\mu}} \cdot \frac{\Delta}{2} \cdot \min\{\Pi_{\mathcal{S}}(A'_1), \Pi_{\mathcal{S}}(A'_2)\} \\
&= \frac{\mu}{8 + 4\sqrt{\mu}} \cdot \frac{\Delta}{2} \cdot \frac{\min\{\Pi(A'_1), \Pi(A'_2)\}}{\Pi(\mathcal{S})} \\
&\geq \frac{\mu}{8 + 4\sqrt{\mu}} \cdot \frac{\Delta}{4} \cdot \frac{\min\{\Pi(A_1 \cap \mathcal{S}), \Pi(A_2 \cap \mathcal{S})\}}{\Pi(\mathcal{S})} ,
\end{aligned} \tag{10}$$

where the last inequality is due to the assumption that $\Pi(A'_i) \geq \frac{\Pi(A_i \cap \mathcal{S})}{2}$ for $i \in \{1, 2\}$.

Combining Equation (10) and Equation (9) in Equation (8), we get the net lower bound in conjunction with Equation (7)

$$\int_{A_1} \mathcal{T}_x(A_2) \pi(x) \, dx \geq \frac{3\Delta}{16} \cdot \min\left\{1, \max\left\{\frac{\widetilde{\mu}}{4}, \frac{1}{8\sqrt{\nu}}\right\}\right\} \cdot \min\{\Pi(A_1 \cap \mathcal{S}), \Pi(A_2 \cap \mathcal{S})\} . \tag{11}$$

$\blacksquare$

### D.3.2. PROOF OF LEMMA 14

**Proof** The KL divergence between two distribution $\rho_1$ and $\rho_2$ whre $\rho_1$ is absolutely continuous with respect to $\rho_2$ is defined as

$$\mathrm{d_{KL}}(\rho_1\|\rho_2) = \int \mathrm{d}\rho_1(x) \log \frac{\mathrm{d}\rho_1(x)}{\mathrm{d}\rho_2(x)} \,.$$

Pinsker's inequality yields a bound on the TV distance as

$$\mathrm{d_{TV}}(\mathcal{N}(\mu_1,\,\Sigma_1),\mathcal{N}(\mu_2,\,\Sigma_2)) \leq \sqrt{\frac{1}{2} \cdot \mathrm{d_{KL}}(\mathcal{N}(\mu_1,\,\Sigma_1) \,\|\, \mathcal{N}(\mu_2,\,\Sigma_2))} \,.$$

Hence, to obtain a bound on $\mathrm{d_{TV}}(\mathcal{P}_y, \mathcal{P}_x)$, it suffices to bound $\mathrm{d_{KL}}(\mathcal{P}_y\|\mathcal{P}_x)$.

When $\rho_1, \rho_2$ are Gaussian distributions, the KL divergence has a closed form that is stated below.

$$\mathrm{d_{KL}}(\mathcal{N}(\mu_1,\,\Sigma_1) \,\|\, \mathcal{N}(\mu_2,\,\Sigma_2))$$
$$= \frac{1}{2}\left(\mathrm{trace}(\Sigma_1\Sigma_2^{-1} - \mathrm{I}_{d\times d}) - \log\det \Sigma_1\Sigma_2^{-1} + \|\mu_2 - \mu_1\|_{\Sigma_2^{-1}}^2\right) \,.$$

Since $\mathcal{P}_y$ and $\mathcal{P}_x$ are Gaussian distributions, we can obtain $\mathrm{d_{KL}}(\mathcal{P}_y \,\|\, \mathcal{P}_x)$ by substituting

$$\mu_1 \leftarrow y - h \cdot \mathscr{G}(y)^{-1}\nabla f(y)\,, \quad \mu_2 \leftarrow x - h \cdot \mathscr{G}(x)^{-1}\nabla f(x)\,, \quad \Sigma_1 \leftarrow 2h \cdot \mathscr{G}(y)\,, \quad \Sigma_2 \leftarrow 2h \cdot \mathscr{G}(x)^{-1}\,,$$

in the above formula, which gives

$$\mathrm{d_{KL}}(\mathcal{P}_y \,\|\, \mathcal{P}_x) = \frac{1}{2} \cdot \left(\mathrm{trace}(\mathscr{G}(x)\mathscr{G}(y)^{-1} - \mathrm{I}_{d\times d}) - \log\det \mathscr{G}(x)\mathscr{G}(y)^{-1}\right)$$
$$+ \frac{1}{4h} \cdot \left\|(x - h \cdot \mathscr{G}(x)^{-1}\nabla f(x)) - (y - h \cdot \mathscr{G}(y)^{-1}\nabla f(y))\right\|_{\mathscr{G}(x)}^2$$
$$= \frac{1}{2} \cdot \underbrace{\left(\mathrm{trace}(\mathscr{G}(y)^{-1/2}\mathscr{G}(x)\mathscr{G}(y)^{-1/2} - \mathrm{I}_{d\times d}) - \log\det \mathscr{G}(y)^{-1/2}\mathscr{G}(x)\mathscr{G}(y)^{-1/2}\right)}_{T_1^P}$$
$$+ \frac{1}{4h} \cdot \underbrace{\left\|(x - h \cdot \mathscr{G}(x)^{-1}\nabla f(x)) - (y - h \cdot \mathscr{G}(y)^{-1}\nabla f(y))\right\|_{\mathscr{G}(x)}^2}_{T_2^P} \,.$$

In the lemma, the metric $\mathscr{G}$ is assumed to be self-concordant, and $x, y$ are such that $\|x - y\|_{\mathscr{G}(y)} \leq \frac{\sqrt{h}}{10} < 1$. Hence, from Lemma 26(2) we have

$$(1 - \|x - y\|_{\mathscr{G}(y)})^2 \cdot \mathscr{G}(y) \preceq \mathscr{G}(x) \preceq \frac{1}{(1 - \|x - y\|_{\mathscr{G}(y)})^2} \cdot \mathscr{G}(y) \tag{12a}$$

$$\Leftrightarrow (1 - \|x - y\|_{\mathscr{G}(y)})^2 \cdot \mathrm{I}_{d\times d} \preceq \mathscr{G}(y)^{-1/2}\mathscr{G}(x)\mathscr{G}(y)^{-1/2} \preceq \frac{1}{(1 - \|x - y\|_{\mathscr{G}(y)})^2} \cdot \mathrm{I}_{d\times d} \,. \tag{12b}$$

For convenience, $M$ to denote $\mathscr{G}(y)^{-1/2}\mathscr{G}(x)\mathscr{G}(y)^{-1/2}$, and $r_y$ to denote $\|x-y\|_{\mathscr{G}(y)}$. Equation (12b) asserts that all eigenvalues of $M$ lie in the range $[(1-r_y)^2, (1-r_y)^{-2}]$. Lemmas 27 and 28 imply that

$$\forall\, i \in [d], \quad \lambda_i(M) - 1 - \log \lambda_i(M) \leq \frac{((1-r_y)^2 - 1)^2}{(1-r_y)^2} \leq \frac{4r_y^2}{(1-r_y)^2}\,.$$

This leads to a bound for $T_1^P$ as follows when $h \leq 1$.

$$\begin{aligned}
T_1^P = \operatorname{trace}(M - I) - \log \det M &= \sum_{i=1}^{d} (\lambda_i(M) - 1 - \log \lambda_i(M)) \\
&\leq d \cdot \max_{i \in [d]} (\lambda_i(M) - 1 - \log \lambda_i(M)) \\
&\leq d \cdot \frac{4r_y^2}{(1-r_y)^2} \\
&\leq d \cdot \frac{4 \cdot \frac{h}{100}}{(1 - \frac{\sqrt{h}}{10})^2} \leq \frac{h \cdot d}{20}\,.
\end{aligned}$$

Using Equation (12a) and the fact that $x, y \in \mathcal{S}$, we also obtain a bound for $T_2^P$ as shown below.

$$\begin{aligned}
T_2^P = \|(x - h \cdot \mathscr{G}(x)^{-1}\nabla f(x)) - (y - h \cdot \mathscr{G}(y)^{-1}\nabla f(y))\|_{\mathscr{G}(x)}^2 & \\
\leq 2 \cdot \|x - y\|_{\mathscr{G}(x)}^2 + 2h^2 \cdot \|\mathscr{G}(y)^{-1}\nabla f(y) - \mathscr{G}(x)^{-1}\nabla f(x)\|_{\mathscr{G}(x)}^2 & \\
\leq 2 \cdot \|x - y\|_{\mathscr{G}(x)}^2 + 4h^2 \cdot \|\mathscr{G}(x)^{-1}\nabla f(x)\|_{\mathscr{G}(x)}^2 + 4h^2 \cdot \|\mathscr{G}(y)^{-1}\nabla f(y)\|_{\mathscr{G}(x)}^2 & \\
\leq \frac{2 \cdot \|x - y\|_{\mathscr{G}(y)}^2}{(1-r_y)^2} + 4h^2 \cdot \|\nabla f(x)\|_{\mathscr{G}(x)^{-1}}^2 + 4h^2 \cdot \frac{\|\nabla f(y)\|_{\mathscr{G}(y)^{-1}}^2}{(1-r_y)^2} & \\
\leq 2 \cdot \frac{r_y^2}{(1-r_y)^2} + 4h^2 \cdot \mathsf{U}_{f,\mathcal{S}}^2 + \frac{4h^2 \cdot \mathsf{U}_{f,\mathcal{S}}^2}{(1-r_y)^2}\,. &
\end{aligned}$$

Since $r_y \leq \frac{\sqrt{h}}{10}$ for $h \leq 1$, $\frac{1}{(1-r_y)^2} \leq \frac{5}{4}$. Hence, we have the net bound

$$d_{\mathrm{KL}}(\mathcal{P}_y \parallel \mathcal{P}_x) = \frac{1}{2} \cdot T_1^P + \frac{1}{4h} \cdot T_2^P = \frac{1}{2} \cdot \left( \frac{h \cdot d}{20} + \frac{1}{80} + \frac{9h \cdot \mathsf{U}_{f,\mathcal{S}}^2}{2} \right)$$

which results in

$$d_{\mathrm{TV}}(\mathcal{P}_x, \mathcal{P}_y) \leq \frac{1}{2}\sqrt{\frac{h \cdot d}{20} + \frac{1}{80} + \frac{9h \cdot \mathsf{U}_{f,\mathcal{S}}^2}{2}}\,.$$

∎

### D.3.3. PROOFS OF LEMMAS 15 AND 16

First, we state a collection of facts about metrics that satisfy various notions of self-concordance defined in Section 4.1. These will come in handy in proving Lemmas 15 and 16.

**Lemma 20** *Let the metric $\mathscr{G}$ be self-concordant. For any $x, y \in \mathrm{int}(\mathcal{K})$ such that $\|y - x\|_{\mathscr{G}(x)} \leq \frac{3}{10}$,*

$$\log \det \mathscr{G}(y) - \log \det \mathscr{G}(x) \geq -3d \cdot \|y - x\|_{\mathscr{G}(x)} \, ,$$
$$\|y - x\|_{\mathscr{G}(x)}^2 - \|y - x\|_{\mathscr{G}(y)}^2 \geq -6 \cdot \|y - x\|_{\mathscr{G}(x)}^3 \, .$$

**Lemma 21** *Let the metric $\mathscr{G}$ be strongly self-concordant. Then, for any $x \in \mathrm{int}(\mathcal{K})$,*

$$\|\mathscr{G}(x)^{-1/2} \nabla \log \det \mathscr{G}(x)\| \leq 2\sqrt{d} \, .$$

*Additionally, if $\mathscr{G}$ is also $\alpha$-lower trace self-concordant, then for any $y \in \mathcal{E}_x^{\mathscr{G}}(1)$,*

$$\log \det \mathscr{G}(y)\mathscr{G}(x)^{-1} \geq \langle \nabla \log \det \mathscr{G}(x), y - x \rangle - \frac{17}{8} \cdot (\alpha + 4) \cdot \|y - x\|_{\mathscr{G}(x)}^2 \, .$$

**Lemma 22** *Let the metric $\mathscr{G}$ be self-concordant, and $x \in \mathrm{int}(\mathcal{K})$. Consider $y, w \in \mathcal{E}_x^{\mathscr{G}}(1)$, and define $\Delta(u; x) = \|u - x\|_{\mathscr{G}(u)}^2 - \|u - x\|_{\mathscr{G}(x)}^2$. There exists $t^\star \in (0, 1)$ such that*

$$\Delta(w; x) - \Delta(y; x) \leq 6 \cdot \frac{\|y + t^\star(w - y) - x\|_{\mathscr{G}(x)}^2 \cdot \|w - y\|_{\mathscr{G}(x)}}{(1 - \|y + t^\star(w - y) - x\|_{\mathscr{G}(x)})^3} \, .$$

The proofs of Lemmas 15 and 16 follow a similar structure, and we will state them as one to avoid redundancy, and highlight the point at which they branch out.

**Proof** We adopt the shorthand notation $R_{x \to z}$ for the ratio $\frac{\pi(z)p_z(x)}{\pi(x)p_x(z)}$ for the convenience. The transition distribution $\mathcal{T}_x$ has an atom at $x$ and satisfies

$$\mathcal{T}_x(\{x\}) = 1 - \mathbb{E}_{z \sim \mathcal{P}_x}\left[\min\left\{1, R_{x \to z} \cdot \mathbf{1}\{z \in \mathcal{K}\}\right\}\right] \, .$$

Consequently, the TV distance $\mathrm{d}_{\mathrm{TV}}(\mathcal{T}_x, \mathcal{P}_x)$ for any $x \in \mathrm{int}(\mathcal{K})$ can be given as

$$\mathrm{d}_{\mathrm{TV}}(\mathcal{T}_x, \mathcal{P}_x) = \frac{1}{2}\left(\mathcal{T}_x(\{x\}) + \int_{\mathbb{R}^d \setminus \{x\}} (1 - \min\{1, R_{x \to z} \cdot \mathbf{1}\{z \in \mathcal{K}\}\}) \, p_x(z) \, \mathrm{d}z\right)$$
$$= 1 - \underbrace{\mathbb{E}_{z \sim \mathcal{P}_x}\left[\min\left\{1, R_{x \to z} \cdot \mathbf{1}\{z \in \mathcal{K}\}\right\}\right]}_{\mathcal{Q}} \, . \tag{13}$$

Thus, a lower bound for $\mathcal{Q}$ implies an upper bound for $\mathrm{d}_{\mathrm{TV}}(\mathcal{T}_x, \mathcal{P}_x)$. One strategy to obtain a lower bound is using Markov inequality; for any $\tau \in (0, 1)$,

$$\mathbb{E}_{z \sim \mathcal{P}_x}\left[\min\left\{1, R_{x \to z} \cdot \mathbf{1}\{z \in \mathcal{K}\}\right\}\right] \geq \tau \cdot \mathbb{P}_{z \sim \mathcal{P}_x}\left(R_{x \to z} \cdot \mathbf{1}\{z \in \mathcal{K}\} \geq \tau\right)$$
$$\geq \tau \cdot \mathbb{P}_{z \sim \mathcal{P}_x}\left(R_{x \to z} \cdot \mathbf{1}\{z \in \mathcal{K}\} \geq \tau \mid \mathfrak{E}\right) \cdot \mathbb{P}_{z \sim \mathcal{P}_x}(\mathfrak{E}) \, . \tag{14}$$

Therefore, it suffices to identify an event $\mathfrak{E}$ that satisfies three key desiderata: (1) implies that $z \in \mathcal{K}$, (2) yields a lower bound $\tau$ that is bounded away from 0, and (3) occurs with high probability. From the definition of $\mathcal{P}_x$, any $z \sim \mathcal{P}_x$ is distributionally equivalent to the random vector

$$x - h \cdot \mathscr{G}(x)^{-1} \nabla f(x) + \sqrt{2h} \cdot \mathscr{G}(x)^{-1/2} \gamma, \quad \gamma \sim \mathcal{N}(0, \mathrm{I}_{d \times d}) \, .$$

Due to this, we can view $z$ as a function of $\gamma$, and henceforth we consider the underlying random variable to be $\gamma$, and the event $\mathfrak{E}$ is defined in terms of this $\gamma$. Before defining $\mathfrak{E}$, we first provide a simplified lower bound for $R_{x \to z}$ using certain properties of the potential $f$ assumed in the lemmas.

$$
\log R_{x \to z} = \overbrace{f(x) - f(z)}^{T_F} + \frac{1}{2} \log \det \mathscr{G}(z) \mathscr{G}(x)^{-1}
$$
$$
+ \underbrace{\frac{\|z - x + h \cdot \mathscr{G}(x)^{-1} \nabla f(x)\|_{\mathscr{G}(x)}^2 - \|x - z + h \cdot \mathscr{G}(z)^{-1} \nabla f(z)\|_{\mathscr{G}(z)}^2}{4h}}_{T_D} .
$$

First, we work with $T_F$. We have $z \in \mathcal{K}$, otherwise these calculations would be irrelevant. Let $x_t = x + t \cdot (z - x)$ for $t \in [0, 1]$.

$$
\begin{aligned}
T_F :&= f(x) - f(z) \\
&= \frac{1}{2}(f(x) - f(z)) + \frac{1}{2}(f(x) - f(z)) \\
&\overset{(a)}{\geq} \frac{1}{2} \cdot \langle \nabla f(z), x - z \rangle - \frac{1}{2} \cdot \langle \nabla f(x), z - x \rangle - \frac{1}{4} \cdot \langle z - x, \nabla^2 f(x_{t^\star})(z - x) \rangle \\
&\overset{(b)}{\geq} \frac{1}{2} \cdot \langle \nabla f(z), x - z \rangle - \frac{1}{2} \cdot \langle \nabla f(x), z - x \rangle - \frac{\lambda}{4} \cdot \langle z - x, \mathscr{G}(x_{t^\star})(z - x) \rangle .
\end{aligned}
$$

Inequality $(a)$ uses the fact that $f$ is convex, and consider a second-order Taylor expansion of $f$ around $x$. Inequality $(b)$ uses the fact that $f$ satisfies $(\lambda, \mathscr{G})$-curvature upper bound. Next for $T_D$,

$$
\begin{aligned}
T_D :&= \frac{1}{4h} \cdot \left( \|z - x + h \cdot \mathscr{G}(x)^{-1} \nabla f(x)\|_{\mathscr{G}(x)}^2 - \|x - z + h \cdot \mathscr{G}(z)^{-1} \nabla f(z)\|_{\mathscr{G}(z)}^2 \right) \\
&= \frac{1}{4h} \cdot \left( \|z - x\|_{\mathscr{G}(x)}^2 - \|z - x\|_{\mathscr{G}(z)}^2 + 2h \cdot \langle z - x, \nabla f(x) \rangle + 2h \cdot \langle z - x, \nabla f(z) \rangle \right. \\
&\qquad \left. + h^2 \cdot \|\nabla f(x)\|_{\mathscr{G}(x)^{-1}}^2 - h^2 \cdot \|\nabla f(z)\|_{\mathscr{G}(z)^{-1}}^2 \right) \\
&\overset{(a)}{\geq} \frac{\|z - x\|_{\mathscr{G}(x)}^2 - \|z - x\|_{\mathscr{G}(z)}^2}{4h} - \frac{h}{4} \cdot \|\nabla f(z)\|_{\mathscr{G}(z)^{-1}}^2 \\
&\qquad + \frac{1}{2} \cdot \langle z - x, \nabla f(x) \rangle + \frac{1}{2} \cdot \langle z - x, \nabla f(z) \rangle .
\end{aligned}
$$

Inequality $(a)$ simply uses the fact that $\|\nabla f(x)\|_{\mathscr{G}(x)^{-1}}^2 \geq 0$. This results in the lower bound

$$
\begin{aligned}
\log R_{x \to z} \geq &-\frac{h}{4} \cdot \underbrace{\|\nabla f(z)\|_{\mathscr{G}(z)^{-1}}^2}_{T_0^A} - \frac{\lambda}{4} \cdot \underbrace{\|z - x\|_{\mathscr{G}(x_{t^\star})}^2}_{T_1^A} \\
&+ \frac{1}{2} \cdot \underbrace{\log \det \mathscr{G}(z) \mathscr{G}(x)^{-1}}_{T_2^A} + \frac{1}{4h} \cdot \underbrace{\left( \|z - x\|_{\mathscr{G}(x)}^2 - \|z - x\|_{\mathscr{G}(z)}^2 \right)}_{T_3^A} .
\end{aligned} \tag{15}
$$

For $\gamma \sim \mathcal{N}(\mathbf{0}, \mathrm{I}_{d \times d})$, define $\xi = x + \sqrt{2h} \cdot \mathscr{G}(x)^{-1/2} \gamma$. The proposal $z \sim \mathcal{P}_x$ is distributionally equivalent to $\xi - h \cdot \mathscr{G}(x)^{-1} \nabla f(x)$. For $\varepsilon > 0$, define

$$
\mathsf{N}_\varepsilon = 1 + 2 \log \frac{1}{\varepsilon} + 2 \sqrt{\log \frac{1}{\varepsilon}} \qquad \mathsf{I}_\varepsilon = \sqrt{2 \log \frac{1}{\varepsilon}} \tag{16}
$$

and the events

$$\mathfrak{E}_1 := \|\gamma\|^2 \leq d \cdot \mathsf{N}_\varepsilon \qquad\qquad \mathfrak{E}_2 := -\langle \gamma, \mathscr{G}(x)^{-1/2} \nabla f(x) \rangle \leq \mathsf{U}_{f,\mathcal{S}} \cdot \mathsf{I}_\varepsilon$$

$$\mathfrak{E}_3 := -\langle \gamma, \mathscr{G}(x)^{-1/2} \nabla \log \det \mathscr{G}(x) \rangle \leq 2\sqrt{d} \cdot \mathsf{I}_\varepsilon \quad \mathfrak{E}_4 := \|\xi - x\|^2_{\mathscr{G}(\xi)} - \|\xi - x\|^2_{\mathscr{G}(x)} \leq 4h \cdot \varepsilon$$

Let $\gamma \sim \mathcal{N}(\mathbf{0}, \mathrm{I}_{d \times d})$. Then, we have the following facts about $\mathfrak{E}_i$ for $i \in [4]$.

- By Lemma 24, $\mathbb{P}(\mathfrak{E}_1) \geq 1 - \varepsilon$.

- Since $x \in \mathcal{S}$ according to the statements of the lemmas,

$$\|\mathscr{G}(x)^{-1/2} \nabla f(x)\| = \|\nabla f(x)\|_{\mathscr{G}(x)^{-1}} \leq \mathsf{U}_{f,\mathcal{S}} \Rightarrow \mathbb{P}(\mathfrak{E}_2) \geq 1 - \varepsilon$$

  and the implication is due to Lemma 25.

- If $\mathscr{G}$ is strongly self-concordant (as assumed in Lemma 16), then from Lemma 21,

$$\|\mathscr{G}(x)^{-1/2} \nabla \log \det \mathscr{G}(x)\| \leq 2\sqrt{d} \Rightarrow \mathbb{P}(\mathfrak{E}_3) \geq 1 - \varepsilon$$

  where the implication again follows from Lemma 25.

- When $\mathscr{G}$ is average self-concordant and $h \leq \frac{r_\varepsilon^2}{2d}$, then by definition $\mathbb{P}(\mathfrak{E}_4) \geq 1 - \varepsilon$.

Conditioning on $\mathfrak{E}_1$ and $\mathfrak{E}_2$, we get

$$\begin{aligned}
\|z - x\|^2_{\mathscr{G}(x)} &= \|x - h \cdot \mathscr{G}(x)^{-1} \nabla f(x) + \sqrt{2h} \cdot \mathscr{G}(x)^{-1/2} \gamma - x\|^2_{\mathscr{G}(x)} \\
&= h^2 \cdot \|\nabla f(x)\|^2_{\mathscr{G}(x)^{-1}} - (2h)^{3/2} \cdot \langle \mathscr{G}(x)^{-1/2} \nabla f(x), \gamma \rangle + 2h \cdot \|\gamma\|^2 \\
&\leq h^2 \cdot \mathsf{U}^2_{f,\mathcal{S}} + (2h)^{3/2} \cdot \mathsf{U}_{f,\mathcal{S}} \cdot \mathsf{I}_\varepsilon + 2h \cdot d \cdot \mathsf{N}_\varepsilon .
\end{aligned} \tag{17}$$

When the step size $h$ satisfies the bound

$$h \leq \min \left\{ \frac{1}{6\mathsf{U}_{f,\mathcal{S}}}, \frac{3}{200\mathsf{N}_\varepsilon \cdot d}, \frac{1}{25\mathsf{I}_\varepsilon^{2/3} \mathsf{U}_{f,\mathcal{S}}^{2/3}} \right\}, \tag{$\mathsf{B}_0$}$$

we have $\|z - x\|_{\mathscr{G}(x)} \leq \frac{3}{10}$ which implies $z \in \mathcal{E}_x^{\mathscr{G}}(1)$. In the setting of both Lemmas 15 and 16, the metric $\mathscr{G}$ is self-concordant, and by Lemma 26(1). Moreover, this also shows that $x_t$ also satisfies $\|x_t - x\|_{\mathscr{G}(x)} = t \cdot \|z - x\|_{\mathscr{G}(x)} \leq \frac{3}{10}$ for $t \in [0, 1]$.

All of the calculations henceforth in this proof are performed when conditioning on $\mathfrak{E}_1$ and $\mathfrak{E}_2$, and that the step size satisfies the bound in $\mathsf{B}_0$, as this ensures that $\|z - x\|_{\mathscr{G}(x)} \leq \frac{3}{10}$. Now, we work from Equation (15) to find a suitable $\tau$ in Equation (14).

For $T_0^A$, we directly have from the definition of $\mathcal{S}$ that

$$T_0^A = \|\nabla f(z)\|^2_{\mathscr{G}(z)^{-1}} \leq \sup_{z: \|z-x\|_{\mathscr{G}(x)} \leq \frac{1}{2}} \|\nabla f(z)\|^2_{\mathscr{G}(z)^{-1}} = \mathsf{U}^2_{f,\mathcal{S}} .$$

For $T_1^A$, we use the self-concordance of the metric $\mathscr{G}$ (Lemma 26(2)) to obtain

$$
\begin{aligned}
T_1^A &= \|z - x\|_{\mathscr{G}(x_{t^\star})}^2 \\
&\leq \frac{1}{(1 - \|z - x\|_{\mathscr{G}(x)})^2} \cdot \|z - x\|_{\mathscr{G}(x)}^2 \\
&\leq \frac{17}{8} \cdot \left( h^2 \cdot \mathsf{U}_{f,\mathcal{S}}^2 + (2h)^{3/2} \cdot \mathsf{U}_{f,\mathcal{S}} \cdot \mathsf{I}_\varepsilon + 2h \cdot d \cdot \mathsf{N}_\varepsilon \right).
\end{aligned}
$$

The last inequality uses Equation (17) and that $\|z - x\|_{\mathscr{G}(x)} \leq \frac{3}{10}$. For $T_2^A$ and $T_3^A$ in Equation (15), we use different properties of metric corresponding to the settings of Lemmas 15 and 16.

**When $\mathscr{G}$ is self-concordant (Lemma 15)** Since $\|z - x\|_{\mathscr{G}(x)} \leq \frac{3}{10}$, we use Lemma 20 to get

$$
\begin{aligned}
T_2^A &= \log \det \mathscr{G}(z) - \log \det \mathscr{G}(x) \\
&\geq -3d \cdot \|x - z\|_{\mathscr{G}(x)} \\
&= -3d \cdot \sqrt{h^2 \cdot \mathsf{U}_{f,\mathcal{S}}^2 + 2h \cdot d \cdot \mathsf{N}_\varepsilon + (2h)^{3/2} \cdot \mathsf{U}_{f,\mathcal{S}} \cdot \mathsf{I}_\varepsilon} \\
&\overset{(a)}{\geq} -3 \cdot \left( h \cdot d \cdot \mathsf{U}_{f,\mathcal{S}} + \sqrt{2h \cdot d^3 \cdot \mathsf{N}_\varepsilon} + \sqrt{(2h)^{3/2} \cdot d^2 \cdot \mathsf{U}_{f,\mathcal{S}} \cdot \mathsf{I}_\varepsilon} \right)
\end{aligned}
$$

$$
\begin{aligned}
T_3^A &= \|z - x\|_{\mathscr{G}(x)}^2 - \|z - x\|_{\mathscr{G}(z)}^2 \\
&\geq -6 \cdot \|x - z\|_{\mathscr{G}(x)}^3 \\
&= -6 \cdot \left( h^2 \cdot \mathsf{U}_{f,\mathcal{S}}^2 + 2h \cdot d \cdot \mathsf{N}_\varepsilon + (2h)^{3/2} \cdot \mathsf{U}_{f,\mathcal{S}} \cdot \mathsf{I}_\varepsilon \right)^{3/2} \\
&\overset{(b)}{\geq} -6\sqrt{3} \cdot \left( h^3 \cdot \mathsf{U}_{f,\mathcal{S}}^3 + (2h \cdot d \cdot \mathsf{N}_\varepsilon)^{3/2} + ((2h)^{3/2} \cdot \mathsf{U}_{f,\mathcal{S}} \cdot \mathsf{I}_\varepsilon)^{3/2} \right).
\end{aligned}
$$

Inequality $(a)$ uses the fact that $\sqrt{a + b + c} \leq \sqrt{a} + \sqrt{b} + \sqrt{c}$ for $a, b, c \geq 0$, and inequality $(b)$ uses the convexity of $t \mapsto t^{3/2}$. Collecting the bounds for each $T_i^A$, $i \in \{0, 1, 2, 3\}$, we can give a lower bound for $\log R_{x \to z}$. Recall that this holds when conditioned on $\mathfrak{E}_1$ and $\mathfrak{E}_2$, and when the step size $h$ satisfies the bound $\mathsf{B}_0$.

$$
\begin{aligned}
\log R_{x \to z} &= -\frac{h}{4} \cdot T_0^A - \frac{17\lambda}{32} \cdot T_1^A + \frac{1}{2} \cdot T_2^A + \frac{1}{4h} \cdot T_3^A \\
&\geq -\frac{h \cdot \mathsf{U}_{f,\mathcal{S}}^2}{4} - \frac{\lambda}{4} \cdot (h^2 \cdot \mathsf{U}_{f,\mathcal{S}}^2 + 2h \cdot d \cdot \mathsf{N}_\varepsilon + (2h)^{3/2} \cdot \mathsf{U}_{f,\mathcal{S}} \cdot \mathsf{I}_\varepsilon) \\
&\quad - \frac{3}{2} \cdot \left( h \cdot d \cdot \mathsf{U}_{f,\mathcal{S}} + \sqrt{2h \cdot d^3 \cdot \mathsf{N}_\varepsilon} + \sqrt{(2h)^{3/2} \cdot d^2 \cdot \mathsf{U}_{f,\mathcal{S}} \cdot \mathsf{I}_\varepsilon} \right) \\
&\quad - \frac{3\sqrt{3}}{2} \cdot \left( h^2 \cdot \mathsf{U}_{f,\mathcal{S}}^3 + (2\mathsf{N}_\varepsilon)^{3/2} \cdot \sqrt{h \cdot d^3} + (2^{3/2}\mathsf{I}_\varepsilon)^{3/2} \cdot (h^5 \cdot \mathsf{U}_{f,\mathcal{S}}^6)^{1/4} \right). \quad (18)
\end{aligned}
$$

Let $C_1(\varepsilon) = \frac{\varepsilon^2}{486 \cdot \mathsf{N}_\varepsilon^3}$. Define the function $b_1(d, \mathsf{U}_{f,\mathcal{S}}, \lambda)$ as

$$
b_1(d, \mathsf{U}_{f,\mathcal{S}}, \lambda) \overset{\text{def}}{=} C_1(\varepsilon) \cdot \min \left\{ \frac{1}{\mathsf{U}_{f,\mathcal{S}}^2}, \frac{1}{\mathsf{U}_{f,\mathcal{S}}^{2/3}}, \frac{1}{(\mathsf{U}_{f,\mathcal{S}} \cdot \lambda)^{2/3}}, \frac{1}{d^3}, \frac{1}{d \cdot \lambda} \right\}.
$$

If the step size satisfies $h \leq b_1(d, \mathsf{U}_{f,\mathcal{S}}, \lambda)$, then $\mathsf{B}_0$ holds since

$$C_1(\varepsilon) \leq \min \left\{ \frac{1}{6}, \frac{3}{200\mathsf{N}_\varepsilon}, \frac{1}{25\mathsf{I}_\varepsilon^{2/3}} \right\} .$$

Moreover, when $h \leq b_1(d, \mathsf{U}_{f,\mathcal{S}}, \lambda)$, we have a lower bound for $\log R_{x \to z}$ in terms of $\varepsilon$ alone. From Lemma 31 with $x \leftarrow h, V \leftarrow C_1(\varepsilon), a \leftarrow \mathsf{U}_{f,\mathcal{S}}, b \leftarrow \lambda$, we have

$$\log R_{x \to z} \geq -4\varepsilon .$$

Therefore, choosing $\tau = e^{-4\varepsilon}$ and $\mathfrak{E} = \mathfrak{E}_1 \cap \mathfrak{E}_2$, we have for $x \in \mathcal{S}$ using Equation (14) that

$$\begin{aligned}
\mathbb{E}_{z \sim \mathcal{P}_x}[\min\{1, R_{x \to z} \cdot \mathbf{1}\{z \in \mathcal{K}\}\}] &\geq e^{-4\varepsilon} \cdot (1 - 2\varepsilon) \\
&\geq (1 - 4\varepsilon) \cdot (1 - 2\varepsilon) \geq 1 - 6\varepsilon .
\end{aligned}$$

From Equation (13), we get $\mathrm{d}_{\mathrm{TV}}(\mathcal{T}_x, \mathcal{P}_x) \leq 1 - (1 - 6\varepsilon) = 6\varepsilon$. We set $\varepsilon = \frac{1}{96}$, and note that $C_1(1/96) \leq \frac{1}{20}$ to complete the proof of Lemma 15.

**When $\mathscr{G}$ is self-concordant$_{++}$ (Lemma 16)** Recall that self-concordant$_{++}$ is equivalent to strongly, $\alpha$-lower trace, and average self-concordant. Since $\|z - x\|_{\mathscr{G}(x)} \leq \frac{3}{10}$, we have a lower bound for $T_2^A$ through Lemma 21, and the definition of $z$ in terms of $\gamma \sim \mathcal{N}(\mathbf{0}, \mathrm{I}_{d \times d})$.

$$\begin{aligned}
T_2^A &= \log \det \mathscr{G}(z) - \log \det \mathscr{G}(x) \\
&\geq \langle \nabla \log \det \mathscr{G}(x), z - x \rangle - (\alpha + 4) \cdot \frac{\|z - x\|_{\mathscr{G}(x)}^2}{(1 - \|z - x\|_{\mathscr{G}(x)})^2} \\
&= -h \cdot \langle \nabla \log \det \mathscr{G}(x), \mathscr{G}(x)^{-1} \nabla f(x) \rangle + \sqrt{2h} \cdot \langle \nabla \log \det \mathscr{G}(x), \mathscr{G}(x)^{-1/2} \gamma \rangle \\
&\quad - (\alpha + 4) \cdot \frac{\|z - x\|_{\mathscr{G}(x)}^2}{(1 - \|z - x\|_{\mathscr{G}(x)})^2} \\
&\geq -h \cdot \|\mathscr{G}(x)^{-1/2} \nabla \log \det \mathscr{G}(x)\| \cdot \|\mathscr{G}(x)^{-1/2} \nabla f(x)\| + \sqrt{2h} \cdot \langle \nabla \log \det \mathscr{G}(x), \mathscr{G}(x)^{-1/2} \gamma \rangle \\
&\quad - \frac{17}{8} \cdot (\alpha + 4) \cdot \|z - x\|_{\mathscr{G}(x)}^2 .
\end{aligned}$$

The final inequality uses the Cauchy-Schwarz inequality. Additionally, by conditioning on $\mathfrak{E}_3$ and using the fact that $x \in \mathcal{S}$,

$$T_2^A \geq -2h \cdot \sqrt{d} \cdot \mathsf{U}_{f,\mathcal{S}} - 2\sqrt{2h} \cdot \sqrt{d} \cdot \mathsf{I}_\varepsilon - \frac{17}{8} \cdot (\alpha + 4) \cdot (h^2 \cdot \mathsf{U}_{f,\mathcal{S}}^2 + 2h \cdot d \cdot \mathsf{N}_\varepsilon + (2h)^{3/2} \cdot \mathsf{U}_{f,\mathcal{S}} \cdot \mathsf{I}_\varepsilon) .$$

For $T_3^A$, we first note that a Dikin proposal $\xi = x + \sqrt{2h} \cdot \mathscr{G}(x)^{-1/2} \gamma$ satisfies

$$\|\xi - x\|_{\mathscr{G}(x)}^2 = \|\sqrt{2h} \cdot \mathscr{G}(x)^{-1/2} \gamma\|_{\mathscr{G}(x)}^2 = 2h \cdot \|\gamma\|^2 .$$

Since the calculations here are considered when conditioning on $\mathfrak{E}_1$ and $\mathfrak{E}_2$, and when the step size satisfies the bound $\mathsf{B}_0$,

$$\|\xi - x\|_{\mathscr{G}(x)}^2 \leq 2h \cdot d \cdot \mathsf{N}_\varepsilon \leq \frac{6}{200} ,$$

which implies $\|\xi - x\|_{\mathscr{G}(x)} \leq \frac{3}{10}$. Lemma 22 with $w \leftarrow z$ and $y \leftarrow \xi$ states that there exists $\bar{t} \in (0, 1)$ such that

$$T_3^A = \|z - x\|_{\mathscr{G}(x)}^2 - \|z - x\|_{\mathscr{G}(z)}^2$$

$$\geq \|\xi - x\|_{\mathscr{G}(x)}^2 - \|\xi - x\|_{\mathscr{G}(\xi)}^2 - 6 \cdot \frac{\|\xi + \bar{t}(z - \xi) - x\|_{\mathscr{G}(x)}^2 \cdot \|z - \xi\|_{\mathscr{G}(x)}}{(1 - \|\xi + \bar{t}(z - \xi) - x\|_{\mathscr{G}(x)})^3} \ .$$

Note that $z - \xi = -h \cdot \mathscr{G}(x)^{-1} \nabla f(x)$, and therefore

$$\|\xi + \bar{t}(z - \xi) - x\|_{\mathscr{G}(x)}^2 \leq \max\{\|z - x\|_{\mathscr{G}(x)}^2, \|\xi - x\|_{\mathscr{G}(x)}^2\}$$

$$\leq h^2 \cdot \mathsf{U}_{f,\mathcal{S}}^2 + 2h \cdot d \cdot \mathsf{N}_\varepsilon + (2h)^{3/2} \cdot \mathsf{U}_{f,\mathcal{S}} \cdot \mathsf{I}_\varepsilon$$

$$\|z - \xi\|_{\mathscr{G}(x)} \leq h \cdot \mathsf{U}_{f,\mathcal{S}} \ .$$

When conditioning on the event $\mathfrak{E}_4$ and assuming that the step size $h$ additionally satisfies $h \leq \frac{r_\varepsilon^2}{2d}$ for $r_\varepsilon$ in the definition of average self-concordance

$$T_3^A \geq -18h \cdot \mathsf{U}_{f,\mathcal{S}} \cdot (h^2 \cdot \mathsf{U}_{f,\mathcal{S}}^2 + 2h \cdot d \cdot \mathsf{N}_\varepsilon + (2h)^{3/2} \cdot \mathsf{U}_{f,\mathcal{S}} \cdot \mathsf{I}_\varepsilon) - 4h \cdot \varepsilon \ .$$

In summary, let the step size $h$ satisfy the bound $\mathsf{B}_0$ and $h \leq \frac{r_\varepsilon^2}{2d}$. When conditioned on the events $\mathfrak{E}_i$ for $i \in [4]$, then continuing from Equation (15)

$$\log R_{x \to z} \geq -\frac{h}{4} \cdot T_0^A - \frac{\lambda}{4} \cdot T_1^A + \frac{1}{2} \cdot T_2^A + \frac{1}{4h} \cdot T_3^A$$

$$\geq -\frac{h \cdot \mathsf{U}_{f,\mathcal{S}}^2}{4} - \frac{17\lambda}{32} \cdot (h^2 \cdot \mathsf{U}_{f,\mathcal{S}}^2 + 2h \cdot d \cdot \mathsf{N}_\varepsilon + (2h)^{3/2} \cdot \mathsf{U}_{f,\mathcal{S}} \cdot \mathsf{I}_\varepsilon)$$

$$- h \cdot \sqrt{d} \cdot \mathsf{U}_{f,\mathcal{S}} - \sqrt{2h \cdot d \cdot \mathsf{I}_\varepsilon^2}$$

$$- \frac{17}{16} \cdot (\alpha + 4) \cdot (h^2 \cdot \mathsf{U}_{f,\mathcal{S}}^2 + 2h \cdot d \cdot \mathsf{N}_\varepsilon + (2h)^{3/2} \cdot \mathsf{U}_{f,\mathcal{S}} \cdot \mathsf{I}_\varepsilon)$$

$$- \varepsilon - \frac{9}{2} \cdot (h^2 \cdot \mathsf{U}_{f,\mathcal{S}}^3 + 2h \cdot d \cdot \mathsf{U}_{f,\mathcal{S}} \cdot \mathsf{N}_\varepsilon + (2h)^{3/2} \cdot \mathsf{U}_{f,\mathcal{S}}^2 \cdot \mathsf{I}_\varepsilon) \ . \tag{19}$$

Let $C_2(\varepsilon) = \min\left\{\frac{\varepsilon^2}{100 \cdot \mathsf{I}_\varepsilon^2}, r_\varepsilon^2\right\}$. Define the function $b_2(d, \mathsf{U}_{f,\mathcal{S}}, \lambda, \alpha)$ as

$$b_2(d, \mathsf{U}_{f,\mathcal{S}}, \lambda, \alpha)$$

$$\stackrel{\text{def}}{=} C_2(\varepsilon) \cdot \min\left\{\frac{1}{d \cdot (\alpha + 4)}, \frac{1}{d \cdot \mathsf{U}_{f,\mathcal{S}}}, \frac{1}{d \cdot \lambda}, \frac{1}{(\lambda \cdot \mathsf{U}_{f,\mathcal{S}})^{2/3}}, \frac{1}{(\mathsf{U}_{f,\mathcal{S}} \cdot (\alpha + 4))^{2/3}}, \frac{1}{\mathsf{U}_{f,\mathcal{S}}^2}\right\} \ .$$

If $h \leq b_2(d, \mathsf{U}_{f,\mathcal{S}}, \lambda, \alpha)$, then $\mathsf{B}_0$ holds since $\alpha \geq 0$ and

$$C_2(\varepsilon) \leq \min\left\{\frac{1}{6}, \frac{3}{200 \mathsf{N}_\varepsilon}, \frac{1}{25 \mathsf{I}_\varepsilon^{2/3}}\right\} \ .$$

Also, when $h \leq b_2(d, \mathsf{U}_{f,\mathcal{S}}, \lambda, \alpha)$, we have a lower bound for $\log R_{x \to z}$ solely in terms of $\varepsilon$. From Lemma 32 with $x \leftarrow h, V \leftarrow C_1(\varepsilon), a \leftarrow \mathsf{U}_{f,\mathcal{S}}, b \leftarrow \lambda, c \leftarrow (\alpha + 4)$, we have

$$\log R_{x \to z} \geq -2\varepsilon \ .$$

Therefore, choosing $\tau = e^{-2\varepsilon}$ and $\mathfrak{E} = \bigcap_{i=1}^{4} \mathfrak{E}_i$, we have for $x \in \mathcal{S}$ using Equation (14) that

$$
\mathbb{E}_{z \sim \mathcal{P}_x}[\min\{1, R_{x \to z} \cdot \mathbf{1}\{z \in \mathcal{K}\}\}] \geq e^{-2\varepsilon} \cdot (1 - 4\varepsilon)
$$
$$
\geq (1 - 2\varepsilon) \cdot (1 - 4\varepsilon) \geq 1 - 6\varepsilon .
$$

From Equation (13), we get $\mathrm{d}_{\mathrm{TV}}(\mathcal{T}_x, \mathcal{P}_x) \leq 1 - (1 - 6\varepsilon) = 6\varepsilon$. We set $\varepsilon = \frac{1}{96}$, and note that $C_2(1/96) \leq \frac{1}{20}$ to complete the proof of Lemma 16. ∎

## D.4. Proofs of isoperimetry lemmas in Section D.3

Here, we give the proofs of Lemmas 17 and 18 which were used to give lower bounds on the conductance of the reversible Markov chain $\mathbf{T}$ formed by an iteration of MAPLA.

### D.4.1. PROOF OF LEMMA 17

The proof of this lemma follows the proof of Kook and Vempala (2024, Lem. B.6), with care given to restrictions of the distribution to subsets of the support.

**Proof** Let $\mathbb{B}(0, r)$ be a ball of radius $r > 0$ centered at the the origin, and define the set $\mathcal{D}_{|r} \stackrel{\text{def}}{=} \mathcal{D} \cap \mathbb{B}(0, r)$ for a set $\mathcal{D} \subseteq \mathcal{K}$. Consider the restriction $\Pi_{\mathcal{S}_{|r}}$ of $\Pi$ over $\mathcal{S}_{|r}$. Since $\mathcal{S}$ is convex, $\Pi_{\mathcal{S}_{|r}}$ is log-concave. For any partition $\{A_1, A_2, A_3\}$ of $\mathcal{S}$, $\{A_{1|r}, A_{2|r}, A_{3|r}\}$ forms a partition of $\mathcal{S}_{|r}$, and by Lovász and Vempala (2003, Thm. 2.2), we have

$$
\Pi_{\mathcal{S}_{|r}}(A_{3|r}) \geq \mathrm{CR}(A_{2|r}, A_{1|r}; \mathcal{S}_{|r}) \cdot \Pi_{\mathcal{S}_{|r}}(A_{2|r}) \cdot \Pi_{\mathcal{S}_{|r}}(A_{1|r}) .
$$

Above, $\mathrm{CR}(y, x; \mathcal{D})$ is the cross-ratio between $x$ and $y$ in $\mathcal{D}$, and is defined as

$$
\mathrm{CR}(y, x; \mathcal{D}) = \frac{\|y - x\| \cdot \|p - q\|}{\|y - p\| \cdot \|x - q\|}.
$$

where $p$ and $q$ are the end points of the extensions of the line segment between $y$ and $x$ to the boundary of $\mathcal{D}$ respectively. The cross ratio between sets is defined as

$$
\mathrm{CR}(\mathcal{D}_1, \mathcal{D}_2; \mathcal{D}) = \inf_{y \in \mathcal{D}_1, x \in \mathcal{D}_2} \mathrm{CR}(y, x; \mathcal{D}) .
$$

Since $\mathscr{G}$ is $\nu$-symmetric, from Laddha et al. (2020, Lem. 2.3), we have for any $x \in A_{1|r}, y \in A_{2|r}$ that

$$
\mathrm{CR}(y, x; \mathcal{S}_{|r}) \geq \frac{\|y - x\|_{\mathscr{G}(y)}}{\sqrt{\nu}} \Rightarrow \mathrm{CR}(A_{2|r}, A_{1|r}; \mathcal{S}_{|r}) \geq \inf_{y \in A_{2|r}, x \in A_{1|r}} \frac{\|y - x\|_{\mathscr{G}(y)}}{\sqrt{\nu}} .
$$

Since $A_{i|r} \subseteq A_i$, the lower bound above is at least $\inf_{y \in A_2, x \in A_1} \frac{\|x - y\|_{\mathscr{G}(y)}}{\sqrt{\nu}}$. By setting $r \to \infty$, the dominated convergence theorem implies that

$$
\Pi_{\mathcal{S}}(A_3) \geq \inf_{y \in A_2, x \in A_1} \frac{\|y - x\|_{\mathscr{G}(y)}}{\sqrt{\nu}} \cdot \Pi_{\mathcal{S}}(A_2) \cdot \Pi_{\mathcal{S}}(A_1)
$$

which is the statement of the lemma. ∎

D.4.2. PROOF OF LEMMA 18

We first state a one-dimensional inequality that the proof of Lemma 18 we give relies on. The proof of the below one-dimensional inequality is given in Section D.4.3.

**Lemma 23** *Consider a differentiable function* $\mathfrak{g} : \mathbb{R} \to (0, \infty)$. *Assume that there exists* $\kappa > 0$ *such that for all* $x \in \mathbb{R}$, $|\mathfrak{g}'(x)| \leq \frac{2}{\kappa} \cdot \mathfrak{g}(x)^{3/2}$. *If a twice differentiable function* $V : \mathbb{R} \to \mathbb{R}$ *satisfies* $(1, \mathfrak{g})$-*curvature lower bound, then for all* $x \in \mathbb{R}$,

$$\frac{\exp(-V(x))}{\sqrt{\mathfrak{g}(x)}} \geq C_\kappa \cdot \min \left\{ \int_{-\infty}^x \exp(-V(t)) \mathrm{d}t, \int_x^\infty \exp(-V(t)) \mathrm{d}t \right\}$$

*where* $C_\kappa = \frac{\kappa}{8+4\kappa}$.

**Proof** This proof follows the structure of the proofs of Gopi et al. (2023, Lem. 9) and Kook and Vempala (2024, Lem. B.7), and we give a few more details for the convenience of the reader. We begin by first noting that $f$ is 1-relatively convex with respect to itself. Let the extended version of $f$ be $\tilde{f} : \mathbb{R}^d \to \mathbb{R}$ which is defined as

$$\tilde{f}(x) = \begin{cases} f(x) & x \in \text{int}(\mathcal{K}) \\ \infty & x \notin \text{int}(\mathcal{K}) \end{cases}$$

and the associated extended density $\tilde{\pi} \propto \exp(-\tilde{f})$. For positive functions $f_1, f_2, f_3, f_4$ where $f_1$ and $f_2$ are upper semicontinuous, and $f_3$ and $f_4$ are lower continuous, we have the following equivalence from Gopi et al. (2023, Lem. 8):

$$\left( \int f_1(x) \tilde{\pi}(x) \mathrm{d}x \right) \cdot \left( \int f_2(x) \tilde{\pi}(x) \mathrm{d}x \right) \leq \left( \int f_3(x) \tilde{\pi}(x) \mathrm{d}x \right) \cdot \left( \int f_4(x) \tilde{\pi}(x) \mathrm{d}x \right)$$

$$\Updownarrow$$

$$\left( \int_E f_1 \exp(-\tilde{f}) \right) \cdot \left( \int_E f_2 \exp(-\tilde{f}) \right) \leq \left( \int_E f_3 \exp(-\tilde{f}) \right) \cdot \left( \int_E f_4 \exp(-\tilde{f}) \right)$$

for any $a, b \in \mathbb{R}^d$, $\gamma \in \mathbb{R}$, where $\int_E h \overset{\text{def}}{=} \int_0^1 h(a + t(b - a)) \exp(-\gamma t) \mathrm{d}t$. Since $\tilde{\pi}(x) = \exp(-\tilde{f}(x)) = 0$ for $x \notin \text{int}(\mathcal{K})$, this equivalence also holds with the substitutions $\tilde{\pi} \leftarrow \pi$ and $\tilde{f} \leftarrow f$. Let $C_{\sqrt{\mu}} = \frac{\sqrt{\mu}}{8 + 4\sqrt{\mu}}$ and $\tilde{\mathcal{G}} = \mu \cdot \mathcal{G}$ (which implies that $d_{\tilde{\mathcal{G}}} = \sqrt{\mu} \cdot d_{\mathcal{G}}$). The inequality we would like to show can be expressed in terms of indicator functions as

$$\int \pi(x) \mathbf{1}_{S_3}(x) \mathrm{d}x \cdot \int (C_{\sqrt{\mu}} \cdot d_{\tilde{\mathcal{G}}}(S_1, S_2))^{-1} \pi(x) \mathrm{d}x \geq \int \pi(x) \mathbf{1}_{S_2}(x) \mathrm{d}x \cdot \int \pi(x) \mathbf{1}_{S_1}(x) \mathrm{d}x.$$

Consider $f_1 = \mathbf{1}_{\overline{S_1}}$, $f_2 = \mathbf{1}_{\overline{S_2}}$, $f_3 = \mathbf{1}_{\text{int}(\mathcal{K}) \backslash \overline{S_1} \backslash \overline{S_2}}$, and $f_4 = (C_{\sqrt{\mu}} \cdot d_{\tilde{\mathcal{G}}}(S_1, S_2))^{-1}$, where $\overline{S}$ is closure of a set $S$. With this construction, $f_1, f_2$ are upper-semicontinuous, and $f_3, f_4$ are lower-semicontinuous. If the top inequality in the equivalence holds with these $\{f_i\}_{i=1}^4$, then this implies

our required inequality as

$$\int \pi(x)\mathbf{1}_{S_3}\mathrm{d}x \cdot \int (C_{\sqrt{\mu}} \cdot d_{\tilde{\mathscr{G}}}(S_1, S_2))^{-1}\pi(x)\mathrm{d}x = \int f_3(x)\pi(x)\mathrm{d}x \cdot \int f_4(x)\pi(x)\mathrm{d}x$$

$$\geq \int f_1(x)\pi(x)\mathrm{d}x \cdot \int f_2(x)\pi(x)\mathrm{d}x$$

$$\geq \int \pi(x)\mathbf{1}_{S_1}\mathrm{d}x \cdot \int \pi(x)\mathbf{1}_{S_2}(x)\mathrm{d}x \ .$$

Due to the equivalence from the localisation lemma, as noted in Kook and Vempala (2024, Lem. B.7) it suffices to establish that for all $\gamma \in \mathbb{R}$ and $a, b \in \mathbb{R}^d$, and $a + t \cdot (b - a) \in \mathrm{int}(\mathcal{K})$ for $t \in [0, 1]$

$$C_{\sqrt{\mu}}\cdot d_{\tilde{\mathscr{G}}}(S_1, S_2) \int_0^1 e^{\gamma t - f(a+t\cdot(b-a))}\mathbf{1}_{\overline{S_1}}(a+t\cdot(b-a))\mathrm{d}t\cdot\int_0^1 e^{\gamma t - f(a+t\cdot(b-a))}\mathbf{1}_{\overline{S_2}}(a+t\cdot(b-a))\mathrm{d}t$$

$$\leq \int_0^1 e^{\gamma t - f(a+t\cdot(b-a))}\mathrm{d}t \cdot \int_0^1 e^{\gamma t - f(a+t\cdot(b-a))}\mathbf{1}_{S_3}(a + t \cdot (b - a))\mathrm{d}t \ .$$

We refer to the above inequality as the "needle" inequality.

Now we define the following quantities for arbitrary $a, b \in \mathbb{R}^d$ and $\gamma \in \mathbb{R}$.

$$V(t) \stackrel{\mathrm{def}}{=} f(a + t \cdot (b - a)) - \gamma t$$

$$T_i \stackrel{\mathrm{def}}{=} \{t \in [0, 1] : a + t \cdot (b - a) \in S_i\} \qquad \text{for each } i \in [3]$$

$$\mathfrak{g}(t) \stackrel{\mathrm{def}}{=} (b - a)^\top \tilde{\mathscr{G}}(a + t \cdot (b - a))(b - a) \ .$$

Note that $V$ is 1-relatively convex with respect to $\mathfrak{g}$ as

$$V''(t) = (b - a)^\top \nabla^2 f(a + t \cdot (b - a))(b - a)$$

$$\geq \mu \cdot (b - a)^\top \mathscr{G}(a + t \cdot (b - a))(b - a)$$

$$= (b - a)^\top \tilde{\mathscr{G}}(a + t \cdot (b - a))(b - a) = \mathfrak{g}(t) \ .$$

Additionally note the following property of $\mathfrak{g}$.

$$|\mathfrak{g}'(t)| = |\mathrm{D}\tilde{\mathscr{G}}(a + t \cdot (b - a))[b - a, b - a, b - a]|$$

$$\leq \mu \cdot |\mathrm{D}\mathscr{G}(a + t \cdot (b - a))[b - a, b - a, b - a]|$$

$$\leq 2\mu \cdot \|b - a\|^3_{\mathscr{G}(a+t\cdot(b-a))}$$

$$\leq \frac{2}{\sqrt{\mu}} \cdot \|b - a\|^3_{\tilde{\mathscr{G}}(a+t\cdot(b-a))} = \frac{2}{\sqrt{\mu}} \cdot \mathfrak{g}(t)^{3/2} \ .$$

The "needle" inequality can be expressed in terms of one-dimensional integrals as shown below.

$$C_{\sqrt{\mu}} \cdot d_{\tilde{\mathscr{G}}}(S_1, S_2) \cdot \int_{t\in T_1} e^{-V(t)}\mathrm{d}t \cdot \int_{t\in T_2} e^{-V(t)}\mathrm{d}t \leq \int_{t\in[0,1]} e^{-V(t)}\mathrm{d}t \cdot \int_{t\in T_3} e^{-V(t)}\mathrm{d}t \ .$$

Let $d_{\mathfrak{g}}(v_1, v_2) = \int_{v_1}^{v_2} \sqrt{\mathfrak{g}(t)}\mathrm{d}t$. As remarked in the proof of Gopi et al. (2023, Lem. 9),

$$d_{\mathfrak{g}}(T_1, T_2) \geq d_{\tilde{\mathscr{G}}}(S_1, S_2) \ .$$

Therefore, if

$$C_{\sqrt{\mu}} \cdot d_{\mathfrak{g}}(T_1, T_2) \cdot \int_{t \in T_1} e^{-V(t)} \mathrm{d}t \cdot \int_{t \in T_2} e^{-V(t)} \mathrm{d}t \leq \int_{t \in [0,1]} e^{-V(t)} \mathrm{d}t \cdot \int_{t \in T_3} e^{-V(t)} \mathrm{d}t$$

is true, then the "needle" inequality is true, which implies the inequality in the statement of the lemma. The remainder of the proof follows the proof of Gopi et al. (2023, Lem. 9). Assume that $T_3$ is a single interval. This implies that $T_1 = [c, c'], T_3 = [c', d'], T_2 = [d', d]$ for $0 \leq c < c' < d' < d \leq 1$. Then,

$$d_{\mathfrak{g}}(T_1, T_2) = \inf_{u \in T_1, v \in T_2} d_{\mathfrak{g}}(u, v) = d_{\mathfrak{g}}(c', d') = \int_{c'}^{d'} \sqrt{\mathfrak{g}(t)} \mathrm{d}t .$$

With Lemma 23 with $\kappa \leftarrow \sqrt{\mu}$, we show the inequality in this setting as shown below.

$$
\begin{aligned}
\int_{c'}^{d'} \exp(-V(t)) \mathrm{d}t &\geq \min_{t \in [c', d']} \frac{\exp(-V(t))}{\sqrt{\mathfrak{g}(t)}} \cdot \int_{c'}^{d'} \sqrt{\mathfrak{g}(t)} \mathrm{d}t \\
&\geq \min_{t \in [c', d']} C_{\sqrt{\mu}} \cdot \min\left\{ \int_{-\infty}^{t} e^{-V(t)} \mathrm{d}t, \int_{t}^{\infty} \exp(-V(t)) \mathrm{d}t \right\} \cdot \int_{c'}^{d'} \sqrt{\mathfrak{g}(t)} \mathrm{d}t \\
&\geq C_{\sqrt{\mu}} \cdot \min\left\{ \int_{c}^{c'} e^{-V(t)} \mathrm{d}t, \int_{d'}^{d} e^{-V(t)} \mathrm{d}t \right\} \cdot \int_{c'}^{d'} \sqrt{\mathfrak{g}(t)} \mathrm{d}t \\
&= C_{\sqrt{\mu}} \cdot \min\left\{ \frac{\int_{c}^{c'} \exp(-V(t)) \mathrm{d}t}{Z}, \frac{\int_{d'}^{d} \exp(-V(t)) \mathrm{d}t}{Z} \right\} \cdot Z \cdot \int_{c'}^{d'} \sqrt{\mathfrak{g}(t)} \mathrm{d}t \\
&\geq C_{\sqrt{\mu}} \cdot \int_{c}^{c'} \exp(-V(t)) \mathrm{d}t \cdot \int_{d'}^{d} \exp(-V(t)) \mathrm{d}t \cdot \int_{c'}^{d'} \sqrt{\mathfrak{g}(t)} \mathrm{d}t \cdot \frac{1}{Z} .
\end{aligned}
$$

where $Z = \int_{c}^{d} e^{-V(t)} \mathrm{d}t$. The final statement uses $\min\{a, b\} \geq ab$ for $0 < a, b \leq 1$. When $T_3$ is a collection of disjoint intervals, the general trick from Lovász and Simonovits (1993) applies here as performed in Gopi et al. (2023), and this inequality is applied to each interval in $T_3$ and its neighbouring intervals. Thus, we have shown the inequality, which implies the "needle" inequality, and therefore proving the original statement of the lemma. ■

### D.4.3. PROOF OF LEMMA 23

**Proof** This proof is inspired by Gopi et al. (2023, Proof of Lem. 7). Assume that $V'(x) \geq 0$, and define $r = x + \frac{\kappa}{4\sqrt{\mathfrak{g}(x)}}$. By the property of $\mathfrak{g}$, we have for any $t \in [x, r]$.

$$\frac{1}{\sqrt{\mathfrak{g}(t)}} - \frac{1}{\sqrt{\mathfrak{g}(x)}} = -\int_{x}^{t} \frac{\mathfrak{g}'(s)}{2\mathfrak{g}(s)^{3/2}} \leq \frac{t - x}{\kappa} \leq \frac{r - x}{\kappa} = \frac{1}{4\sqrt{\mathfrak{g}(x)}} \Rightarrow \mathfrak{g}(t) \geq \frac{1}{2}\mathfrak{g}(x) .$$

Since $V$ is 1-relatively convex w.r.t. $\mathfrak{g}$, for all $t \in [x, r]$,

$$V''(t) \geq \mathfrak{g}(t) \geq \frac{1}{2}\mathfrak{g}(x) .$$

and with the assumption that $V'(x) \geq 0$, we get

$$V(t) \geq V(x) + V'(x)(t - x) + \int_x^t (t - s)V''(s)\mathrm{d}s \geq V(x) + \frac{(t - x)^2}{4}\mathfrak{g}(x) \, ,$$

$$V'(r) = V'(x) + \int_x^r V''(s)\mathrm{d}s \geq \frac{r - x}{2}\mathfrak{g}(x) = \frac{\kappa}{8}\sqrt{\mathfrak{g}(x)} \, .$$

We also have for $t > r$ that

$$V(t) \geq V(r) + V'(r)(t - r) \geq V(x) + V'(x)(r - x) + V'(r)(t - r) \geq V(x) + \frac{\kappa(t - r)}{8}\sqrt{\mathfrak{g}(x)} \, .$$

With these, we get

$$\int_x^\infty \exp(-V(t))\mathrm{d}t = \int_x^r \exp(-V(t))\mathrm{d}t + \int_r^\infty \exp(-V(t))\mathrm{d}t$$

$$\leq \exp(-V(x)) \int_x^r \exp\left(-\frac{(t - x)^2}{4}\mathfrak{g}(x)\right) \mathrm{d}t$$

$$+ \int_r^\infty \exp(-V(x)) \exp\left(-\frac{\kappa(t - r)\sqrt{\mathfrak{g}(x)}}{8}\right) \mathrm{d}t$$

$$\leq \exp(-V(x)) \cdot \frac{4 + \frac{8}{\kappa}}{\sqrt{\mathfrak{g}(x)}} \, .$$

For the setting $V'(x) \leq 0$, define $r = x - \frac{\alpha}{4\sqrt{\mathfrak{g}(x)}}$ and for any $t \in [r, x]$,

$$\frac{1}{\sqrt{\mathfrak{g}(x)}} - \frac{1}{\sqrt{\mathfrak{g}(t)}} = -\int_t^x \frac{\mathfrak{g}'(s)}{2\mathfrak{g}(s)^{3/2}} \geq -\frac{(x - t)}{\kappa} \geq \frac{r - x}{\kappa} = -\frac{1}{4\sqrt{\mathfrak{g}(x)}} \Rightarrow \mathfrak{g}(t) \geq \frac{1}{2}\mathfrak{g}(x) \, .$$

With the assumption that $V'(x) \leq 0$, we get

$$V(t) \geq V(x) + V'(x)(t - x) + \int_x^t (t - s)V''(s)\mathrm{d}s \geq V(x) \geq \int_t^x (t - s)V''(s)\mathrm{d}s$$

$$\geq \frac{(t - x)^2}{4}\mathfrak{g}(x) \, ,$$

$$V'(r) = V'(x) + \int_x^r V''(s)\mathrm{d}s \leq -\int_r^x V''(s)\mathrm{d}s \leq \frac{r - x}{2}\mathfrak{g}(x) = -\frac{\kappa}{8}\sqrt{\mathfrak{g}(x)} \, .$$

We also have for $t < r$ that

$$V(t) \geq V(r) + V'(r)(t - r)$$

$$\geq V(x) + V'(x)(r - x) + V'(r)(t - r)$$

$$\geq V(x) + \frac{\kappa(r - t)}{8}\sqrt{\mathfrak{g}(x)} \, .$$

Analogous to the case $V'(x) \geq 0$, consider the integrals $\int_{-\infty}^r \exp(-V(t))\mathrm{d}t$ and $\int_r^x \exp(-V(t))\mathrm{d}t$. This also results in

$$\int_{-\infty}^x \exp(-V(t))\mathrm{d}t \leq \exp(-V(x)) \cdot \frac{4 + \frac{8}{\kappa}}{\sqrt{\mathfrak{g}(x)}} \, .$$

■

## D.5. Other technical lemmas and their proofs

**Proof** [Proof of Lemma 19] Consider any geodesic $\xi : [0,1] \to \mathrm{int}(\mathcal{K})$ such that $\xi(0) = x$ and $\xi(1) = y$. Let $\bar{t}$ be the time when $\xi(\bar{t})$ hits the boundary of $\mathcal{E}_x^{\mathscr{G}}(r)$, and note that $\bar{t} \leq 1$ as $\xi(1) = y$. Then, for $\delta(t) := \|\xi(t) - x\|_{\mathscr{G}(x)}$,

$$\frac{\mathrm{d}}{\mathrm{d}t}\delta(t)^2 = 2\delta(t)\delta'(t) = 2\langle \xi'(t), \xi(t) - x \rangle_{\mathscr{G}(x)} \leq 2\|\xi'(t)\|_{\mathscr{G}(x)}\|\xi(t) - x\|_{\mathscr{G}(x)}$$

and this implies that $\delta'(t) \leq \|\xi'(t)\|_{\mathscr{G}(x)}$. Therefore,

$$d_{\mathscr{G}}(x,y) = \int_0^1 \|\xi'(t)\|_{\mathscr{G}(\xi(t))}\mathrm{d}t \geq \int_0^{\bar{t}} \|\xi'(t)\|_{\mathscr{G}(\xi(t))}\mathrm{d}t \geq \int_0^{\bar{t}} \|\xi'(t)\|_{\mathscr{G}(x)} \cdot (1 - \|\xi(t) - x\|_{\mathscr{G}(x)})\mathrm{d}t \,,$$

where the last inequality is due to the self-concordance of $\mathscr{G}$ (Lemma 26(2)) and that $\|\xi(t) - x\|_{\mathscr{G}(x)} < 1$ for $t \in (0, \bar{t})$. The final integral in the above chain is at least

$$\int_0^{\bar{t}} \delta'(t)(1 - \delta(t))\mathrm{d}t = \delta(\bar{t}) - \frac{1}{2}\delta(\bar{t})^2 = r - \frac{1}{2}r^2 \,.$$

The second statement of the lemma directly follows from the remainder of proof of Nesterov and Todd (2002, Lem. 3.1). ∎

**Proof** [Proof of Lemma 20] From Lemma 26(2) and the assumption that $\|y - x\|_{\mathscr{G}(x)} < 1$, we have

$$\mathscr{G}(y) \succeq (1 - \|y - x\|_{\mathscr{G}(x)})^2 \cdot \mathscr{G}(x) \Leftrightarrow \mathscr{G}(x)^{-1/2}\mathscr{G}(y)\mathscr{G}(x)^{-1/2} \succeq (1 - \|y - x\|_{\mathscr{G}(x)})^2 \cdot \mathrm{I}_{d \times d} \,.$$

Hence, for $M = \mathscr{G}(x)^{-1/2}\mathscr{G}(y)\mathscr{G}(x)^{-1/2}$, we have

$$\log \det \mathscr{G}(y)\mathscr{G}(x)^{-1} = \log \det M = \sum_{i=1}^d \log \lambda_i(M)$$

$$\geq 2d \cdot \log(1 - \|y - x\|_{\mathscr{G}(x)}) \geq -3d \cdot \|y - x\|_{\mathscr{G}(x)} \,,$$

where the last step uses Lemma 29. Also, from Lemma 26(2), we have

$$\mathscr{G}(y) \preceq \frac{1}{(1 - \|y - x\|_{\mathscr{G}(x)})^2} \cdot \mathscr{G}(x) \Rightarrow \|x - y\|_{\mathscr{G}(y)}^2 \leq \frac{\|x - y\|_{\mathscr{G}(x)}^2}{(1 - \|y - x\|_{\mathscr{G}(x)})^2} \,.$$

Using Lemma 30, this implies that

$$\|x - y\|_{\mathscr{G}(x)}^2 - \|x - y\|_{\mathscr{G}(y)}^2 \geq \|x - y\|_{\mathscr{G}(x)}^2 - \frac{1}{(1 - \|x - y\|_{\mathscr{G}(x)})^2} \cdot \|x - y\|_{\mathscr{G}(x)}^2 \geq -6 \cdot \|x - y\|_{\mathscr{G}(x)}^3 \,.$$

∎

**Proof** [Proof of Lemma 21] Let $s(t) = x + t \cdot (y - x)$, and $\varphi(t) = \log \det \mathscr{G}(s(t))$. By Taylor's theorem, there exists $t^\star \in (0,1)$ such that

$$\log \det \mathscr{G}(y)\mathscr{G}(x)^{-1} = \varphi(1) - \varphi(0) = \varphi'(0) + \varphi''(t^\star) \,.$$

We have the following expressions for $\varphi'(0)$ and $\varphi''(t)$.

$$\varphi'(0) = \langle \nabla \log \det \mathscr{G}(x), y - x \rangle$$
$$\varphi''(t) = \text{trace}(\mathscr{G}(s(t))^{-1} \mathrm{D}^2 \mathscr{G}(s(t))[y - x, y - x])$$
$$- \|\mathscr{G}(s(t))^{-1/2} \mathrm{D}\mathscr{G}(s(t))[y - x]\mathscr{G}(s(t))^{-1/2}\|_{\mathrm{F}}^2 .$$

As strong self-concordance implies self-concordance, Lemma 26(2) implies that for any $t \in [0, 1]$

$$\mathscr{G}(s(t)) \preceq \frac{\mathscr{G}(x)}{(1 - \|s(t) - x\|_{\mathscr{G}(x)})^2} \preceq \frac{\mathscr{G}(x)}{(1 - \|y - x\|_{\mathscr{G}(x)})^2} ,$$

where the last step uses the fact that $\|s(t) - x\|_{\mathscr{G}(x)} = t \cdot \|y - x\|_{\mathscr{G}(x)}$. Finally, we use the definitions of strong self-concordance and $\alpha$-lower trace self-concordance to get

$$\varphi''(t^\star) = \text{trace}(\mathscr{G}(s(t^\star))^{-1} \mathrm{D}^2 \mathscr{G}(s(t^\star))[y - x, y - x])$$
$$- \|\mathscr{G}(s(t^\star))^{-1/2} \mathrm{D}\mathscr{G}(s(t^\star))[y - x]\mathfrak{G}(s(t^\star))^{-1/2}\|_{\mathrm{F}}^2$$
$$\geq -\alpha \cdot \|y - x\|_{\mathscr{G}(s(t^\star))}^2 - 4 \cdot \|y - x\|_{\mathscr{G}(s(t^\star))}^2$$
$$\geq -(\alpha + 4) \cdot \frac{\|y - x\|_{\mathscr{G}(x)}^2}{(1 - \|y - x\|_{\mathscr{G}(x)})^2} .$$

$\blacksquare$

**Proof** [Proof of Lemma 22] For $t, s \in [0, 1]$, define the following functions.

$$y(t) = y + t \cdot (w - y), \quad p(t) = \|y_t - x\|_{\mathscr{G}(y_t)}^2 - \|y_t - x\|_{\mathscr{G}(x)}^2$$
$$u(s; t) = x + s \cdot (y(t) - x), \quad q(s; t) = 2\langle y(t) - x, (w - y)\rangle_{\mathscr{G}(u(s;t))} .$$

By Taylor's theorem, there exists $t^\star \in (0, 1)$ such that

$$\Delta(w; x) - \Delta(y; x) = p(1) - p(0)$$
$$= p'(t^\star)$$
$$= 2\langle y(t^\star) - x, w - y\rangle_{\mathscr{G}(y(t^\star))} - 2\langle y(t^\star) - x, w - y\rangle_{\mathscr{G}(x)}$$
$$+ \mathrm{D}\mathscr{G}(y(t^\star))[y(t^\star) - x, y(t^\star) - x, w - y]$$
$$= q(1; t^\star) - q(0; t^\star) + \mathrm{D}\mathscr{G}(y(t^\star))[y(t^\star) - x, y(t^\star) - x, w - y] .$$

Since $q(.; t^\star)$ is differentiable, we also have by Taylor's theorem that there exists $\bar{s} \in (0, 1)$.

$$q(1; t^\star) - q(0; t^\star) = q'(\bar{s}; t^\star)$$
$$= 2\mathrm{D}\mathscr{G}(u(\bar{s}; t^\star))[y(t^\star) - x, y(t^\star) - x, w - y] .$$

The self-concordance of $\mathscr{G}$ enables the bounds

$$\Delta(w; x) - \Delta(y; x) = 2\mathrm{D}\mathscr{G}(u(\bar{s}; t^\star))[y(t^\star) - x, y(t^\star) - x, w - y]$$
$$+ \mathrm{D}\mathscr{G}(y(t^\star))[y(t^\star) - x, y(t^\star) - x, w - y]$$
$$\leq 4\|y(t^\star) - x\|_{\mathscr{G}(u(\bar{s};t^\star))}^2 \cdot \|w - y\|_{\mathscr{G}(u(\bar{s};t^\star))}$$
$$+ 2\|y(t^\star) - x\|_{\mathscr{G}(y(t^\star))}^2 \cdot \|w - y\|_{\mathscr{G}(y(t^\star))} .$$

Note that

$$\|u(\bar{s}; t^\star) - x\|_{\mathscr{G}(x)} = \bar{s} \cdot \|y(t^\star) - x\|_{\mathscr{G}(x)} .$$

Since $y, w \in \mathcal{E}_x^{\mathscr{G}}(1)$ which is convex subset of $\mathcal{K}$, $y(t^\star) \in \mathcal{E}_x^{\mathscr{G}}(1)$, and this implies that $u(\bar{s}; t^\star) \in \mathcal{E}_x^{\mathscr{G}}(1)$ as well. Therefore, by Lemma 26(2),

$$\Delta(w; x) - \Delta(y; x) \le 6 \cdot \frac{\|y(t^\star) - x\|_{\mathscr{G}(x)}^2 \cdot \|w - y\|_{\mathscr{G}(x)}}{(1 - \|y(t^\star) - x\|_{\mathscr{G}(x)})^3} .$$

■

## Appendix E. Miscellaneous technical facts and algebraic lemmas

### E.1. Concentration inequalities

**Lemma 24** *Let $\gamma \sim \mathcal{N}(0, I_d)$. For $\varepsilon \in (0, 1)$, the event $\|\gamma\|^2 \le d \cdot \mathsf{N}_\varepsilon$ occurs with probability $1 - \varepsilon$.*

**Proof** From Laurent and Massart (2000, Lem. 1), we have for any $t > 0$,

$$\mathbb{P}\left(\|\gamma\|^2 > d \cdot \left\{1 + 2\sqrt{\frac{t}{d}} + 2\frac{t}{d}\right\}\right) \le e^{-t}$$

As $d \ge 1$, $\frac{t}{d} \le t$, and hence

$$\mathbb{P}\left(\|\gamma\|^2 > d \cdot \{1 + 2\sqrt{t} + 2t\}\right) \le e^{-t} .$$

Substituting $t = \log\left(\frac{1}{\varepsilon}\right)$ completes the proof. ■

**Lemma 25** *Let $\gamma \sim \mathcal{N}(0, I_d)$, and $v \in \mathbb{R}^d$ be a vector such that $\|v\| \le \mathfrak{B}$. For $\varepsilon \in (0, 1)$, the events $\langle v, \gamma \rangle \le \mathfrak{B} \cdot \mathsf{I}_\varepsilon$ and $\langle v, \gamma \rangle \ge -\mathfrak{B} \cdot \mathsf{I}_\varepsilon$ each occur with probability at least $1 - \varepsilon$.*

**Proof** Note that $\langle v, \gamma \rangle$ is a Gaussian random variable with mean 0 and variance $\|v\|^2$. Thus, for any $t > 0$,

$$\mathbb{P}(\langle v, \gamma \rangle < -t) = \mathbb{P}(\langle v, \gamma \rangle > t) \le e^{-\frac{t^2}{2\|v\|^2}} \le e^{-\frac{t^2}{2\mathfrak{B}^2}} .$$

Substituting $t = \mathfrak{B} \cdot \mathsf{I}_\varepsilon$ recovers the second statement. ■

### E.2. Facts about self-concordant metrics

**Lemma 26** *Consider a self-concordant metric $\mathscr{G} : \mathrm{int}(\mathcal{K}) \to \mathbb{S}_+^d$. This satisfies the following properties*

*(1) For every $x \in \mathrm{int}(\mathcal{K})$,*

$$\mathcal{E}_x^{\mathscr{G}}(1) \subseteq \mathrm{int}(\mathcal{K}) \ .$$

*(2) For any pair $x, y \in \mathrm{int}(\mathcal{K})$ such that $\|x - y\|_{\mathscr{G}(x)} < 1$, then*

$$(1 - \|x - y\|_{\mathscr{G}(x)})^2 \cdot \mathscr{G}(x) \preceq \mathscr{G}(y) \preceq \frac{1}{(1 - \|x - y\|_{\mathscr{G}(x)})^2} \cdot \mathscr{G}(x) \ .$$

*(3) If $\mathcal{K}$ does not contain a straight line, then $\mathscr{G}(x)$ is non-degenerate for all $x \in \mathrm{int}(\mathcal{K})$.*

**Proof** Define the matrix function $\mathscr{G}_\epsilon : \mathrm{int}(\mathcal{K}) \to \mathbb{S}_+^d$, where $\mathscr{G}_\epsilon(x) = \mathscr{G}(x) + \epsilon \cdot \mathrm{I}_{d \times d}$. For any $x \in \mathrm{int}(\mathcal{K})$ and $v \in \mathbb{R}^d$,

$$|\mathrm{D}\mathscr{G}_\epsilon(x)[v, v, v]| = |\mathrm{D}\mathscr{G}(x)[v, v, v]| \leq 2 \cdot \|v\|_{\mathscr{G}(x)}^3 \leq 2 \cdot \|v\|_{\mathscr{G}_\epsilon(x)}^3$$

and this shows that $\mathscr{G}_\epsilon$ is self-concordant. For a given $x \in \mathrm{int}(\mathcal{K})$, let $v \in \mathbb{R}^d$ be such that $\|v\|_{\mathscr{G}(x)} > 0$. Then, it holds that $\|v\|_{\mathscr{G}_\epsilon(x)} > 0$ as well. With such $x$ and $v$, consider the univariate function $\phi(t) = \langle v, \mathscr{G}_\epsilon(x + tv)v \rangle^{-1/2}$. The derivative $\phi'(t)$ satisfies

$$\phi'(t) = -\frac{\mathrm{D}\mathscr{G}_\epsilon(x + tv)[v, v, v]}{2\|v\|_{\mathscr{G}_\epsilon}^3} \Rightarrow |\phi'(t)| \leq 1 \ .$$

Note that $t \in (-\phi(0), \phi(0))$ belongs in the domain of $\phi$. This is due to Taylor's theorem, the observation about $\phi'$, which states for any $t$ that $\phi(t) - \phi(0) \geq -|t|$, and the fact that $\phi(t) > 0$. Therefore, any point of the form $x + tv$ for $t^2 \leq \phi(0)^2$ belongs in $\mathrm{int}(\mathcal{K})$. In other words,

$$\left\{ x + tv : t^2 \|v\|_{\mathscr{G}(x)}^2 + \epsilon \cdot t^2 \|v\|^2 \leq 1 \right\} \subseteq \mathrm{int}(\mathcal{K}).$$

As $\epsilon$ can be arbitrarily close to 0, setting $\epsilon \to 0$ proves the first part of the lemma.

For the second statement of the lemma, consider $v = y - x$ from the statement, and note that $\phi(0) > 1$ by the assumption that $\|y - x\|_{\mathscr{G}(x)} < 1$. As a result,

$$\phi(1) - \phi(0) \geq -1 \Rightarrow \frac{1}{\|y - x\|_{\mathscr{G}(y)}} \geq \frac{1}{\|y - x\|_{\mathscr{G}(x)}} - 1 \ .$$

This is equivalent to

$$\|y - x\|_{\mathscr{G}(y)} \leq \frac{\|y - x\|_{\mathscr{G}(x)}}{1 - \|y - x\|_{\mathscr{G}(x)}} \ .$$

Let $x_t = x + t \cdot (y - x)$, and define $\psi(t) = v^\top \mathscr{G}(x_t)v$ for some arbitrary $v \in \mathbb{R}^d$. Note that $x_t - x = t \cdot (y - x)$. By self-concordance of $\mathscr{G}$,

$$\psi'(t) \leq \mathrm{D}\mathscr{G}(x_t)[v, v, y - x] \leq 2 \cdot \|v\|_{\mathscr{G}(x_t)}^2 \cdot \|y - x\|_{\mathscr{G}(x_t)} \leq \frac{2\psi(t)}{t} \cdot \frac{\|y - x\|_{\mathscr{G}(x)}}{1 - t \cdot \|y - x\|_{\mathscr{G}(x)}} \ .$$

The remainder of the proof follows from the proof of Nesterov (2018, Thm. 5.1.7).

For the third assertion of the lemma, if for any $x \in \text{int}(\mathcal{K})$, $\mathscr{G}(x)$ is degenerate, then there exists $v \in \mathbb{R}^d$ that is non-zero such that $\mathscr{G}(x)v = \mathbf{0}$. Consider $\bar{x} = x + r \cdot v$ for $r \in \mathbb{R}$. Note that $\|\bar{x} - x\|_{\mathscr{G}(x)} = r \cdot \|v\|_{\mathscr{G}(x)} = 0$, and hence the line $\{x + r \cdot v : r \in \mathbb{R}\}$ belongs in $\mathcal{E}_x^{\mathscr{G}}(1)$. Since $\mathcal{E}_x^{\mathscr{G}}(1) \subseteq \text{int}(\mathcal{K})$ by Lemma 26(1), this is a contradiction of the assumption that $\mathcal{K}$ does not contain any straight lines, and therefore the assumption that there exists $x \in \text{int}(\mathcal{K})$ such that $\mathscr{G}(x)$ is degenerate is false. ∎

### E.3. Miscellaneous algebraic lemmas

**Lemma 27**  *Let $f(x) = x - 1 - \log(x)$. Then, for any $a \in (0,1)$,*

$$\max_{x \in [a, a^{-1}]} f(x) \leq \frac{(a-1)^2}{a} \ .$$

**Proof**  For we have for every $x > 0$ (see Srinivasan et al. (2024, Lem. 22) for a proof)

$$f(x) \leq \frac{(x-1)^2}{x} \quad \Rightarrow \quad \max_{x \in [a, a^{-1}]} f(x) \leq \max_{x \in [a, a^{-1}]} \frac{(x-1)^2}{x} \ .$$

The function $\frac{(x-1)^2}{x}$ is convex, and hence the maximum over $[a, a^{-1}]$ is attained at the end points.

$$\max_{x \in [a, a^{-1}]} \frac{(x-1)^2}{x} = \max \left\{ \frac{(a-1)^2}{a}, \frac{(a^{-1}-1)^2}{a^{-1}} \right\} = \frac{(a-1)^2}{a} \ .$$

∎

**Lemma 28**  *Let $f(x) = |(1-x)^2 - 1|$. If $x \in [0,4]$,*

$$f(x) \leq 2x$$

**Proof**

$$\{(1-x)^2 - 1\}^2 = \{2x - x^2\}^2 = 4x^2 + x^4 - 4x^3 \leq 4x^2$$

since $x^4 \leq 4x^3$ for $x \in [0,4]$. ∎

**Lemma 29**  *For any $t \in [0, 0.3]$,*

$$\log(1-t) \geq -\frac{3}{2}t \ .$$

**Proof**  Let $f(t) = e^{-\frac{3}{2}t} + t - 1$. The derivative of $f(t)$ is $-\frac{3}{2} \exp(-\frac{3}{2}t) + 1$. For any $t \in [0, 0.3]$, $f'(t) < 0$, and consequently,

$$f(t) \leq f(0) = 0 \Rightarrow e^{-\frac{3}{2}t} + t - 1 \leq 0 \Leftrightarrow \log(1-t) \geq -\frac{3}{2}t \ .$$

∎

**Lemma 30** *For any $t \in [0, 0.3]$,*

$$t^2 - \frac{t^2}{(1-t)^2} \geq -6t^3 .$$

**Proof** Through algebraic simplifications,

$$t^2 - \frac{t^2}{(1-t)^2} = \frac{t^4}{(1-t)^2} - \frac{2t^3}{(1-t)^2} \geq -\frac{2t^3}{(1-t)^2} \geq -2t^3 \cdot \frac{17}{8} \geq -6t^3 .$$

∎

**Lemma 31** *Let $a, b \geq 0$ and $d \geq 1$. If $x \leq V \cdot \min\left\{\frac{1}{a^2}, \frac{1}{a^{2/3}}, \frac{1}{(a \cdot b)^{2/3}}, \frac{1}{d^3}, \frac{1}{b \cdot d}\right\}$ for $V \geq 0$, then each of the terms*

$$\left\{ \begin{array}{ccc} a^2 \cdot x & a^2 b \cdot x^2 & bd \cdot x \\ ab \cdot x^{3/2} & ad \cdot x & d^3 \cdot x \\ ad^2 \cdot x^{3/2} & a^3 \cdot x^2 & a^{3/2} \cdot x^{5/4} \end{array} \right.$$

*are bounded from above as a function of $V$ alone.*

**Proof** We make use the following observation: for $t \geq 0$ and $p > 0$, $\min\{t, t^{-p}\} \leq 1$.

- $a^2 \cdot x \leq a^2 \cdot V \cdot \frac{1}{a^2} = V$.

- $a^2 b \cdot x^2 \leq V^2 \cdot \min\left\{\frac{a^2 b}{a^4}, \frac{a^2 b}{a^{4/3} \cdot b^{4/3}}\right\} = V^2 \cdot \min\left\{\frac{b}{a^2}, \frac{a^{2/3}}{b^{1/3}}\right\} \leq V^2$.

- $bd \cdot x \leq bd \cdot V \cdot \frac{1}{bd} = V$.

- $ab \cdot x^{3/2} \leq ab \cdot V^{3/2} \cdot \frac{1}{ab} = V^{3/2}$.

- $ad \cdot x \leq V \cdot \min\left\{\frac{ad}{a^2}, \frac{ad}{d^3}\right\} \leq V \cdot \min\left\{\frac{d}{a}, \frac{a}{d^2}\right\} \leq V \cdot \min\left\{\frac{d}{a}, \frac{a}{d}\right\} \leq V$.

- $d^3 \cdot x \leq d^3 \cdot V \cdot \frac{1}{d^3} = V$.

- $ad^2 \cdot x^{3/2} \leq V^{3/2} \cdot \min\left\{\frac{ad^2}{a^3}, \frac{ad^2}{d^{9/2}}\right\} \leq V^{3/2} \cdot \min\left\{\frac{d^2}{a^2}, \frac{a}{d^{7/2}}\right\} \leq V^{3/2} \cdot \min\left\{\frac{d^2}{a^2}, \frac{a}{d}\right\} \leq V^{3/2}$.

- $a^3 \cdot x^2 \leq V^2 \cdot \min\left\{\frac{a^3}{a^4}, \frac{a^3}{a^{4/3}}\right\} = V^2 \cdot \min\left\{\frac{1}{a}, a^{5/3}\right\} \leq V^2$.

- $a^{3/2} \cdot x^{5/4} \leq V^{5/4} \cdot \min\left\{\frac{a^{3/2}}{a^{5/2}}, \frac{a^{3/2}}{a^{5/6}}\right\} = V^{5/4} \cdot \min\left\{\frac{1}{a}, a^{2/3}\right\} \leq V^{5/4}$.

∎

**Lemma 32** *Let $a, b \geq 0$, $c \geq 4$, and $d \geq 1$. If $x \leq V \cdot \min\left\{\frac{1}{a^2}, \frac{1}{(a \cdot b)^{2/3}}, \frac{1}{(a \cdot c)^{2/3}}, \frac{1}{d \cdot a}, \frac{1}{d \cdot b}, \frac{1}{d \cdot c}\right\}$, then each of the terms*

$$\left\{ \begin{array}{cccc} a^2 \cdot x & a^2 b \cdot x & bd \cdot x & ab \cdot x^{3/2} \\ a\sqrt{d} \cdot x & d \cdot x & a^2 c \cdot x^2 & cd \cdot x \\ ac \cdot x^{3/2} & a^3 \cdot x^2 & ad \cdot x & a^2 \cdot x^{3/2} \end{array} \right.$$

*are bounded from above as a function of $V$ alone.*

**Proof** We again make use the following observation: for $t \geq 0$ and $p > 0$, $\min\{t, t^{-p}\} \leq 1$.

- $a^2 \cdot x \leq a^2 \cdot V \cdot \frac{1}{a^2} = V$.

- $a^2 b \cdot x^2 \leq V^2 \cdot \min\left\{\frac{a^2 b}{a^4}, \frac{a^2 b}{a^{4/3} \cdot b^{4/3}}\right\} = V^2 \cdot \min\left\{\frac{b}{a^2}, \frac{a^{2/3}}{b^{1/3}}\right\} \leq V^2$.

- $bd \cdot x \leq bd \cdot V \cdot \frac{1}{bd} = V$.

- $ab \cdot x^{3/2} \leq ab \cdot V^{3/2} \cdot \frac{1}{ab} = V^{3/2}$.

- $a\sqrt{d} \cdot x \leq V \cdot \frac{a\sqrt{d}}{a \cdot d} \leq V$.

- $d \cdot x \leq V \cdot \frac{d}{d \cdot c} \leq \frac{V}{4}$.

- $a^2 c \cdot x^2 \leq V^2$ (analogous to $a^2 b \cdot x^2 \leq V^2$).

- $cd \cdot x \leq V \cdot \frac{cd}{cd} = V$.

- $ac \cdot x^{3/2} \leq V^{3/2}$ (analogous to $ab \cdot x^{3/2} \leq V^{3/2}$).

- $a^3 \cdot x^2 \leq V^2 \cdot \min\left\{\frac{a^3}{a^4}, \frac{a^3}{d^2 a^2}\right\} \leq V^2 \cdot \min\left\{\frac{1}{a}, \frac{a}{d^2}\right\} \leq V^2 \cdot \min\left\{\frac{1}{a}, a\right\} \leq V^2$.

- $ad \cdot x \leq V \cdot \frac{ad}{ad} = V$.

- $a^2 \cdot x^{3/2} \leq a^2 c \cdot x^{3/2} \leq V^{3/2}$.

∎

