# OpenReview forum: "High-accuracy sampling from constrained spaces with the Metropolis-adjusted Preconditioned Langevin Algorithm"
_algorithmiclearningtheory.org/ALT/2025/Conference — ALT 2025_

### Official Review · Reviewer_A1KC · 2024-11-08
**Review of MAPLA**

**Rating:** 7
**Confidence:** 4

**Review:**

***Summary***
The paper proposes and analyses a Metropolised version of the preconditioned Langevin algorithm, which can be viewed as a crude version of Riemannian Langevin. Rates are obtained under various concordance/convexity assumptions, which largely match those of zeroth order samplers.

***Strengths/Weaknesses***
I did not check all the details of the analysis, but broadly the results seem to be believable and quite analogous to those derived for the Dikin walk.

The paper does a good job of explaining the differences between the various notions of self-concordance which have since proliferated in the literature.

The new isoperimetric inequality (Lemma 17) is a solid contribution and will be useful for the existing literature.

The paper is forthright that its result is not better in asymptotic order than that of Dikin walk. While the analytic route to getting there was not trivial and is a worthy contribution in itself, it would be much better if the paper could at least speculate on what might be a plausible route to improving the rate estimates.

Furthermore, it would be helpful if the main text contained additional motivational information on the algorithm, as compared to the Dikin walk.

***Questions***
Here is a naive question: what is the expected rate of PLA without any Metropolis adjustment? Can one hope to improve the dimension dependence of this algorithm if an underdamped version is used instead? Finally, is there a proximal variant of this algorithm, which alternates some type of Gibbs sampling procedure?

The final remark about combining with Gaussian cooling should probably cite Kook and Vempala rather than Cousins and Vempala (which is only valid for the uniform case).

Page 5: $\mu$ in (WLD) is confusing for a deterministic function, since $\mu$ was used earlier for a measure

The equation for $\rho$ is also very small in (WLD) for some reason.

**Paper Award:**

No

---

> ### Author Response · Authors · 2024-11-20
>
> Thank you for your review of our paper. With regard to improving rate estimates in the paper, an open question which we think has scope beyond just this work is understanding concentration phenomenon of the gradients of the potential $f$ when $f$ satisfies the curvature upper and lower bounds.
> We believe that showing this would eliminate the dependence on $\beta$ and permit a wholly $s$-conductance based analysis, similar to the one performed to prove theorem 4.3.
> For additional motivation for the development of MAPLA, we would like to point you to our response [to Reviewer iZr8](https://openreview.net/forum?id=L68sXPqqjE&noteId=28fspQQD4x), which also presents a more comprehensive comparison to MAMLA.
>
> Since PLA can be viewed as the "primal" version of the mirror Langevin algorithm (formed by the Euler-Maruyama discretisation of the mirror Langevin dynamics), we expect that PLA and MLA would have similar guarantees.
> The most recent guarantee for MLA were given in [3], under a rather restrictive assumption on the mirror function (called modified self-concordance), which is a consequence of their analysis technique.
> With this same analysis technique and a projection step, we feel that it would be possible to provide guarantees for PLA, albeit under a modified self-concordance assumption on the metric, although this is something that we have deferred for future work.
> A proximal variant of this algorithm would indeed be interesting, particularly along the lines of the more recent In-And-Out algorithm in [4], which has a dependence on the condition number of domain much like Hit-And-Run and BallWalk, and incorporating the metric in it could perhaps avoid this condition number dependence.
> In essence, this is what DikinWalk attempted to do in comparison to BallWalk; it replaced sampling from a Euclidean ball to sampling from a Dikin ellipsoid.
>
> We will address the typographical errors that you have pointed out in the final version of this draft, thank you for pointing them out.
>
> [3] Li, Tao, Vempala, Wibisono, The Mirror Langevin Algorithm Converges with Vanishing Bias, In International Conference on Algorithmic Learning Theory, 2022.
>
> [4] Kook, Vempala, Zhang, In-And-Out: Algorithmic Diffusions for sampling convex bodies, In Neural Information Processing Systems, 2024.

---

### Official Review · Reviewer_LALf · 2024-11-08
**Metropolis-adjusted preconditioned Langevin algorithm with good mixing time guarantees**

**Rating:** 8
**Confidence:** 4

**Review:**

This paper proposes and analyzes a new MCMC scheme for sampling from Gibbs distributions supported on a convex set (the authors refer to this as the constrained setting). The target distribution $\Pi$ has a density $\pi(x)$ proportional to $\exp(-f(x)){\bf 1}_{\cal K}$, where $f$ is the potential function and ${\cal K}$ is the convex constraint set. The scheme uses a preconditioned Langevin algorithm to generate proposal moves, then applies the Metropolis-Hastings filter. This results in an unbiased scheme (its steady-state distribution is equal to $\Pi$). The local geometry of ${\cal K}$ is encoded in a Riemannian metric tensor, which in turn determines the preconditioning. Under a variety of regularity assumptions on the potential, the metric tensor, and the constraint set, the authors obtain precise upper bounds on the total variation mixing time of their scheme from a suitable warm-start initialization. The discussion of prior art is thorough, including a detailed discussion of computational complexity per iteration as compared with the recent Dikin Walk scheme of Kook and Vempala (in turn inspired by the work of Narayanan on uniform sampling from convex polytopes), as well as a discussion of the relation between their proposed scheme and metropolis-adjusted mirror Langevin algorithm.


Overall, this is a high-quality contribution to the literature on iterative stochastic algorithms for sampling. The authors provide good motivation in terms of applications and the discussion of relevant challenges. The comparison with existing approaches is informative, highlighting the relative advantages and disadvantages. Overall, the paper is relatively easy to follow, although the plethora of definitions of self-concordance is somewhat hard to parse. The authors do a decent job motivating these definitions and explain where one can reap the benefits of stronger (more restrictive) notions of self-concordance. The discussion of the cases where the new definitions coincide with existing ones is also helpful. Given the authors' goal to present a reasonably complete picture, I can't fault them for not including any proofs in the 12 pages of the main submission. The overall strategy underlying the proof is a conductance-based analysis along the lines of Lovász and Simonovits, which is more or less standard as far as Metropolis-Hastings MCMC goes. The main innovations here are in the definitions and in the tradeoffs between performance and complexity; some new isoperimetric inequalities are also established along the way. While I have not checked all the proofs minutely, I have verified the major steps, and am fairly certain the analysis is correct.

**Paper Award:**

No

---

> ### Author Response · Authors · 2024-11-20
>
> Thank you for your review of our paper, and we appreciate the positive feedback.
> We will try to ease the exposition around the non-conventional notions of self-concordance, particularly around lower trace self-concordance and average self-concordance to enable a more streamlined presentation.

---

### Official Review · Reviewer_8F7S · 2024-11-15
**Accept**

**Rating:** 7
**Confidence:** 3

**Review:**

This paper analyzes the Metropolis-adjusted Preconditioned Langevin Algorithm (MAPLA) for the purposes of sampling from convex bodies with respect to some underlying metric. As the authors point out, the naive Preconditioned Langevin Algorithm (PLA) is biased even for vanishing stepsize, and the additional Metropolis adjustment corrects for this. As the authors also helpfully point out, MAPLA can be thought of as an interpolation between the vanilla DinkinWalk for sampling from convex bodies, and the Manifold Metropolis-adjusted Langevin algorithm (ManifoldMALA). They require several technical assumptions on the underlying (preconditioning) metric $\mathcal{G}$ to make their analysis work, though there are related works with similar if not identical assumptions. At the end of the day, they provide mixing-time guarantees for MAPLA which depend on the underlying dimension "d" and these technical assumptions on the metric. Some of these are recover prior work, namely when $\mathcal{G}$ is the Hessian of the potential of a potential in a certain class, and improve upon others (e.g., Riemannian Hamiltonian Monte Carlo) in the dimension.

The paper is clearly written (minor typos here and there), even for people like myself who are not the most familiar with this body of work. The background section is particularly clearly written. My only criticism would be that the comparison to Metropolis-adjusted Mirror Langevin Algorithm (MAMLA) was not thorough enough, as the main difference between these works is the generality of the preconditioner. In that sense, this paper supersedes MAMLA, though it is not clear to me (someone not in this field) if the proofs are morally the same, which diminishes the contributions of this work (I will also confess that I did not read the proofs in detail, so please take the comment with a grain of salt). Nevertheless, I enjoyed this paper, and I recommend accept.

Comments:
- I recommend changing the wording near the introduction of (PLA) in the background section. As written, the equation is just thrown in-between paragraphs and some structure is lost. Maybe add the equation just after the italicized "*preconditioned Langevin algorithm*" in the text

**Paper Award:**

No

---

> ### Author Response · Authors · 2024-11-20
>
> Thank you for you remarks on our paper, and we are glad that you enjoyed reading it.
> We would like to you point to our rebuttal [to Reviewer iZr8](https://openreview.net/forum?id=L68sXPqqjE&noteId=28fspQQD4x) wherein we have included a more thorough comparison to MAMLA both algorithmically and theoretically.
> The proof technique that we use is rather standard as Reviewer LALf points, which is based on the classical result of Lovasz and Simonovits (1993), and the one-step overlap technique due to Lovasz (1999).
> The salient differences are in establishing the isoperimetric inequality in terms of the potential $f$ and the metric $\mathscr{G}$ (Lemma 17), and identifying that ``self-concordance++'' that was useful to analyse DikinWalk can also be used to analyse MAPLA, which impacts the details of how we show the one-step overlap.
> We would like to emphasise that this is not a direct consequence of the analysis of DikinWalk.
>
> Thank you for the suggestion regarding the placement of the definition of PLA; we will make this change in the final version.

---

### Official Review · Reviewer_iZr8 · 2024-11-16
**Review for the paper "High-accuracy sampling from constrained spaces with the Metropolis-adjusted Preconditioned Langevin Algorithm"**

**Rating:** 4
**Confidence:** 3

**Review:**

This paper introduces a first-order sampling method termed the Metropolis-adjusted Preconditioned Langevin Algorithm (MAPLA) for approximate sampling from a target distribution $p(x) \propto e^{-f(x)}$, where the support is a proper convex subset $\mathcal{K}$ in $\mathbb{R}^{d}$. MAPLA extends the Metropolis-adjusted Langevin Algorithm by incorporating a preconditioner matrix $G$ into the Langevin dynamics update, followed by a Metropolis-Hastings acceptance/rejection step. The authors derive a mixing time bound under standard conditions on $f$ and $G$.
However, my primary concern with this paper is its lack of novelty. The results closely mirror those of Srinivasan et al. (2024), who analyzed the Metropolis-adjusted Mirror Langevin Algorithm (MAMLA) and provided mixing time bounds under standard assumptions on $
f$ and the mirror map. While MAPLA offers a different perspective by using a preconditioning matrix $G$, the paper does not adequately address critical questions such as how to choose $G$, when MAPLA should be preferred over other methods, or the algorithm’s practical advantages.

Moreover, the proof techniques employed here largely follow those of Srinivasan et al. (2024) without improving upon the existing results. Although the paper discusses two additional cases involving specific choices of $G$ and $f$, these cases are not sufficient to demonstrate substantial novelty. The general results remain comparable to those of MAMLA, differing primarily in how smoothness and strong convexity are defined—in terms of the mirror map for MAMLA versus the preconditioner matrix $G$ for MAPLA.

Given the limited contribution and the lack of a clear argument for the advantages of MAPLA over existing approaches, I cannot recommend acceptance at this time.

**Paper Award:**

No

---

> ### Author Response · Authors · 2024-11-20
>
> Thank you for your review of this work.
> We would like to help better contextualise our contributions in light of the recent work of Srinivasan et al (2024) on MAMLA.
> As you point out, MAPLA is based on a different perspective of MAMLA, analogous to how natural gradient methods are generalisations of mirror methods (see [1]).
> This is also why it is unsurprising that the guarantees in Theorem 4.1 in our work and Theorem 6 in Srinivasan et al (2024)) are similar.
> However, these guarantees are very general; they hold for a self-concordant metric and a self-concordant mirror function respectively.
>
> Our proposal of MAPLA was originally motivated by two main algorithmic properties about the mirror Langevin algorithm and its Metropolis-adjusted variant: (1) the use of the inverse mirror map, and (2) applicability only when the domain is compact.
> Property (1) is very liberally assumed about mirror methods in general, however even for simple structured domains such as a polytope, the map $\nabla\phi^{\star}$ cannot be computed efficiently, and would require solving an optimisation problem defined by the Fenchel dual.
> Property (2) is essential because the dual space is $\mathbb{R}^{d}$ in this case.
> However, when the domain is not compact, this dual space is not necessarily $\mathbb{R}^{d}$ (an example: the domain $(0, \infty)$ with the mirror function $\phi(x) = -\log x$).
> In this case, the accept-reject filter would have to include an membership oracle for the dual space instead, which is non-trivial.
> MAPLA circumvents both of these concerns: we wholly work in the primal space, and only require a membership oracle for the primal space.
> When the metric $\mathscr{G}$ in MAPLA is chosen as the Hessian of the mirror function in MAMLA, MAPLA and MAMLA are equivalent, but we pursue a more general treatment with metrics (as [Reviewer 8F7S remarks accurately](https://openreview.net/forum?id=L68sXPqqjE&noteId=QcdHL9EUqz), this work supersedes MAMLA).
> Moreover, we decided to work with a Metropolis-adjusted PLA as PLA is highly likely to carry a bias, and would require a projection oracle since PLA by itself could generate samples outside $\mathcal{K}$.
>
> Theoretically, our theorems 4.2 and 4.3 were motivated by going beyond the self-concordance setting of theorem 4.1.
> This is aided by recent advances in developing metrics (also called barriers) for non-Euclidean sampling, and identifying essential sufficient conditions ("self-concordance++") [2] that are known to yield good mixing time guarantees for DikinWalk, and we found that these properties also yield good mixing time guarantees for MAPLA which was non-trivial as [noted by Reviewer A1KC](https://openreview.net/forum?id=L68sXPqqjE&noteId=KNKN1lG7RX).
> For several structured domains (polytopes, ellipsoids, $\ell_{p}$-balls), the theory of interior point methods in optimisation provides us with suitable metrics $\mathscr{G}$, for which "self-concordance++" holds.
> It is natural to ask if these sufficient conditions also transfer to the dual space and the associated metric $\nabla^{2}\phi^{\star}$ as this would result in better mixing time guarantees for MAMLA as it is based in the dual space, but we feel that this is non-trivial, and is an interesting theoretical avenue towards improving the guarantees for MAMLA.
> Nonetheless, this would still not spare the algorithmic difficulties associated with MAMLA as explained above.
>
> In the final version of this paper, we will include the exposition about the algorithmic motivations for studying MAPLA and comparisons to MAMLA as expanded on previously.
> We will also include a discussion about these structured domains and their metrics for a more complete picture.
> As far as practical advantages of MAPLA compared to other methods, MAMLA is fundamentally constrained by the computation of $\nabla\phi^{\star}$.
> The complexity of MAPLA scales similar to that of DikinWalk, as the computational bottleneck is in generating a random vector from $\mathcal{N}(0, \mathscr{G}(x)^{-1})$ which is common to both methods.
>
> [1] Raskutti and Mukherjee, The information geometry of mirror descent, In IEEE Transactions on Information Theory, 2015.
>
> [2] Kook and Vempala, Gaussian Cooling and Dikin Walks: The Interior-Point Method for Logconcave Sampling, In Conference on Learning Theory, 2024.

---

### Author Rebuttal · Authors · 2024-11-21

We have submitted rebuttals individually to each reviewer as comments to their reviews.

---

### Meta-Review · Area_Chair_CwBG · 2024-12-13

**Recommendation:** Accept
**Confidence:** 5

**Metareview:**

This paper studied a general preconditioned version of Metropolis-adjusted Langevin Algorithm, and provided convergence guarantees. Extending upon the mirror version of this algorithm, the authors produced non-trivial results on a more general version of preconditioned that can be interpreted as a non-Hessian Riemannian metric. In particular, the extension of a natural self-concordance condition beyond the mirror setting and the analysis following is definitely a novel contribution.

The reviews are overall very positive, and the only concern raised on the novelty of the technical contributions was adequately addressed. Based on the reviews and rebuttals, I believe this paper is a welcomed addition to the sampling literature, and hence I would recommend accept.

**Paper Award:**

No